# All-Purpose Mean Estimation over ℝ: Optimal Sub-Gaussianity with Outlier Robustness and Low Moments Performance

**Jasper C.H. Lee** [1]   **Walter McKelvie** [2 3 4]   **Maoyuan Song** [2]   **Paul Valiant** [2]

## Abstract

We consider the basic statistical challenge of designing an "all-purpose" mean estimation algorithm that is recommendable across a variety of settings and models. Recent work by Lee & Valiant (2022) introduced the first 1-d mean estimator whose error in the standard finite-variance+i.i.d. setting is optimal even in its constant factors; experimental demonstration of its good performance was shown by Gobet et al. (2022). Yet, unlike for classic (but not necessarily practical) estimators such as median-of-means and trimmed mean, this new algorithm lacked proven robustness guarantees in other settings, including the settings of adversarial data corruption and heavy-tailed distributions with infinite variance. Such robustness is important for practical use cases. This raises a research question: is it possible to have a mean estimator that is robust, *without* sacrificing provably optimal performance in the standard i.i.d. setting? In this work, we show that Lee and Valiant's estimator is in fact an "all-purpose" mean estimator by proving:

(A) It is robust to an $\eta$-fraction of data corruption, even in the strong contamination model; it has optimal estimation error $O(\sigma\sqrt{\eta})$ for distributions with variance $\sigma^2$.

(B) For distributions with finite $z^{\text{th}}$ moment, for $z \in (1, 2)$, it has optimal estimation error, matching the lower bounds of Devroye et al. (2016) up to constants.

We further show (C) that outlier robustness for 1-d mean estimators in fact implies neighborhood optimality, a notion of beyond worst-case and distribution-dependent optimality recently introduced by Dang et al. (2023). Previously, such an

optimality guarantee was only known for median-of-means, but now it holds also for all estimators that are simultaneously *robust* and *sub-Gaussian*, including Lee and Valiant's, resolving a question raised by Dang et al. Lastly, we show (D) the asymptotic normality and efficiency of Lee and Valiant's estimator, as further evidence for its performance across many settings.

## 1. Introduction

The past decade has seen reinvigorated interest in understanding fundamental statistical problems, and in particular, the foundational problem of mean estimation: given $n$ samples from an unknown distribution, and an allowed failure probability $\delta$, what is the most accurate estimate of the distribution mean? The sample mean is the traditional estimate, yet, its sensitivity to extreme values makes it an unreliable estimator in practice. Other classic estimators, for example the well-known median-of-means estimator (Alon et al., 1999; Nemirovsky & Yudin, 1983; Jerrum et al., 1986), also suffer from poor empirical performance.

Recent work by Lee & Valiant (2022), sitting in a line of research initiated by the seminal paper of Catoni (2012), gave the first 1-dimensional mean estimator whose estimation error is optimal (i.e. sub-Gaussian error) and tight even in the leading constant. Gobet et al. (2022) further demonstrated its good empirical performance against other competing estimators.

**Fact 1.1** (Lee & Valiant 2022). *Given any distribution $D$ with mean $\mu$ and variance $\sigma^2$ (both unknown), parameters $n, \delta > 0$, let $X$ be a set of $n$ independent samples from $D$. Then, with probability at least $1 - \delta$ over the sampling process, Estimator 1 on input $\delta$ and $X$ will output an estimate $\hat{\mu}$ with error at most*

$$|\hat{\mu} - \mu| \leq \sigma \cdot \left( (\sqrt{2} + o(1)) \sqrt{\frac{\log \frac{1}{\delta}}{n}} \right)$$

*Here, the $o(1)$ term tends to $0$ as $\left( \frac{\log \frac{1}{\delta}}{n}, \delta \right) \to (0, 0)$ and, crucially, is independent of $D$.*

[1]University of California, Davis [2]Purdue University [3]Columbia University [4]Harvard University. Correspondence to: Jasper C.H. Lee <jasperlee@ucdavis.edu>.

*Proceedings of the 42nd International Conference on Machine Learning*, Vancouver, Canada. PMLR 267, 2025. Copyright 2025 by the author(s).

**Estimator 1** Mean Estimator of Lee and Valiant

Inputs: $n$ i.i.d. samples $\{x_i\}$; Confidence parameter $\delta$.

1. Compute an initial estimate $\kappa$ using **any** sub-Gaussian, scale-and-translation-equivariant and robust mean estimator (cf. Facts A.18 and A.23). One possible choice is the median-of-means estimate: evenly partition the data into $\log \frac{1}{\delta}$ groups and let $\kappa$ be the median of the set of means of the groups;

2. Find the solution $\alpha$ to the monotonic, piecewise-linear equation

$$\sum_i \min(\alpha(x_i - \kappa)^2, 1) = \tfrac{1}{3}\log\tfrac{1}{\delta}$$

3. Output:

$$\hat{\mu} = \kappa + \tfrac{1}{n}\sum_i (x_i - \kappa)(1 - \min(\alpha(x_i - \kappa)^2, 1))$$

The constant "$\sqrt{2}$" in Fact 1.1 is optimal, with matching lower bounds. Essentially, Estimator 1 uses Step 1 to find a preliminary estimate (e.g. using median-of-means, but it can be other choices as well), which is then refined in Steps 2 and 3 into a better estimate—whose accuracy is optimal even in the leading multiplicative constant. Even though the form of the estimator is somewhat complicated—including a function inversion in Step 2—the estimator can in fact be computed easily in quasilinear time via a sorting operation, or even in linear time using a more subtle algorithm. See (Lee & Valiant, 2022) for more discussion of the design and analysis of Estimator 1.

While Fact 1.1 is a strong recommendation for Estimator 1, getting optimal performance in this regime is not the only practical desideratum for an estimator. Practitioners have long observed that some real-world distributions are very heavy-tailed, with variances that are gigantic (thus rendering guarantees like Fact 1.1 effectively useless), but whose lower moments—between the first and second moments—might be much more bounded. Further, real-world data can be subject to random or even adversarial corruption. Performance guarantees in these more demanding settings are therefore critical for practice. However, while there are relatively straightforward folklore analyses of classic algorithms in these settings—including, notably, median-of-means and trimmed mean—Lee and Valiant's estimator has not been studied in these contexts. The breadth of guarantees shown for these classic estimators might therefore lead researchers and practitioners to choose either of them over Lee and Valiant's estimator, despite the latter being a more accurate estimator in standard finite variance i.i.d. sample settings.

The natural and pressing research question, then, is

*Is it possible for a mean estimator to satisfy the important robustness guarantees of classic estimators,* **without** *losing the strong and optimal-constant performance that the Lee and Valiant estimator enjoys in the standard setting?*

In this paper, we show 4 strong positive results demonstrating the robustness and performance of Estimator 1:

**(A)** In the presence of $\eta$-fraction of arbitrary adversarial corruption, the error of Estimator 1 degrades by only $O(\sigma\sqrt{\eta})$ from the sub-Gaussian error guaranteed in Fact 1.1. This extra error term is optimal under the assumption that the distribution has (finite) variance $\sigma^2$.

**(B)** When samples are drawn from a distribution with finite $z^{\text{th}}$ moment, for $z \in (1, 2)$, Estimator 1 achieves error matching the lower bound of Devroye et al. (2016) up to constants.

**(C)** Estimator 1 enjoys a fine-grained beyond-worst-case notion of optimality called *neighborhood optimality*, recently introduced by Dang et al. (2023). Previously, only median-of-means was known to be neighborhood optimal, and our result answers the question raised by Dang et al. on whether more modern estimators are also neighborhood optimal.

**(D)** Under the standard finite variance and i.i.d. setting, Lee and Valiant's estimator is asymptotically normal and efficient.

We emphasize that these new results hold without any changes at all to either the structure or the parameters of Estimator 1. That is, the estimator does not need to know that it is being expected to perform in these challenging settings. The guarantees we show also smoothly revert to Fact 1.1 as the amount of corruption decreases to 0, or as the $z^{\text{th}}$-moment assumption tends to $z \to 2$.

These results, in aggregate, demonstrate that Lee and Valiant's estimator is in fact "all-purpose", enjoying the same breadth of guarantees as either median-of-means or trimmed mean, while also having what is essentially the smallest possible estimation error (even in the leading constant) in the standard setting. Thus, Lee and Valiant's estimator should be used in practice over existing estimators.

## 2. Our Results at a Glance

We now formally state our technical contributions.

### 2.1. Outlier robustness

Lee and Valiant's estimator is robust against data corruption from the *strong contamination model*, the most adversarial data corruption model in the literature.

**Definition 2.1** (Strong Contamination Model)**.** Given a corruption parameter $\eta$ and a distribution $D$ on the uncorrupted data, an algorithm gets a set of $n$ $\eta$-corrupted samples from $D$ as follows. The algorithm specifies $n$ and nature draws

$n$ i.i.d. samples from $D$. Then, an arbitrarily powerful adversary can inspect the $n$ samples from $D$, and arbitrarily replace $\lceil \eta n \rceil$ of them before giving the (new) set of samples to the algorithm, with no indication of which samples were corrupted.

In the setting of $\eta$-corruption, when we only assume that $D$ has a finite variance $\sigma^2$, it is well-known (Diakonikolas & Kane, 2023) that the minimum estimation error is $\Omega(\sigma\sqrt{\eta})$ even if there are infinitely many samples. Our analysis of Lee and Valiant's estimator shows that it achieves this estimation error bound, as long as the amount of corruption is $O(\log\frac{1}{\delta}/n)$—where $\delta$, the desired failure probability, is a parameter under the user's control. Put differently, the $\delta$ parameter for Estimator 1 can be viewed as "dual use", parameterizing not just the allowed failure probability of the estimator, but also its desired robustness, expressing the maximum tolerated level of corruption.

We emphasize that in the following theorem, Estimator 1 does not need to know the precise value of $\eta$, the fraction of corruption. It only makes the assumption that $\eta \leq O(\log\frac{1}{\delta}/n)$, or in other words, Estimator 1 *adapts* to the level of corruption.

**Theorem 2.2.** *Given any distribution $D$ with mean $\mu$ and variance $\sigma^2$, parameters $n, \delta, \eta > 0$, let $\tilde{X}$ be a set of $n$ $\eta$-corrupted samples from $D$.*

*Suppose both $\frac{\log\frac{1}{\delta}}{n}$ and $\delta$ are bounded by some small universal constant, and suppose $\eta \leq \frac{1}{24n}\log\frac{1}{\delta}$. Then, with probability at least $1 - \delta$ over the sampling process, Estimator 1 on input $\delta$ and $\tilde{X}$ will output an estimate $\hat{\mu}$ with error at most*

$$|\hat{\mu} - \mu| \leq \sigma \cdot \left( (\sqrt{2} + o(1))\sqrt{\frac{\log\frac{1}{\delta}}{n}} + 222\sqrt{\eta} \right)$$

In the above theorem, Estimator 1 gets $\delta$ as its input parameter, and fails with probability $\delta$. Accordingly, the estimator error is the sum of the sub-Gaussian error term from Fact 1.1, and a new "robustness" term of $O(\sigma\sqrt{\eta})$. On the other hand, the standard robustness analysis and guarantee of trimmed mean takes a different form (see Oliveira et al. (2025))—if the desired failure probability is $\delta'$ and the corruption parameter is $\eta$, then the number of trimmed samples is chosen as $\log\frac{1}{\delta} = \Theta(\log\frac{1}{\delta'} + \eta n)$. We give an analogous theorem for Estimator 1, where again we preserve the optimal constant of $\sqrt{2}$ in the sub-Gaussian error term. Comparing Theorem 2.2 with Theorem 2.3, the latter asks for a failure probability $\delta' > \delta$, but correspondingly has a smaller sub-Gaussian error term (since $\log\frac{1}{\delta'} < \log\frac{1}{\delta}$).

**Theorem 2.3.** *Given any distribution $D$ with mean $\mu$ and variance $\sigma^2$, parameters $n, \delta, \eta > 0$, let $\tilde{X}$ be a set of $n$ $\eta$-corrupted samples from $D$.*

*Suppose both $\frac{\log\frac{1}{\delta}}{n}$ and $\delta$ are bounded by some small universal constant, and suppose $\eta \leq \frac{1}{9n}\log\frac{1}{\delta}$. Let $\delta'$ be so that $\frac{1}{3}\log\frac{1}{\delta} = \frac{1}{3}\log\frac{1}{\delta'} + 3\eta n$. Then, with probability at least $1 - \delta'$ over the sampling process, Estimator 1 on input $\delta$ and $\tilde{X}$ will output an estimate $\hat{\mu}$ with error at most*

$$|\hat{\mu} - \mu| \leq \sigma \cdot \left( (\sqrt{2} + o(1))\sqrt{\frac{\log\frac{1}{\delta'}}{n}} + (135 + o(1))\sqrt{\eta} \right)$$

We emphasize again that, similar to Theorem 2.2, Theorem 2.3 says that Estimator 1 adapts to the value of $\eta$ without knowing what it is precisely. Namely, Theorem 2.3 simultaneously holds for all values of $\delta'$ even though the algorithm is only given a fixed $\delta$.

We sketch the proof of the above two theorems in Section 4 and give full details in Appendix A. While the $\sqrt{2}$ constant in the sub-Gaussian terms is tight, we expect that the constants in the robust error terms (222 and 135) might be significantly improved with a different proof strategy.

In Appendix B, we also show that Estimator 1 is robust in the Huber and Total Variation Contamination Models, in addition to the Strong Contamination Model above.

Median-of-means also has guarantees analogous to the above, but they are folklore. See the proofs of Theorem 2.2 and Theorem 2.3 in Appendix A.4 and Appendix A.3 respectively for the formal statements and proofs of these folklore guarantees. Trimmed mean also gives analogous robustness guarantees, but current analysis (see Oliveira et al. (2025)) requires precise knowledge of $\eta$ in order to match the optimal rate. In contrast, although Estimator 1 does require an upper bound on $\eta$ (from the necessary assumption that $\eta \leq \frac{1}{24n}\log\frac{1}{\delta}$), the accuracy will gracefully improve if the actual corruption rate is less than the pessimistic upper bound.

We further stress that, while median-of-means and trimmed mean both have relatively straightforward and folklore proofs of robustness, the analysis required for the Lee and Valiant estimator is much more intricate—this is due to the fact that Lee and Valiant's estimator is (a lot) more complicated, in order to yield optimal constants in the i.i.d. setting.

### 2.2. Low moment performance

Next, we study the performance of Lee and Valiant's estimator when given data drawn from a distribution that only has finite low moments, specifically, moments between 1 and 2.

**Theorem 2.4.** *Given any distribution $D$ with mean $\mu$ and $z^{th}$ moment $M_z$ for some $z \in (1, 2)$, let $X$ be a set of $n$ i.i.d. samples from $D$. Then, with probability at least $1 - \delta$ over the randomness of $X$, Estimator 1 on input $\delta$ and $X$*

*will output an estimate $\hat{\mu}$ with error at most*

$$|\hat{\mu} - \mu| \leq (M_z)^{\frac{1}{z}} \cdot (1 + o(1)) \left( c_z \frac{\log \frac{1}{\delta}}{n} \right)^{1 - \frac{1}{z}}$$

*where $c_z = 2(5.6)^{\frac{1}{z-1} - 1}$. Here, the $o(1)$ term tends to $0$ as $\left( \frac{\log \frac{1}{\delta}}{n}, \delta \right) \rightarrow (0, 0)$, in a manner independent of $D$ and independent of $z$.*

The above result matches the lower bounds shown by Devroye et al. (2016) up to constants. As $z \rightarrow 2$, the guarantee converges to Fact 1.1. Analogous results (up to constants) were shown for median-of-means in Bubeck et al. (2012), but the multiplicative constant we achieve for Estimator 1, $(2(5.6)^{\frac{1}{z-1} - 1})$ is better than that of median-of-means $(8\sqrt{3}(12)^{\frac{1}{z-1} - 1})$ across all values of $z \in (1, 2]$.

## 2.3. Neighborhood optimality

Going beyond the worst-case analysis of Estimator 1, we give a finer-grained analysis of its performance. We provide finite-sample error bounds that *optimally adapt* to the underlying distribution on an instance-by-instance basis, which is far stronger than just having optimal dependence on the *variance*.

Recent work by Dang et al. (2023) gave a first study of the fine-grained optimality of 1-dimensional mean estimators, providing upper and lower bounds that match up to constants. At a high level, they showed that sub-Gaussian error bounds are essentially all one can hope for, for any distribution with a finite mean. More specifically, they define an error rate function $\epsilon_{n,\delta}(D)$ over distributions $D$, and prove that **i)** median-of-means attains this error bound; and **ii)** for any distribution $D$, there exists a "reasonable" counterpart distribution $D'$ such that no algorithm can distinguish between the distributions, and thus no estimator can simultaneously get error $\ll \epsilon_{n,\delta}(D)$ using samples from $D$, while also getting error $\ll \epsilon_{n,\delta}(D')$ using samples from $D'$. We define $\epsilon_{n,\delta}(D)$ below. The combination of lower and upper bounds of i) and ii) is extended into a new optimality notion called *neighborhood optimality* by Dang et al.

Intuitively, these are bounds that are optimal in an instance-by-instance basis, because property (ii) shows that *no algorithm can get error better than $\epsilon_{n,\delta}(D)$ on samples from distribution $D$*—even an algorithm *customized* specifically for distribution $D$—without getting unacceptably bad error on a designated nearby distribution $D'$.

For concreteness, we define $\epsilon_{n,\delta}(D)$ below.

**Definition 2.5** (Dang et al. 2023). *Given a (continuous) distribution $D$ with mean $\mu$ and a real number $t \in [0, 1]$, define the $t$-trimming operation on $D$ as follows: select a radius $r$ such that the probability mass in $[\mu - r, \mu + r]$*

*equals $1 - t$; then, return the distribution $D$ **conditioned** on lying in $[\mu - r, \mu + r]$.*

Given $n$ and $\delta$, define the trimmed distribution $D^*_{n,\delta}$ to be the $\frac{0.45}{n} \log \frac{1}{\delta}$-trimmed version of $D$. When $\delta$ is implicit, we may denote this as $D^*_n$. Now define the error function $\epsilon_{n,\delta}(D) = |\mu - \mu^*_n| + \sigma^*_n \sqrt{\frac{\log \frac{1}{\delta}}{n}}$, where $\mu^*_n$ and $\sigma^*_n$ are the mean and standard deviation of $D^*_n$ respectively.

Dang et al. show that median-of-means achieves error $O(\epsilon_{n,\delta}(D))$, and raises the question of whether more modern estimators such as Lee and Valiant's (Estimator 1) also achieve this error bound and are hence neighborhood optimal. We show a more general result: in fact, every sub-Gaussian and robust estimator (satisfying a slight variant of Theorem 2.2) achieves this error bound.

**Proposition 2.6.** *Let $\hat{\mu}$ be an arbitrary estimator that, when given $\delta > 0$ and a set of $n$ $\eta$-corrupted samples from any distribution $D$ with mean $\mu$ and variance $\sigma^2$, outputs a mean estimate satisfying*

$$|\hat{\mu} - \mu| \leq O\left( \sigma \left( \sqrt{\frac{\log \frac{1}{\delta}}{n}} + \sqrt{\eta} \right) \right)$$

*with probability at least $1 - \frac{\delta}{2}$ over the randomness of the (uncorrupted) samples.*

*Then, the same estimator $\hat{\mu}$, on input $n$ i.i.d. samples drawn from a distribution $D$ with finite mean, will output an estimate with error upper bounded by $O(\epsilon_{n,\delta}(D))$ (as defined in Definition 2.5) with probability at least $1 - \delta$.*

We formally prove Proposition 2.6 in Appendix C. The precondition of Proposition 2.6 is implied by a mild variant of Theorem 2.2 (decreasing the failure probability from $\delta$ to $\delta/2$), which holds also for Estimator 1. As a consequence:

**Corollary 2.7** (Informal). *Estimator 1 is neighborhood optimal, in the sense of Dang et al. (2023).*

In Section 5, we state the formal definition of neighborhood optimality and discuss the intuition on Proposition 2.6.

## 2.4. Asymptotic normality and efficiency

Lastly, we show that Lee and Valiant's estimator is asymptotically normal and efficient, under the standard finite variance and i.i.d. sample assumption. In particular, we show that, if the $\delta$ parameter in Estimator 1 is fixed, and the number of samples $n \rightarrow \infty$, then Estimator 1 converges in probability to the sample mean at the appropriate scale. The Central Limit Theorem for the sample mean then implies the asymptotic optimality of Lee and Valiant's estimator as a corollary.

**Theorem 2.8.** *Let $D$ be a distribution with mean $\mu$ and variance $\sigma^2$.*

*Let $\hat{\mu}$ denote Estimator 1 on input parameter $\delta$ and $n$ i.i.d. samples from $D$. Also let $\bar{X}_n$ denote the sample mean. Then, fixing $\delta$ and $D$ and taking $n \to \infty$, we have*

$$\sqrt{n}\hat{\mu} \overset{p}{\to} \sqrt{n}\bar{X}_n$$

*that is, $|\sqrt{n}\hat{\mu} - \sqrt{n}\bar{X}_n| \overset{p}{\to} 0$, that $\sqrt{n}\hat{\mu}$ converges to $\sqrt{n}\bar{X}_n$ in probability.*

*As a corollary, by the Central Limit Theorem, we have*

$$\sqrt{n}(\hat{\mu} - \mu) \overset{d}{\to} \mathcal{N}\left(0, \sigma^2\right)$$

*That is, $\hat{\mu}$ is asymptotically normal and efficient.*

The above theorem contrasts with the asymptotic behavior of median-of-means, whose error—scaled by $\sqrt{n}$—converges to $\mathcal{N}(0, (\pi/2)\sigma^2)$ (Minsker, 2023); median-of-means thus has asymptotic variance a $\pi/2$-factor larger than desired.

## 3. Related Work

**Mean estimation in 1 dimension.** Mean estimation, even in 1-dimension, has been studied algorithmically since the 1980s. The classic median-of-means estimator was the first big-O optimal sub-Gaussian mean estimator proposed in the literature, independently invented by different groups of authors (Alon et al., 1999; Nemirovsky & Yudin, 1983; Jerrum et al., 1986). Catoni's influential work (2012) gave the first sub-Gaussian mean estimator that yields the tight multiplicative constant in its error, but under strong assumptions that either the variance is known (to extremely high accuracy) or the distribution kurtosis (normalized $4^{\text{th}}$ moment) is bounded. Followup work by Devroye et al. (2016) studied "multiple-$\delta$" estimators, also with sharp error constants, in the same setting. More recently, Lee & Valiant (2022) constructed a sub-Gaussian mean estimator with tight constants, under the bare minimum assumption that the variance exists, and absent any extra knowledge or moment assumptions—this estimator is the subject of study in the current work.

See the survey of Lugosi & Mendelson (2021) on mean estimation results prior to 2019.

In low moment settings where the underlying distribution might have infinite variance, Bubeck et al. (2012) studied the performance of median-of-means. Devroye et al. (2016) then showed lower bounds that match up to constants. Our work shows that Lee and Valiant's estimator achieves analogous results as median-of-means in these regimes (Theorem 2.4), with sharper dependence on the $z^{\text{th}}$ moment, for every $z \in (1, 2]$.

"Beyond worst-case analysis" of 1-d mean estimators is a new research topic of recent interest in the community. In the standard i.i.d. setting, Dang et al. (2023) characterized the optimal distribution-specific error rates up to constants,

showing that median-of-means achieve such rates. Our work shows that in fact all estimators that simultaneously achieve (big-O) optimal sub-Gaussian and robust estimation error must also achieve the distribution-specific optimal error rates (Proposition 2.6). In addition to the standard i.i.d. setting, a different line of work has also studied distribution-specific mean estimation error rates for various differential privacy settings (Asi & Duchi, 2020a;b; Huang et al., 2021).

**Robust mean estimation.** Robust statistics, the setting where part of the input data can be corrupted by an adversary, has been an active area of research in the statistics community since the 1960s (Huber, 1992; Tukey, 1960). However, it was only in the past decade that polynomial-time algorithms for these statistical problems were found. See the textbook of Diakonikolas & Kane (2023) for a detailed introduction to these recent advances. Most directly relevant to our present work are results that give simultaneously sub-Gaussian and robust mean estimators, even in arbitrary high dimensions (Diakonikolas et al., 2020; Depersin & Lecué, 2022; Hopkins et al., 2020). Median-of-means is also known to be such a robust and sub-Gaussian estimator in 1-dimension — this is a folklore result, but see Laforgue et al. (2021) for more details on the analysis in the robust setting. Similarly, trimmed mean also has a folklore analysis for its robustness, and see Oliveira et al. (2025) for more results on the robustness and additional properties of trimmed mean.

## 4. Outlier Robustness

In this section, we outline the proof of Theorem 2.2 (restated below for the reader's convenience), which says that Estimator 1 is robust against adversarial data contamination from the strong contamination model of Definition 2.1. The proof of Theorem 2.3 has analogous structure.

**Theorem 2.2.** *Given any distribution $D$ with mean $\mu$ and variance $\sigma^2$, parameters $n, \delta, \eta > 0$, let $\tilde{X}$ be a set of $n$ $\eta$-corrupted samples from $D$.*

*Suppose both $\frac{\log \frac{1}{\delta}}{n}$ and $\delta$ are bounded by some small universal constant, and suppose $\eta \leq \frac{1}{24n}\log\frac{1}{\delta}$. Then, with probability at least $1 - \delta$ over the sampling process, Estimator 1 on input $\delta$ and $\tilde{X}$ will output an estimate $\hat{\mu}$ with error at most*

$$|\hat{\mu} - \mu| \leq \sigma \cdot \left( (\sqrt{2} + o(1))\sqrt{\frac{\log\frac{1}{\delta}}{n}} + 222\sqrt{\eta} \right)$$

The proof strategy is to bound the difference between the estimates returned by Estimator 1 on (corrupted) samples $\tilde{X}$ versus its behavior on *uncorrupted* samples $X$, fixing the confidence parameter $\delta$.

Changing the input from uncorrupted samples to corrupted samples has two effects on the resulting estimate:

1. The $\alpha$ "influence parameter" (as computed in Step 2 of Estimator 1) may change. However, we show that in a certain sense, when the fraction of corruption $\eta$ is small compared to $\log \frac{1}{\delta}/n$, this corruption will not change the computed $\alpha$ value by much (Lemma 4.2). We further show that artificially changing the $\alpha$ value by a small amount will not change the mean estimate of Step 3 by much, with high probability, when given *uncorrupted* samples.

2. For a fixed influence parameter $\alpha$, corrupting the samples from $X$ to $\tilde{X}$ changes the returned mean estimate. However, we show a (high probability) *lower bound* on the value $\alpha$ computed by the algorithm on input $\tilde{X}$ (Lemma 4.3); and this lower bound on $\alpha$ gives us a natural *upper bound* on how much any corrupted input value can affect the final mean estimate.

We state here the two key structural lemmas (and a corresponding prerequisite definition) for the $\alpha$ value computed from corrupted samples $\tilde{X}$.

**Definition 4.1.** Let $X = \{x_i\}$ be a set of clean samples, and let $\tilde{X} = \{\tilde{x}_i\}$ be the corresponding set of $\eta$-corrupted samples. Denote by $\alpha_\rho$ the "influence parameter" computed from the clean samples so as to satisfy a version of Step 2 but with a modified right hand side ($\frac{1}{3}\log \frac{1}{\delta} + \rho n$ instead of $\frac{1}{3}\log \frac{1}{\delta}$):

$$\sum_i \min(\alpha_\rho x_i^2, 1) = \frac{1}{3}\log\frac{1}{\delta} + \rho n$$

and denote by $\tilde{\alpha}_\rho$ the corresponding "influence parameter" computed instead from the *corrupted* samples:

$$\sum_i \min(\tilde{\alpha}_\rho \tilde{x}_i^2, 1) = \frac{1}{3}\log\frac{1}{\delta} + \rho n$$

**Lemma 4.2.** *Consider an arbitrary set of samples $X$ and a new sample set $\tilde{X}$ $\eta$-corrupted from $X$. Consider also an arbitrary input parameter $\delta$. Using $\tilde{\alpha}$ to denote the influence parameter of Estimator 1 on inputs $(\delta, \tilde{X})$, i.e. $\tilde{\alpha}_0$ in Definition 4.1, we have*

$$\alpha_{-2\eta} \leq \tilde{\alpha} \leq \alpha_{2\eta}$$

Considering the right hand side of the condition in Step 2 of Estimator 1 as expressing the level of "desired robustness": Lemma 4.2 states that the modified influence parameter from $\eta$-corruption is always sandwiched between the uncorrupted influence parameters, but at slightly different levels of desired robustness. We point out that Lemma 4.2 is a deterministic lemma, that always holds, regardless of the sampling over $X$.

**Lemma 4.3.** *In the setting of Lemma 4.2, suppose both $\frac{\log \frac{1}{\delta}}{n}$ and $\delta$ are bounded by some small universal constant, and suppose $\eta \leq \frac{1}{24n}\log\frac{1}{\delta}$. With probability at least $1 - 4\delta/11$ over the sampling of $n$ samples $X$ from a distribution $D$ with variance $\sigma^2$, we have $\tilde{\alpha} \geq 0.0008496\eta$.*

These lemmas let us bound $\tilde{\alpha}$ even when given corrupted data, and relate it to the uncorrupted $\alpha$; these bounds are the crucial tools needed to bound the mean estimation error in both Theorem 2.2 and Theorem 2.3. Lemma 4.3 is used in the proof of Theorem 2.2, and shown inside Proposition A.22 in Appendix A.4. The proof of Theorem 2.3 has an analogous lemma with slightly different parameters.

The full analysis of the outlier robustness of Estimator 1 is given in Appendix A.

## 5. Neighborhood Optimality

Neighborhood optimality is a new notion of fine-grained distribution-dependent optimality recently proposed by Dang et al. (2023). While sub-Gaussian bounds are worst-case optimal for the class of finite variance distributions, neighborhood optimality captures the extent to which estimators can beneficially adapt to the non-Gaussianity of the underlying distribution and outperform the sub-Gaussian bound.

Before we formally state the definition of neighborhood optimality, let us give some preliminary definitions.

Let $\mathcal{P}_1$ be the entire set of all distributions with a finite first moment over $\mathbb{R}$. We say that $N$ is a neighborhood function (defined over $\mathcal{P}_1$) if $N$ maps a distribution $D \in \mathcal{P}_1$ to a set of distributions $N(D) \subseteq \mathcal{P}_1$. Intuitively, the neighborhood $N(D)$ of $D$ is a set of distributions that we expect an estimator to perform similarly well on (and we typically consider neighborhoods where $D \in N(D)$ ). Similarly, an error function $\epsilon$ maps distributions to non-negative numbers, like the function introduced in Definition 2.5. In the later definitions, we use the notations $N_{n,\delta}$ and $\epsilon_{n,\delta}$ to indicate their dependence on the sample complexity $n$ and failure probability $\delta$.

Given these two notions, we can now define the notion of a *neighborhood Pareto bound*, as a property that an error function satisfies. Essentially, the definition imposes admissibility/Pareto efficiency structure within the local neighborhood $N_{n,\delta}(D)$ of every distribution $D \in \mathcal{P}_1$.

**Definition 5.1** (Neighborhood Pareto Bounds (Dang et al., 2023))**.** Let $n$ be the number of samples and $\delta$ be the failure probability. Given a neighborhood function $N_{n,\delta} : \mathcal{P}_1 \to 2^{\mathcal{P}_1}$, we say that the error function $\epsilon_{n,\delta}(D) : \mathcal{P}_1 \to \mathbb{R}_0^+$ is a neighborhood Pareto bound for $\mathcal{P}_1$ with respect to $N_{n,\delta}$ if for all distributions $D \in \mathcal{P}_1$, *no* estimator $\hat{\mu}$ taking $n$ i.i.d. samples can simultaneously achieve the following two

conditions:

- For all $D' \in N_{n,\delta}(D)$, with probability $1 - \delta$ over the $n$ i.i.d. samples from $D'$, we have $|\hat{\mu} - \mu_{D'}| \leq \epsilon_{n,\delta}(D')$.

- With probability $1 - \delta$ over the $n$ i.i.d. samples from $D$, $|\hat{\mu} - \mu_D| < \epsilon_{n,\delta}(D)$.

Neighborhood Pareto ounds essentially play the role of "lower bounds" in an optimality definition, and the strength of the result depends crucially on the choice of the neighborhood function $N$ under consideration. As a basic observation, the strength of this definition is *monotonic* in the size of the neighborhoods returned by $N$: if an error function $\epsilon$ is a neighborhood Pareto bound for a neighborhood function $N$, then for any neighborhood function $N'$ such that $N(D) \subseteq N'(D)$ for every $D \in \mathcal{P}_1$, $\epsilon$ is also a neighborhood Pareto bound for $N'$. Thus, the smaller each neighborhood is, the stronger the neighborhood Pareto bound is.

Finally, we define neighborhood optimality.

**Definition 5.2** (($\kappa, \tau$)-Neighborhood Optimal Estimators (Dang et al., 2023)). Let $\kappa > 1$ be a multiplicative loss factor in estimation error, and $\tau > 1$ be a multiplicative loss factor in sample complexity.
Given the parameters $\kappa, \tau > 1$, sample complexity $n$, failure probability $\delta$ and neighborhood function $N_{n,\delta}$, a mean estimator $\hat{\mu}$ is ($\kappa, \tau$)-neighborhood optimal with respect to $N_{n,\delta}$ if there exists an error function $\epsilon_{n,\delta}(D)$ such that $\min(\epsilon_{n/\tau,\delta}(D), \epsilon_{n,\delta}(D))$ is a neighborhood Pareto bound[1], and $\hat{\mu}$ gives estimation error at most $\kappa \cdot \epsilon_{n,\delta}(D)$ with probability at least $1 - \delta$ when taking $n$ i.i.d. samples from any distribution $D \in \mathcal{P}_1$.

Dang et al. (2023) showed that the error function $\epsilon_{n,\delta}$ from Definition 2.5 yields a neighborhood Pareto bound $\frac{1}{\kappa} \min(\epsilon_{n/\tau,\delta}(D), \epsilon_{n,\delta}(D))$ for an appropriate choice of neighborhood function, for some constants $\kappa, \tau > 1$. Their choice of neighborhood function $N_{n,\delta}$ is technical; we state it in Appendix C, and refer the reader to their paper for the justification. Based on this result, they also showed that median-of-means indeed achieves error $O(\epsilon_{n,\delta})$ from Definition 2.5 and hence is neighborhood optimal by Definition 5.2. Dang et al. (2023) further raised the immediate question of whether other more-modern estimators, such as Lee and Valiant's Estimator 1, can also achieve such estimation error.

We show a general affirmative answer, that in fact, all estimators that are (up to constants) simultaneously sub-Gaussian

---

[1]While it is intuitive to expect that an error function decreases in $n$, it might not be true in general. Indeed, the function used by Dang et al. (2023) (Definition 2.5) is not necessarily monotonic. This is the reason for the "min" in the neighborhood Pareto bound requirement.

and optimally robust to corruption must achieve the error rate from Definition 2.5 (stated formally as Proposition 2.6), and are thus neighborhood optimal as a corollary.

**Proposition 2.6.** *Let $\hat{\mu}$ be an arbitrary estimator that, when given $\delta > 0$ and a set of $n$ $\eta$-corrupted samples from any distribution $D$ with mean $\mu$ and variance $\sigma^2$, outputs a mean estimate satisfying*

$$|\hat{\mu} - \mu| \leq O\left(\sigma\left(\sqrt{\frac{\log \frac{1}{\delta}}{n}} + \sqrt{\eta}\right)\right)$$

*with probability at least $1 - \frac{\delta}{2}$ over the randomness of the (uncorrupted) samples.*

*Then, the same estimator $\hat{\mu}$, on input $n$ i.i.d. samples drawn from a distribution $D$ with finite mean, will output an estimate with error upper bounded by $O(\epsilon_{n,\delta}(D))$ (as defined in Definition 2.5) with probability at least $1 - \delta$.*

The precondition of Proposition 2.6 is satisfied by a variant of Theorem 2.2—with slightly smaller failure probability—which holds for Estimator 1. Thus the neighborhood optimality of Estimator 1 follows as a corollary of Proposition 2.6.

To see the intuition behind Proposition 2.6, recall the definition of the error rate function $\epsilon_{n,\delta}(D)$ from Definition 2.5. The definition constructs a distribution $D^*_{n,\delta}$ from $D$, by removing the tails of $D$ with probability mass $O(\log \frac{1}{\delta}/n)$, and the error function $\epsilon_{n,\delta}(D) = \sigma^*_n \sqrt{\frac{\log \frac{1}{\delta}}{n}} + |\mu - \mu^*_n|$ is the sub-Gaussian error for distribution $D^*_{n,\delta}$ plus the mean difference between $D$ and $D^*_{n,\delta}$. Thus, when given samples from $D$, one could view them as *corrupted* samples from $D^*_{n,\delta}$ where roughly $O(\log \frac{1}{\delta}/n)$ fraction of the samples are corrupted. A sub-Gaussian and robust estimator would thus achieve good error with respect to the mean of $D^*_{n,\delta}$, and by triangle inequality, also with respect to the mean of $D$.

We present proofs of the above statements in Appendix C. For completeness, we also provide a summary of Dang et al. (2023)'s results. We again refer the reader to their paper for a more in-depth discussion on the intricacies of the neighborhood optimality notion.

## 6. Infinite Variance Distributions

In this section, we extend Lee and Valiant's analysis of Estimator 1 to more heavy-tailed distributions. Instead of Fact 1.1, where the performance of the estimator is characterized in terms of the *variance* of the distribution $D$, we instead ask if we can characterize the performance of the estimator on distributions that may *not* have a finite variance but instead only have finite $z^{th}$ moment for some $1 < z \leq 2$.

We restate our main theorem for this section, which matches

the lower bound of Devroye et al. (2016) up to constants.

**Theorem 2.4.** *Given any distribution $D$ with mean $\mu$ and $z^{th}$ moment $M_z$ for some $z \in (1,2)$, let $X$ be a set of $n$ i.i.d. samples from $D$. Then, with probability at least $1 - \delta$ over the randomness of $X$, Estimator 1 on input $\delta$ and $X$ will output an estimate $\hat{\mu}$ with error at most*

$$|\hat{\mu} - \mu| \leq (M_z)^{\frac{1}{z}} \cdot (1 + o(1)) \left( c_z \frac{\log \frac{1}{\delta}}{n} \right)^{1 - \frac{1}{z}}$$

*where $c_z = 2(5.6)^{\frac{1}{z-1} - 1}$. Here, the $o(1)$ term tends to 0 as $\left( \frac{\log \frac{1}{\delta}}{n}, \delta \right) \to (0,0)$, in a manner independent of $D$ and independent of $z$.*

At a high level, our analysis is a generalization of Lee and Valiant's analysis to the low-moment setting, which allows us to prove a guarantee that gracefully reduces to their main result (Fact 1.1, with the sharp constant of $\sqrt{2}$) as $z \to 2$. Furthermore, our value of $c_z$ is smaller than the corresponding multiplicative constant in the analysis of median-of-means by Bubeck et al. (2012), across all values of $z \in (1, 2]$.

Here we give an overview of our analysis.

Without loss of generality, from the shift-and-scale equivariance of Estimator 1, we assume the underlying distribution has mean 0 and $z^{th}$ moment $M_z = 1$. The goal is to prove tailored Chernoff bounds for this estimator to show its concentration. Lee and Valiant's analysis in the finite variance setting provides two useful techniques to address obstacles described in the following subsections.

### 6.1. Estimator 1 is a sum of dependent terms

Estimator 1 is a sum of *dependent* terms, due to the influence parameter $\alpha$ computed in Step 2 involving all the samples. This makes proving Chernoff bounds tricky, given that moment generating functions multiply only for sums of *independent* terms. Lee and Valiant's approach is to *reduce* (via a Lipschitz argument) to analyzing the case where the preliminary estimate $\kappa$ from Step 1 is taken to be exactly equal to the true mean $\mu = 0$, and crucially reformulate Estimator 1 as a "2-parameter $\psi$-estimator".

**Definition 6.1** (Lee & Valiant 2022). Consider Estimator 1 but with Step 1 replaced with "$\kappa = 0$". The estimator can be equivalently expressed as follows:

1. Input: Failure probability $\delta$, independent samples $X = x_1, \ldots, x_n$

2. Solve for the (unique) pair $(\hat{\mu}, \hat{\alpha})$ satisfying $\psi_\mu = 0$ and $\psi_\alpha = 0$, where the functions $\psi_\mu, \psi_\alpha$ are defined

as follows:

$$\psi_\mu(X, \hat{\mu}, \hat{\alpha}) = \sum_{i=1}^n (\hat{\mu} - x_i(1 - \min(\hat{\alpha} x_i^2, 1)));$$

$$\psi_\alpha(X, \hat{\mu}, \hat{\alpha}) = \sum_{i=1}^n \left( \min(\hat{\alpha} x_i^2, 1) - \frac{1}{3n} \log \frac{1}{\delta} \right)$$

3. Output: $\hat{\mu}$ from the previous step.

This reformulation has the advantage that, for any fixed pair $(\hat{\mu}, \hat{\alpha})$, any linear combination of the $\psi_\mu$ and $\psi_\alpha$ functions is a sum of independent terms. The concentration of Estimator 1 is then reduced to proving Chernoff bounds for these linear combinations.

This reformulation and reduction is independent of the finite variance assumption, and therefore also applicable to the low moment setting that our work analyzes.

### 6.2. Proving Chernoff bounds over large distribution classes

Even in the finite variance setting, proving a Chernoff bound that applies for all distributions $D$ with mean 0 and variance 1 is daunting, given how large the distribution class is compared to standard concentration bounds. Lee and Valiant showed that the worst-case Chernoff bound (for linear combinations of the $\psi$-equations from Definition 6.1) can in fact be viewed as a max-min linear programming game.

For simplicity, let us illustrate this by sketching the analysis of the Chernoff bound of a hypothetical linear estimator that is a sum of independent terms: $f(\{x_1, \ldots, x_n\}) = \frac{1}{n} \sum_i f(x_i)$, for some fixed function $f : \mathbb{R} \to \mathbb{R}$. Proving a Chernoff bound is equivalent to upper bounding the moment generating function of the estimator $f$, and choosing the "Chernoff parameter" $t$ accordingly. Thus, it suffices to upper bound the objective of the following max-min game, where the max player chooses any mean-0 variance-1 distribution $D$—represented by variables $\{p_x\}$ denoting the probability mass at $x$ (ignoring probability formalism issues with non-discrete distributions)—and the min player chooses the Chernoff parameter $t$. Using the moment generating function as the objective function, we have

$$\begin{aligned} \max_{\{p_x\}} \quad & \min_t \sum_x p_x e^{t \cdot f(x)} \\ \text{such that} \quad & \sum_x p_x = 1 \\ & \sum_x p_x \cdot x = 0 \\ & \sum_x p_x \cdot x^2 = 1 \\ \text{where} \quad & p_x \geq 0 \end{aligned} \quad (1)$$

By using minimax duality and linear programming duality, the game can then be rewritten into a pure minimization program with the same optimum, where the dual variables

$U, M, V$ correspond to the 3 respective constraints in the program in (1).

$$\min_t \qquad \min_{U,M,V} U + V$$
$$\text{such that} \quad \text{for all } x \in \mathbb{R}, Vx^2 + Mx + U \geq e^{t \cdot f(x)} \tag{2}$$

It thus suffices to choose dual variables $U, M, V$ and an appropriate Chernoff parameter $t$ to certify an upper bound on the optimum.

We modify this approach from Lee & Valiant (2022) by relying on a $z^{\text{th}}$ moment bound instead of a variance bound. The key observation is that the $z^{\text{th}}$ moment bound may be expressed as a linear constraint in the above program, replacing $\sum_x p_x x^2 = 1$ with $\sum_x p_x |x|^z = 1$. The technical challenge from here is to provide a feasible dual solution and choose Chernoff parameter $t$ so as to satisfy the desired bounds, including that the guarantees converge to Fact 1.1 as $z \to 2$. We show our complete proof in Appendix D.

## 7. Asymptotic Normality

In this section, we show that under the standard finite variance assumption, the estimator of Lee and Valiant is asymptotically *normal* and *efficient*. Specifically, we prove that if we fix the input parameter $\delta$ and take the number of samples $n \to \infty$, the estimator converges to the sample mean *in probability*, which by the Central Limit Theorem implies asymptotic normality and efficiency.

This result contrasts with median-of-means, which, under the slightly stronger $2 + \iota$ moment assumption for any $\iota > 0$, is asymptotically normal yet *inefficient* (Minsker, 2023)—the asymptotic distribution of $\sqrt{n}\hat{\mu}_{\text{MoM}}$ is $\mathcal{N}(\mu, (\pi/2)\sigma^2)$ instead of the desired $\mathcal{N}(\mu, \sigma^2)$.

**Theorem 2.8.** *Let $D$ be a distribution with mean $\mu$ and variance $\sigma^2$.*

*Let $\hat{\mu}$ denote Estimator 1 on input parameter $\delta$ and $n$ i.i.d. samples from $D$. Also let $\bar{X}_n$ denote the sample mean. Then, fixing $\delta$ and $D$ and taking $n \to \infty$, we have*

$$\sqrt{n}\hat{\mu} \xrightarrow{p} \sqrt{n}\bar{X}_n$$

*that is, $|\sqrt{n}\hat{\mu} - \sqrt{n}\bar{X}_n| \xrightarrow{p} 0$, that $\sqrt{n}\hat{\mu}$ converges to $\sqrt{n}\bar{X}_n$ in probability.*

*As a corollary, by the Central Limit Theorem, we have*

$$\sqrt{n}(\hat{\mu} - \mu) \xrightarrow{d} \mathcal{N}\left(0, \sigma^2\right)$$

*That is, $\hat{\mu}$ is asymptotically normal and efficient.*

The proof is relatively straightforward. The key idea is that Estimator 1 differs from the sample mean by removing $\Theta(\log \frac{1}{\delta})$ weighted samples, so we might as well bound the difference by $\Theta(\log \frac{1}{\delta})$ times the maximum sample (and

symmetrically the minimum sample), multiplied by a factor of $\sqrt{n}$ since that is the scale that the Central Limit Theorem holds at. Under the finite variance assumption, we can use a (slightly refined) Chebyshev's inequality and a standard Chernoff bound to upper bound the magnitude of the maximum sample with high probability. See the complete calculations in Appendix E.

## Acknowledgments

The work of Jasper C.H. Lee was done in part while he was at UW Madison, supported by NSF Medium Award CCF-2107079. He also thanks Stanislav Minsker for discussions on the asymptotic normality result.

The work of Walter McKelvie was partly supported by the National Science Foundation Graduate Research Fellowship Program under Grant No. 2140743.

Maoyuan Song and Paul Valiant are partially supported by NSF award CCF-2127806 and by Office of Naval Research award N000142412695.

Maoyuan Song is partially supported by NSF award CCF-2228814.

## Impact Statement

This work studies a fundamental statistical problem that is broadly applicable to a wide variety of domains. As such, it does not directly raise any societal or ethical concerns that warrant special consideration.

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

# A. Remaining Proofs of Section 4

In Section 4, we discussed the intuition behind our main results, Theorem 2.2 and 2.3. At a high level, our proof strategy for both main theorems uses the triangle inequality to bound estimation error introduced by adversarial corruption as the sum of two parts, one from changing the influence parameter from $\alpha$ to $\tilde{\alpha}$, and one from the adversary arbitrarily corrupting the samples. We provide formal proofs for relevant lemmas and propositions in this section, and restate the main theorems for completeness:

**Theorem 2.2.** *Given any distribution $D$ with mean $\mu$ and variance $\sigma^2$, parameters $n, \delta, \eta > 0$, let $\tilde{X}$ be a set of $n$ $\eta$-corrupted samples from $D$.*

*Suppose both $\frac{\log \frac{1}{\delta}}{n}$ and $\delta$ are bounded by some small universal constant, and suppose $\eta \leq \frac{1}{24n} \log \frac{1}{\delta}$. Then, with probability at least $1 - \delta$ over the sampling process, Estimator 1 on input $\delta$ and $\tilde{X}$ will output an estimate $\hat{\mu}$ with error at most*

$$|\hat{\mu} - \mu| \leq \sigma \cdot \left( (\sqrt{2} + o(1)) \sqrt{\frac{\log \frac{1}{\delta}}{n}} + 222 \sqrt{\eta} \right)$$

**Theorem 2.3.** *Given any distribution $D$ with mean $\mu$ and variance $\sigma^2$, parameters $n, \delta, \eta > 0$, let $\tilde{X}$ be a set of $n$ $\eta$-corrupted samples from $D$.*

*Suppose both $\frac{\log \frac{1}{\delta}}{n}$ and $\delta$ are bounded by some small universal constant, and suppose $\eta \leq \frac{1}{9n} \log \frac{1}{\delta}$. Let $\delta'$ be so that $\frac{1}{3} \log \frac{1}{\delta} = \frac{1}{3} \log \frac{1}{\delta'} + 3\eta n$. Then, with probability at least $1 - \delta'$ over the sampling process, Estimator 1 on input $\delta$ and $\tilde{X}$ will output an estimate $\hat{\mu}$ with error at most*

$$|\hat{\mu} - \mu| \leq \sigma \cdot \left( (\sqrt{2} + o(1)) \sqrt{\frac{\log \frac{1}{\delta'}}{n}} + (135 + o(1)) \sqrt{\eta} \right)$$

Throughout the proofs, we will compare and make use of different values of "$\alpha$", computed from either corrupted or uncorrupted data, and computed from different choices of parameters in the equation defining $\alpha$. Here, we give a more general definition of notation (generalizing Definition 4.1) that we will be using.

**Definition A.1.** Let $X = \{x_i\}$ be a set of clean samples, and let $\tilde{X} = \{\tilde{x}_i\}$ be the corresponding set of $\eta$-corrupted samples. Denote by $\alpha_{\delta,\rho}$ the "influence parameter" solved from the corresponding condition involving the clean samples, so as to satisfy a version of Step 2 of Estimator 1 but with a modified right hand side ($\frac{1}{3} \log \frac{1}{\delta} + \rho n$ instead of $\frac{1}{3} \log \frac{1}{\delta}$):

$$\sum_i \min(\alpha_{\delta,\rho} x_i^2, 1) = \frac{1}{3} \log \frac{1}{\delta} + \rho n$$

and denote by $\tilde{\alpha}_{\delta,\rho}$ the "influence parameter" solved instead from the corresponding condition involving the *corrupted* samples:

$$\sum_i \min(\tilde{\alpha}_{\delta,\rho} \tilde{x}_i^2, 1) = \frac{1}{3} \log \frac{1}{\delta} + \rho n$$

Theorem 2.3 refers to failure probability $\delta'$, so parts of the analysis will involve the notations $\alpha_{\delta',\eta}$ and $\tilde{\alpha}_{\delta',\eta}$, for example.

We start with showing the crucial preliminary Lemma A.2, which bounds the value of $\tilde{\alpha}$, allowing us to approximate it using different influence parameters on the *clean* data, with no assumption of the *corrupted* data:

**Lemma A.2** (Restatement of Lemma 4.2 under the notation of Definition A.1)**.** *Consider an arbitrary set of samples $X$ and a new sample set $\tilde{X}$ $\eta$-corrupted from $X$. Consider also an arbitrary input parameter $\delta$. Using $\tilde{\alpha}$ to denote the influence parameter of Estimator 1 on inputs $(\delta, \tilde{X})$, i.e. $\tilde{\alpha}_{\delta,0}$ in Definition A.1, we have*

$$\alpha_{\delta,-2\eta} \leq \tilde{\alpha} \leq \alpha_{\delta,2\eta}$$

*Proof.* Let the clean samples be $X = \{x_i\}$ and the corrupted samples be $\tilde{X} = \{\tilde{x}_i\}$.

To prove the first inequality, suppose for the sake of contradiction that $\tilde{\alpha} < \alpha_{\delta,-2\eta}$. Then,

$$\sum_i \min(\tilde{\alpha}\tilde{x}_i^2, 1)$$

$$\leq \sum_i \min(\tilde{\alpha}x_i^2, 1) + \sum_{i\,:\,x_i \text{ corrupted}} \min(\tilde{\alpha}\tilde{x}_i^2, 1)$$

$$\leq \frac{1}{3}\log\frac{1}{\delta} - 2\eta n + \sum_{i\,:\,x_i \text{ corrupted}} \min(\tilde{\alpha}\tilde{x}_i^2, 1) \qquad \text{(since } \tilde{\alpha} < \alpha_{\delta,-2\eta})$$

$$\leq \frac{1}{3}\log\frac{1}{\delta} - \eta n \qquad \text{(since the sum has } \eta n \text{ elements, each at most 1)}$$

$$< \frac{1}{3}\log\frac{1}{\delta}$$

which is a contradiction.

The second inequality follows similarly. Suppose for the sake of contradiction that $\tilde{\alpha} > \alpha_{\delta,2\eta}$. Then,

$$\sum_i \min(\tilde{\alpha}\tilde{x}_i^2, 1)$$

$$\geq \sum_i \min(\tilde{\alpha}x_i^2, 1) - \sum_{i\,:\,x_i \text{ corrupted}} |\min(\tilde{\alpha}\tilde{x}_i^2, 1) - \min(\tilde{\alpha}x_i^2, 1)|$$

$$\geq \frac{1}{3}\log\frac{1}{\delta} + 2\eta n - \sum_{i\,:\,x_i \text{ corrupted}} |\min(\tilde{\alpha}\tilde{x}_i^2, 1) - \min(\tilde{\alpha}x_i^2, 1)| \qquad \text{(since } \tilde{\alpha} > \alpha_{\delta,2\eta})$$

$$\geq \frac{1}{3}\log\frac{1}{\delta} + \eta n \qquad \text{(since the sum has } \eta n \text{ elements, each at most 1)}$$

$$> \frac{1}{3}\log\frac{1}{\delta}$$

which is a contradiction. $\qquad\square$

We in fact generalize the above lemma slightly for use in the proof of Theorem 2.3, which can be proven from Lemma A.2 by reparameterizing $\delta$.

**Corollary A.3.** *For any set of clean samples $X$ and the corresponding $\eta$-corrupted samples $\tilde{X}$, and for any constant $c > 2$, we have $\alpha_{\delta,(c-2)\eta} \leq \tilde{\alpha}_{\delta,c\eta} \leq \alpha_{\delta,(c+2)\eta}$.*

The proof structure of Theorem 2.2 and 2.3 are essentially identical. We present the proof of Theorem 2.3 first.

For the rest of the appendix, we will assume that the uncorrupted data distribution has mean 0 and variance 1 without loss of generality, due to the shift-and-scale equivariance of Estimator 1.

## A.1. Bounding the Error due to Changing the Influence Parameter

We present the following proposition upper bounding the error incurred on Estimator 1 by using influence parameter $\tilde{\alpha} := \tilde{\alpha}_{\delta',3\eta}$ instead of $\alpha := \alpha_{\delta',0}$ on the *clean* samples. That is, we compute $\tilde{\alpha}$ on the corrupted samples, but analyze its effect on the clean samples.

For the following section, recall our assumption that the underlying distribution has mean 0 and variance 1.

**Proposition A.4.** *Suppose both $\frac{\log\frac{1}{\delta'}}{n}$ and $\delta'$ are bounded by some small universal constant. Let $\alpha$ be the influence parameter computed from the **clean** samples with robustness level $\frac{1}{3}\log\frac{1}{\delta'}$, namely $\alpha := \alpha_{\delta',0}$. Let $\tilde{\alpha}$ be the influence parameter computed from the **corrupted** samples with robustness level $\frac{1}{3}\log\frac{1}{\delta'} + 3\eta n$, namely, $\tilde{\alpha} := \tilde{\alpha}_{\delta',3\eta}$. Then with probability at least $1 - \frac{6}{8}\delta'$, the mean estimate using $\alpha$ on the clean samples differs from the mean estimate using $\tilde{\alpha}$ on the **clean** samples by at most $125.5\sqrt{\eta}$, i.e.,*

$$\left|\sum_i x_i \min(\tilde{\alpha}x_i^2, 1) - \sum_i x_i \min(\alpha x_i^2, 1)\right| \leq 125.5n\sqrt{\eta} \qquad (3)$$

To provide some intuition towards our proof strategy for Proposition A.4, first notice that we can bound the left hand side via Cauchy-Schwarz as

$$\sqrt{\left(\sum_i x_i^2\right)\left(\sum_i \left(\min(\tilde{\alpha}x_i^2, 1) - \min(\alpha x_i^2, 1)\right)^2\right)}$$

This turns out to be insufficient; we instead bound (3) by defining the set $S$ of indices where $\min(\tilde{\alpha}x_i^2, 1) \neq \min(\alpha x_i^2, 1)$, and restrict the range of both sums in (3) to the range $i \in S$, since doing so only discards zero terms and does not change the sum. Thus we instead have the Cauchy-Schwarz bound

$$\sqrt{\left(\sum_{i \in S} x_i^2\right)\left(\sum_{i \in S} \left(\min(\tilde{\alpha}x_i^2, 1) - \min(\alpha x_i^2, 1)\right)^2\right)}$$

The bound on the second parenthetical makes crucial use of the comparison of $\tilde{\alpha}$ and $\alpha$ provided by Corollary A.3. The first parenthetical is an empirical variance, but the restriction $i \in S$ means that $|x_i|$ cannot be too large; we thus use Bernstein's inequality to bound this $S$-truncated empirical second moment, in terms of a lower bound on $\alpha$, which we prove next.

To show our lower bound on $\alpha$, we first calculate the following straightforward relations between the empirical and population quantiles.

Throughout this section, we denote by $Q_q(D)$ the $q$ (true) quantile of $D$, i.e., $\mathbb{P}[D \leq Q_q(D)] = q$.

**Lemma A.5.** *Suppose both $\frac{\log \frac{1}{\delta'}}{n}$ and $\delta'$ are bounded by some small universal constant. Denote $c_1 = 0.277 < \frac{1}{3}$, and $\kappa := c_1(\frac{1}{n} \log \frac{1}{\delta'})$. Let constant $c_2 := 102.907$. Then the $1 - \kappa$ empirical quantile of $x_i$ is at most $Q_{1-\kappa/c_2}(D)$ with probability at least $1 - \frac{1}{8}\delta'$.*

*Proof.* For the $1 - \kappa$ empirical quantile of $x_i$ to be greater than $Q_{1-\kappa/c_2}(D)$, there has to be more than $\kappa n$ samples greater than $Q_{1-\kappa/c_2}(D)$. Thus, it suffices to prove that $|\{i \in [n] : x_i \geq Q_{1-\kappa/c_2}(D)\}| \geq \kappa n$ with probability at most $\frac{1}{8}\delta'$.

Denote $Z_i := \mathbf{1}_{x_i \geq Q_{1-\kappa/c_2}(D)}$. Then $Z := |\{i \in [n] : x_i \geq Q_{1-\kappa/c_2}(D)\}| = \sum_{i=1}^n Z_i$. We denote by $p = \frac{\kappa}{c_2}$ the probability that an individual $i$ is in this set; and thus $E[Z] = pn$. Since each $Z_i$ is a coin flip of probability $p$, we further have that $Var[Z_i] = p(1-p)$.

By multiplicative Chernoff,

$$\mathbb{P}[Z \geq c_2 pn] \leq \left(\frac{e^{c_2-1}}{c_2^{c_2}}\right)^{pn}$$

$$= \exp\left((c_2 - 1 - c_2 \log c_2)pn\right)$$

$$= \exp\left(\frac{c_1(c_2 - 1 - c_2 \log c_2)}{c_2}\log\frac{1}{\delta}\right)$$

$$\leq \exp\left((-1.01)\log\frac{1}{\delta'}\right) \qquad \text{by choice of } c_1 \text{ and } c_2$$

$$\leq \exp\left(-\log\frac{8}{\delta'}\right) = \frac{1}{8}\delta' \qquad \text{since } 1.01\log\frac{1}{\delta'} \geq \log\frac{8}{\delta'} \text{ for suff. small } \delta'$$

as desired. $\qquad\square$

By symmetry, we have the following corollary as well:

**Corollary A.6.** *The $\kappa$ empirical quantile of $x_i$ is at least $Q_{\kappa/c_2}(D)$ with probability at least $1 - \frac{1}{8}\delta'$.*

**Lemma A.7.** *Let $D_{trimmed}$ denote a "trimmed" version of $D$: namely, $D$ conditioned on lying in $[Q_{\kappa/c_2}(D), Q_{1-\kappa/c_2}(D)]$. Then $\mathbb{E}[D_{trimmed}^2] \leq \frac{c_2^2}{(c_2-2\kappa)^2}$.*

*Proof.* Denote by $\mathbb{1}_{untrimmed}(x)$ the indicator that returns 1 if $x \in [Q_{\kappa/c_2}(D), Q_{1-\kappa/c_2}(D)]$ and 0 otherwise. Then observe that $D_{trimmed} = \frac{c_2}{c_2 - 2\kappa}(D \cdot \mathbb{1}_{untrimmed})$. Thus

$$\mathbb{E}[D_{trimmed}^2] = \mathbb{E}\left[\left(\frac{c_2}{c_2 - 2\kappa}(D \cdot \mathbb{1}_{untrimmed})\right)^2\right]$$

$$= \frac{c_2^2}{(c_2 - 2\kappa)^2}\,\mathbb{E}[D^2\,\mathbb{1}_{untrimmed}]$$

$$\leq \frac{c_2^2}{(c_2 - 2\kappa)^2}\,\mathbb{E}[D^2]$$

$$= \frac{c_2^2}{(c_2 - 2\kappa)^2}$$

as desired. □

**Lemma A.8.** *Suppose both $\frac{\log \frac{1}{\delta'}}{n}$ and $\delta'$ are bounded by some small universal constant. Let $(x_{trimmed})_i$ denote the sample $x_i$ after trimming, namely: let $(x_{trimmed})_i = x_i$ if $x_i \in [Q_{\kappa/c_2}(D), Q_{1-\kappa/c_2}(D)]$, and $(x_{trimmed})_i = 0$ otherwise. Let constant $c_3 := 251.099$. Then $\sum_i (x_{trimmed})_i^2 \leq \frac{(c_3+1)c_2^2}{(c_2-2\kappa)^2}n$ with probability at least $1 - \frac{2}{8}\delta'$.*

*Proof.* First notice that if we replace any trimmed $x_i$ with a random sample according to $D_{trimmed}$, then the sum $\sum_i (x_{trimmed})_i^2$ can only increase. Thus, to prove the claim, it suffices to bound the sum of $n$ i.i.d. samples from $D_{trimmed}$. With an abuse of notation, we let $\{(x_{trimmed})_i\}_{i \leq n}$ denote a set of such samples.

Also notice that by our choice of $c_3$, we have, crucially, $\frac{3c_1 c_3^2}{(6+2c_3)c_2} \geq 1.01$.

We start by bounding $Q_{\kappa/c_2}(D)$ and $Q_{1-\kappa/c_2}(D)$. Since we assume $D$ has mean 0 and variance 1, by Chebyshev's inequality, $\mathbb{P}\left[|D| \geq \sqrt{\frac{c_2}{\kappa}}\right] \leq \frac{\kappa}{c_2}$, which implies that $Q_{\kappa/c_2} \geq -\sqrt{\frac{c_2}{\kappa}}$ and $Q_{1-\kappa/c_2} \leq \sqrt{\frac{c_2}{\kappa}}$. As a result, $(x_{trimmed})_i^2 \leq \frac{c_2}{\kappa}$ for all $i$.

Then,

$$\mathrm{Var}[(x_{trimmed})_i^2] \leq \mathbb{E}\left[((x_{trimmed})_i^2)^2\right]$$

$$\leq \frac{c_2}{\kappa}\,\mathbb{E}[(x_{trimmed})_i^2]$$

$$\leq \frac{c_2^3}{\kappa(c_2 - 2\kappa)^2} \qquad \text{by Lemma A.7}$$

Thus, by Bernstein's inequality,

$$\mathbb{P}\left[\sum_i ((x_{trimmed})_i)^2 \geq \frac{(c_3+1)c_2^2 n}{(c_2 - 2\kappa)^2}\right]$$

$$= \mathbb{P}\left[\frac{1}{n}\sum_i ((x_{trimmed})_i)^2 - \frac{c_2^2}{(c_2-2\kappa)^2} \geq \frac{c_3 c_2^2}{(c_2-2\kappa)^2}\right]$$

$$\leq \mathbb{P}\left[\frac{1}{n}\sum_i ((x_{trimmed})_i)^2 - \mathbb{E}[((x_{trimmed})_i)^2] \geq \frac{c_3 c_2^2}{(c_2-2\kappa)^2}\right]$$

$$\leq 2\exp\left(-\frac{c_3^2 n \frac{c_2^4}{(c_2-2\kappa)^4}}{2\,\mathrm{Var}[(x_{trimmed})_i^2] + \frac{2c_3 c_2^3}{3\kappa(c_2-2\kappa)^2}}\right)$$

$$\leq 2\exp\left(-\frac{c_3^2 n \frac{c_2^4}{(c_2-2\kappa)^4}}{\frac{2c_2^3}{\kappa(c_2-2\kappa)^2} + \frac{2c_3 c_2^3}{3\kappa(c_2-2\kappa)^2}}\right)$$

$$\begin{aligned}
&= 2\exp\left(-\frac{3c_3^2 c_2 n\kappa}{(6+2c_3)(c_2-2\kappa)^2}\right) \\
&\leq \exp\left(-\frac{3c_3^2 n\kappa}{(6+2c_3)c_2}\right) \\
&= 2\exp\left(-\frac{3c_1 c_3^2}{(6+2c_3)c_2}\log\frac{1}{\delta'}\right) \qquad \text{by definition of } \kappa \\
&\leq 2\exp\left(-1.01\log\frac{1}{\delta'}\right) \qquad \text{by choice of } c_3 \\
&\leq 2\exp\left(-\log\frac{8}{\delta'}\right) \qquad \text{since } 1.01\log\frac{1}{\delta} \geq \log\frac{8}{\delta'} \text{ for suff. small } \delta' \\
&= \frac{2}{8}\delta'
\end{aligned}$$

as desired. $\qquad\qquad\qquad\qquad\qquad\qquad\qquad\qquad\qquad\qquad\qquad\qquad\qquad\qquad\qquad\qquad$ $\square$

Combining Lemma A.5, Corollary A.6, and Lemma A.8, we have the following corollary:

**Corollary A.9.** *Suppose there is a sufficiently small constant that upper bounds* $\delta'$. *Let* $S_{<\kappa}$ *denote the set of indices* $i$ *s.t.* $x_i^2$ *is not in the top* $\kappa$ *(empirical) quantile. Then* $\sum_{i\in S_{<\kappa}} x_i^2 \leq \frac{(c_3+1)c_2^2}{(c_2-2\kappa)^2}n$ *with probability at least* $1-\frac{4}{8}\delta'$.

We are now ready to present a lower bound on $\alpha := \alpha_{\delta',0}$, before proving Proposition A.4.

**Lemma A.10.** *Suppose both* $\frac{\log\frac{1}{\delta'}}{n}$ *and* $\delta'$ *are bounded by some small universal constant. Then* $\alpha := \alpha_{\delta',0} \geq 0.000214\frac{1}{n}\log\frac{1}{\delta'}$ *with probability at least* $1-\frac{4}{8}\delta'$.

*Proof.* Recall that by definition of $\alpha$,

$$\begin{aligned}
\frac{1}{3}\log\frac{1}{\delta'} &= \sum_i \min(\alpha x_i^2, 1) \\
&= \sum_{i:x_i^2 \text{ not in the top } \kappa \text{ quantile}} \min(\alpha x_i^2, 1) + \sum_{i:x_i^2 \text{ in the top } \kappa \text{ quantile}} \min(\alpha x_i^2, 1) \\
&\leq \sum_{i:x_i^2 \text{ not in the top } \kappa \text{ quantile}} \alpha x_i^2 + \sum_{i:x_i^2 \text{ in the top } \kappa \text{ quantile}} 1 \\
&\leq \sum_{i:x_i^2 \text{ not in the top } \kappa \text{ quantile}} \alpha x_i^2 + \kappa n \\
&= \sum_{i:x_i^2 \text{ not in the top } \kappa \text{ quantile}} \alpha x_i^2 + c_1\log\frac{1}{\delta'}
\end{aligned}$$

Rearranging, this is equivalent to

$$\left(\frac{1}{3} - c_1\right)\log\frac{1}{\delta'} \leq \sum_{i:x_i^2 \text{ not in the top } \kappa \text{ quantile}} \alpha x_i^2$$

By Corollary A.9, with probability at least $1-\frac{4}{8}\delta'$,

$$\sum_{i:x_i^2 \text{ not in the top } \kappa \text{ quantile}} x_i^2 \leq \frac{(c_3+1)c_2^2}{(c_2-2\kappa)^2}n \leq \frac{(c_3+1)c_2^2}{(c_2-2)^2}n$$

Which implies that $\alpha \geq \left(\frac{1}{3}-c_1\right)\frac{(c_2-2)^2}{(c_3+1)c_2^2}\frac{1}{n}\log\frac{1}{\delta'} \geq 0.000214\frac{1}{n}\log\frac{1}{\delta'}$ with probability at least $1-\frac{4}{8}\delta'$.

$\qquad\qquad\qquad\qquad\qquad\qquad\qquad\qquad\qquad\qquad\qquad\qquad\qquad\qquad\qquad\qquad\qquad\qquad$ $\square$

*Proof of Proposition A.4.* First, notice that for all $i$ such that $|x_i| \geq \sqrt{\frac{1}{\alpha}} \geq \sqrt{\frac{1}{\tilde{\alpha}}}$, the corresponding term in the sum in the left hand side becomes 0. Thus, using the notation $\sum_{\leq}$ to denote summing over elements $|x_i| \leq \sqrt{\frac{1}{\tilde{\alpha}}}$, the left hand side in the guarantee of Proposition A.4 is equal to

$$\left| \sum_{\leq} x_i \min(\tilde{\alpha} x_i^2, 1) - \sum_{\leq} x_i \min(\alpha x_i^2, 1) \right|$$

Then, rearranging the sums, we have

$$\left| \sum_{\leq} x_i \min(\tilde{\alpha} x_i^2, 1) - \sum_{\leq} x_i \min(\alpha x_i^2, 1) \right|$$

$$\leq \sum_{\leq} \left| x_i \left( \min(\tilde{\alpha} x_i^2, 1) - \min(\alpha x_i^2, 1) \right) \right|$$

$$\leq \sqrt{\left( \sum_{\leq} x_i^2 \right) \left( \sum_{\leq} (\min(\tilde{\alpha} x_i^2, 1) - \min(\alpha x_i^2, 1))^2 \right)} \qquad \text{by Cauchy-Schwarz}$$

for which we can bound the two terms separately.

To bound the first term, since we sum over only those terms where $|x_i| \leq \sqrt{\frac{1}{\alpha}}$ for all $i$, and by Lemma A.10 $\alpha \geq \frac{0.000214}{n} \log \frac{1}{\delta'}$ with probability $1 - \frac{4}{8}\delta'$, we have that $x_i^2 \leq \frac{4672n}{\log \frac{1}{\delta'}}$ for all $i$. Since $X$ has mean 0 and variance 1, we know that $\mathbb{E}[x_i^2] \leq \mathbb{E}[X^2] = 1$, and $\text{Var}[x_i^2] \leq \mathbb{E}[x_i^4] \leq \frac{4672n}{\log \frac{1}{\delta'}} \mathbb{E}[x_i^2] \leq \frac{4672n}{\log \frac{1}{\delta'}}$. Thus, by Bernstein's inequality,

$$\mathbb{P}\left[ \sum_{\leq} x_i^2 \geq 3150n \right] = \mathbb{P}\left[ \frac{1}{n} \sum_{\leq} x_i^2 - 1 \geq 3149 \right]$$

$$\leq \mathbb{P}\left[ \frac{1}{n} \sum_{\leq} x_i^2 - \mathbb{E}[x_i^2] \geq 3149 \right]$$

$$\leq 2\exp\left( -\frac{3149^2 n}{9344 \frac{n}{\log \frac{1}{\delta'}} + 9344 \cdot 3149 \frac{n}{\log \frac{1}{\delta'}}/3} \right)$$

$$\leq 2\exp\left( -1.01\log \frac{1}{\delta'} \right)$$

$$\leq 2\exp\left( -\log \frac{8}{\delta'} \right) \qquad \text{since } 1.01\log \frac{1}{\delta'} \geq \log \frac{8}{\delta'} \text{ for suff. small } \delta'$$

$$= \frac{2}{8}\delta'$$

In other words, conditioning on Lemma A.10 holding, with probability at least $1 - \frac{2}{8}\delta'$, $\sum_{\leq} x_i^2 \leq 3150n$.

To bound the second term, note that

$$\sum_{\leq} \left( \min(\tilde{\alpha} x_i^2, 1) - \min(\alpha x_i^2, 1) \right)^2 \leq \sum_{i} \left( \min(\tilde{\alpha} x_i^2, 1) - \min(\alpha x_i^2, 1) \right)^2 \leq \sum_{i} \left( \min(\tilde{\alpha} x_i^2, 1) - \min(\alpha x_i^2, 1) \right)$$

since $\tilde{\alpha} := \tilde{\alpha}_{\delta',3\eta} \geq \alpha_{\delta',\eta} \geq \alpha := \alpha_{\delta',0}$ by Corollary A.3, and "$\alpha$" is monotonic in the $\eta$ argument, and thus $0 \leq \min(\tilde{\alpha} x_i^2, 1) - \min(\alpha x_i^2, 1) \leq 1$ for all $i$.

To further upper bound the last quantity, we have

$$\sum_{i} \left( \min(\tilde{\alpha} x_i^2, 1) - \min(\alpha x_i^2, 1) \right) = \sum_{i} \min(\tilde{\alpha} x_i^2, 1) - \sum_{i} \min(\alpha x_i^2, 1)$$

$$\leq \sum_i \min(\alpha_{\delta',5\eta} x_i^2, 1) - \sum_i \min(\alpha_{\delta',0} x_i^2, 1) \qquad \text{by Corollary A.3 and by definition of } \alpha$$

$$= \frac{1}{3}\log \frac{1}{\delta'} + 5\eta n - \frac{1}{3}\log \frac{1}{\delta'}$$

$$= 5\eta n$$

Finally, summarizing, we have that with probability at least $1 - \frac{6}{8}\delta'$:

$$\sqrt{\left(\sum_{\leq} x_i^2\right)\left(\sum_{\leq} (\min(\tilde{\alpha} x_i^2, 1) - \min(\alpha x_i^2, 1))^2\right)}$$

$$\leq \sqrt{3150n \cdot 5\eta n}$$

$$= n\sqrt{15750\eta}$$

$$\leq 125.5n\sqrt{\eta}$$

as desired. $\qquad\square$

## A.2. Bounding the Error due to Corrupting the Samples

We now present the following proposition upper bounding the error incurred on Estimator 1 by the arbitrary corruption of the adversary on the clean samples, while *fixing* $\tilde{\alpha} := \tilde{\alpha}_{\delta',3\eta}$ as the influence parameter.

We again assume that the uncorrupted distribution has mean 0 and variance 1.

**Proposition A.11.** *Suppose both $\frac{\log \frac{1}{\delta'}}{n}$ and $\delta'$ are bounded by some small universal constant. Let $\tilde{\alpha}$ be the influence parameter computed from the* **corrupted** *samples with robustness level $\frac{1}{3}\log \frac{1}{\delta'} + 3\eta n$, namely, $\tilde{\alpha} := \tilde{\alpha}_{\delta',3\eta}$. Then with probability at least $1 - \frac{1}{8}\delta'$, the mean estimate using $\tilde{\alpha}$ on the* **clean** *samples differs from the mean estimate using $\tilde{\alpha}$ on the* **corrupted** *samples by at most $8.586\sqrt{\eta}$, i.e.,*

$$\left|\sum_i \tilde{x}_i(1 - \min(\tilde{\alpha}\tilde{x}_i^2, 1)) - \sum_i x_i(1 - \min(\tilde{\alpha} x_i^2, 1))\right| \leq 8.586n\sqrt{\eta}$$

To provide some intuition towards our proof strategy for Proposition A.11, consider the adversary's arbitrary corruption, which can be interpreted piece-wise as moving each clean sample that the adversary wishes to corrupt to a new location. Since the influence parameter controls the contribution of each sample to the mean estimate, based on how far from the mean it is, moving a sample too far from the mean or moving the sample too close to the mean will both incur very little error. The question then, is, what is the maximum estimation error the adversary can incur by corrupting a single sample?

Fixing the value of $\tilde{\alpha}$ as in the statement of Proposition A.11, we can upper bound the maximum magnitude of the expression $\frac{1}{n}\sum_i x_i(1 - \min(\tilde{\alpha} x_i^2, 1))$ by calculus, which is $O(1/\sqrt{\tilde{\alpha}})$. We will show a lower bound of $\tilde{\alpha}$, specifically that $\tilde{\alpha} \geq \Omega(\eta)$, and then conclude that the maximum total error possible by corrupting $\eta n$ samples is at most $\frac{1}{n} \cdot \eta n \cdot \Theta(1/\sqrt{\eta}) = \Theta(\sqrt{\eta})$, as desired.

We use a similar strategy as in the proof of Proposition A.4, using quantile statistics as well as Corollary A.3 to obtain our desired lower bound on $\tilde{\alpha}$, before proving Proposition A.11 in Appendix A.

**Lemma A.12.** *Suppose both $\frac{\log \frac{1}{\delta'}}{n}$ and $\delta'$ are bounded by some small universal constant. Denote $c_1 = \frac{1}{3}$, and $\kappa := c_1(\frac{1}{n}\log \frac{1}{\delta'})$. Let $c_2 := 55.252$. Then the $1 - \kappa$ empirical quantile of $x_i$ is at most $Q_{1-\kappa/c_2}(D)$ with probability at least $1 - \frac{1}{32}\delta'$.*

*Proof.* For the $1 - \kappa$ empirical quantile of $x_i$ to be greater than $Q_{1-\kappa/c_2}(D)$, there has to be more than $\kappa n$ samples greater than $Q_{1-\kappa/c_2}(D)$. Thus, it suffices to prove that $|\{i \in [n] : x_i \geq Q_{1-\kappa/c_2}(D)\}| \geq \kappa n$ with probability at most $\frac{1}{32}\delta'$.

Denote $Z_i := \mathbf{1}_{x_i \geq Q_{1-\kappa/c_2}(D)}$. Then $Z := |\{i \in [n] : x_i \geq Q_{1-\kappa/c_2}(D)\}| = \sum_{i=1}^n Z_i$. Obviously $E[Z] = \kappa n/c_2$. Denote $p := E[Z_i] = \kappa/c_2$. Then $Var[Z_i] = p(1-p)$.

By multiplicative Chernoff,

$$
\begin{aligned}
\mathbb{P}[Z \geq c_2 pn] &\leq \left(\frac{e^{c_2-1}}{c_2^{c_2}}\right)^{pn} \\
&= \exp\left((c_2 - 1 - c_2 \log c_2)pn\right) \\
&= \exp\left(\frac{c_1(c_2 - 1 - c_2 \log c_2)}{c_2}(\log\frac{1}{\delta'})\right) \\
&\leq \exp\left((-1.01)(\log\frac{1}{\delta'})\right) \qquad \text{by choice of } c_1, c_2 \\
&\leq \exp\left(-\log\frac{32}{\delta'}\right) = \frac{1}{32}\delta' \qquad \text{since } 1.01\log\frac{1}{\delta'} \geq \log\frac{32}{\delta'} \text{ for suff. small } \delta'
\end{aligned}
$$

as desired. $\qquad\square$

By symmetry, we have the following corollary as well:

**Corollary A.13.** *The $\kappa$ empirical quantile of $x_i$ is at least $Q_{\kappa/c_2}(D)$ with probability at least $1 - \frac{1}{32}\delta'$.*

**Lemma A.14.** *Let $D_{trimmed}$ denote the trimmed version of $D$ conditioned on lying in $[Q_{\kappa/c_2}(D), Q_{1-\kappa/c_2}(D)]$. Then $\mathbb{E}[D_{trimmed}^2] \leq \frac{c_2^2}{(c_2-2\kappa)^2}$.*

*Proof.* Denote by $\mathbb{1}_{untrimmed}(x)$ the indicator measure that maps $x$ to $\mathbb{1}_{Q_{\kappa/c_2}(D)\leq x \leq Q_{1-\kappa/c_2}(D)}$, the indicator of whether $x$ is untrimmed. Then observe that $D_{trimmed} = \frac{c_2}{c_2-2\kappa}(D \cdot \mathbb{1}_{untrimmed})$. Thus

$$
\begin{aligned}
\mathbb{E}[D_{trimmed}^2] &= \mathbb{E}\left[\left(\frac{c_2}{c_2-2\kappa}(D \cdot \mathbb{1}_{untrimmed})\right)^2\right] \\
&= \frac{c_2^2}{(c_2-2\kappa)^2}\,\mathbb{E}[D^2 \mathbb{1}_{untrimmed}] \\
&\leq \frac{c_2^2}{(c_2-2\kappa)^2}\,\mathbb{E}[D^2] \\
&= \frac{c_2^2}{(c_2-2\kappa)^2}
\end{aligned}
$$

as desired. $\qquad\square$

**Lemma A.15.** *Suppose both $\frac{\log\frac{1}{\delta'}}{n}$ and $\delta'$ are bounded by some small universal constant. Let $(x_{trimmed})_i$ denote the sample $x_i$ after trimming, such that $(x_{trimmed})_i = x_i$ if $x_i \in [Q_{\kappa/c_2}(D), Q_{1-\kappa/c_2}(D)]$, and $(x_{trimmed})_i = 0$ otherwise. Denote $c_3 := 114.532$. Then $\sum_i (x_{trimmed})_i^2 \leq \frac{(c_3+1)c_2^2}{(c_2-2\kappa)^2}n$ with probability at least $1 - \frac{2}{32}\delta'$.*

*Proof.* First notice that if we replace any trimmed $x_i$ with a random sample according to $D_{trimmed}$, then the sum $\sum_i (x_{trimmed})_i^2$ can only increase. Thus, to prove the claim, it suffices to bound the sum of $n$ i.i.d. samples from $D_{trimmed}$. With an abuse of notation, we let $\{(x_{trimmed})_i\}_{i\leq n}$ denote a set of such samples.

Also notice that by our choice of $c_3$, we have, crucially, $\frac{3c_3^2 c_1}{(6+2c_3)c_2} \geq 1.01$.

We start by bounding $Q_{\kappa/c_2}(D)$ and $Q_{1-\kappa/c_2}(D)$. Since we assume $D$ has mean $0$ and variance $1$, by Chebyshev's inequality, $\mathbb{P}\left[|D| \geq \sqrt{\frac{c_2}{\kappa}}\right] \leq \frac{\kappa}{c_2}$, which implies $Q_{\kappa/c_2} \geq -\sqrt{\frac{c_2}{\kappa}}$ and $Q_{1-\kappa/c_2} \leq \sqrt{\frac{c_2}{\kappa}}$. As a result, $(x_{trimmed})_i^2 \leq \frac{c_2}{\kappa}$ for all $i$.

Then,

$$
\begin{aligned}
\mathrm{Var}[(x_{trimmed})_i^2] &\leq \mathbb{E}\left[((x_{trimmed})_i^2)^2\right] \\
&\leq \frac{c_2}{\kappa}\,\mathbb{E}[(x_{trimmed})_i^2]
\end{aligned}
$$

$$\leq \frac{c_2^3}{\kappa(c_2 - 2\kappa)^2} \qquad \text{by Lemma A.14}$$

Thus, by Bernstein's inequality,

$$\mathbb{P}\left[\sum_i ((x_{trimmed})_i)^2 \geq \frac{(c_3 + 1)c_2^2 n}{(c_2 - 2\kappa)^2}\right]$$

$$= \mathbb{P}\left[\frac{1}{n}\sum_i ((x_{trimmed})_i)^2 - \frac{c_2^2}{(c_2 - 2\kappa)^2} \geq \frac{c_3 c_2^2}{(c_2 - 2\kappa)^2}\right]$$

$$\leq \mathbb{P}\left[\frac{1}{n}\sum_i ((x_{trimmed})_i)^2 - \mathbb{E}[((x_{trimmed})_i)^2] \geq \frac{c_3 c_2^2}{(c_2 - 2\kappa)^2}\right]$$

$$\leq 2\exp\left(-\frac{c_3^2 n \frac{c_2^4}{(c_2-2\kappa)^4}}{2\operatorname{Var}[(x_{trimmed})_i^2] + \frac{2c_3 c_2^3}{3\kappa(c_2-2\kappa)^2}}\right)$$

$$\leq 2\exp\left(-\frac{c_3^2 n \frac{c_2^4}{(c_2-2\kappa)^4}}{\frac{2c_2^3}{\kappa(c_2-2\kappa)^2} + \frac{2c_3 c_2^3}{3\kappa(c_2-2\kappa)^2}}\right)$$

$$= 2\exp\left(-\frac{3c_3^2 c_2 n\kappa}{(6 + 2c_3)(c_2 - 2\kappa)^2}\right)$$

$$\leq 2\exp\left(-\frac{3c_3^2 n\kappa}{(6 + 2c_3)c_2}\right)$$

$$= 2\exp\left(-\frac{3c_3^2 c_1}{(6 + 2c_3)c_2}(\log \frac{1}{\delta'})\right) \qquad \text{by definition of } \kappa$$

$$\leq 2\exp\left(-1.01(\log \frac{1}{\delta'})\right) \qquad \text{by choice of } c_3$$

$$\leq 2\exp\left(-(\log \frac{32}{\delta'})\right) \qquad \text{since } 1.01\log\frac{1}{\delta'} \geq \log\frac{32}{\delta'} \text{ for suff. small } \delta'$$

$$= \frac{2}{32}\delta'$$

as desired.

$\square$

Combining Lemma A.12, Corollary A.13, and Lemma A.15, we have the following corollary:

**Corollary A.16.** *Suppose both $\frac{\log \frac{1}{\delta'}}{n}$ and $\delta'$ are bounded by some small universal constant. Let $S_{<\kappa}$ denote the set of indices $i$ s.t. $x_i^2$ is not in the top $\kappa$ (empirical) quantile. Then $\sum_{i \in S_{<\kappa}} x_i^2 \leq \frac{(c_3+1)c_2^2}{(c_2-2\kappa)^2}n$ with probability at least $1 - \frac{1}{8}\delta'$.*

*Proof of Proposition A.11.* We start by consider a single sample $\tilde{x}_i$ which is corrupted such that $x_i \neq \tilde{x}_i$. Fixing all other samples, we solve for the maximum (signed) error, or equivalently mean shift, that the adversary can incur by arbitrarily corrupting this sample $x_i$ only.

Recall that the estimator (fixing influence parameter $\tilde{\alpha} := \tilde{\alpha}_{\delta',3\eta}$) is defined as $\hat{\mu} = \sum_i x_i(1 - \min(\tilde{\alpha}x_i^2, 1))$. Taking the derivative with respect to $x_i$, restricted to the region in which $\min(\tilde{\alpha}x_i^2, 1) = \tilde{\alpha}x_i^2$, we have

$$\frac{\partial}{\partial x_i}\hat{\mu} = 1 - 3\tilde{\alpha}x_i^2 = 0 \Rightarrow x = \pm\frac{1}{\sqrt{3\tilde{\alpha}}}$$

Since $\tilde{\alpha} \geq 0$, a local maximum occurs at $x = \frac{1}{\sqrt{3\tilde{\alpha}}}$. The corresponding maximum contribution of this single term is $\frac{2}{3\sqrt{3\tilde{\alpha}}}$. It is easy to verify that this local maximum is in fact the global maximum based on taking the minimization between $\tilde{\alpha}x_i^2$ and 1. Symmetrically, the minimal contribution of a single term is $-\frac{2}{3\sqrt{3\tilde{\alpha}}}$.

Thus, the maximum error the adversary can incur by corrupting a single term is $\frac{4}{3\sqrt{3\tilde{\alpha}}}$. The maximum error the adversary can incur by corrupting $\eta n$ terms is $\frac{4\eta n}{3\sqrt{3\tilde{\alpha}}}$.

To upper bound $\frac{4\eta n}{3\sqrt{3\tilde{\alpha}}}$, we need to lower bound $\tilde{\alpha}$. Towards this, notice that $\tilde{\alpha} \geq \alpha_{\delta',\eta}$ by Corollary A.3, which by definition satisfies:

$$
\begin{aligned}
\eta n + \frac{1}{3}\log\frac{1}{\delta'} &= \sum_i \min(\alpha_{\delta',\eta}x_i^2, 1) \\
&\leq \sum_i \min(\tilde{\alpha}x_i^2, 1) \qquad \text{by Corollary A.3} \\
&\leq \sum_{i:x_i^2 \text{ not in the top } \kappa \text{ quantile}} \min(\tilde{\alpha}x_i^2, 1) + \sum_{i:x_i^2 \text{ in the top } \kappa \text{ quantile}} \min(\tilde{\alpha}x_i^2, 1) \\
&\leq \sum_{i:x_i^2 \text{ not in the top } \kappa \text{ quantile}} \tilde{\alpha}x_i^2 + \sum_{i:x_i^2 \text{ in the top } \kappa \text{ quantile}} 1 \\
&= \sum_{i:x_i^2 \text{ not in the top } \kappa \text{ quantile}} \tilde{\alpha}x_i^2 + \kappa n \\
&= \sum_{i:x_i^2 \text{ not in the top } \kappa \text{ quantile}} \tilde{\alpha}x_i^2 + \frac{1}{3}\log\frac{1}{\delta'}
\end{aligned}
$$

Rearranging, this is equivalent to

$$
\eta n \leq \sum_{i:x_i^2 \text{ not in the top } \kappa \text{ quantile}} \tilde{\alpha}x_i^2
$$

By Corollary A.16, with probability at least $1 - \frac{1}{8}\delta'$,

$$
\begin{aligned}
&\sum_{i:x_i^2 \text{ not in the top } \kappa \text{ quantile}} x_i^2 \\
&\leq \frac{(c_3+1)c_2^2}{(c_2-2\kappa)^2}n \\
&\leq \frac{(c_3+1)c_2^2}{(c_2-2)^2}n
\end{aligned}
$$

Thus,

$$
\eta n \leq \sum_{i:x_i^2 \text{ not in the top } \kappa \text{ quantile}} \tilde{\alpha}x_i^2 \leq \frac{(c_3+1)c_2^2}{(c_2-2)^2}n\tilde{\alpha}
$$

which is equivalent to

$$
\tilde{\alpha} \geq \frac{(c_2-2)^2}{(c_3+1)c_2^2}\eta \geq 0.00804\eta
$$

Thus, we have

$$
\left| \sum_i \tilde{x}_i(1 - \min(\tilde{\alpha}\tilde{x}_i^2, 1)) - \sum_i x_i(1 - \min(\tilde{\alpha}x_i^2, 1)) \right| \leq \frac{4\eta n}{3\sqrt{3\tilde{\alpha}}} \leq \frac{4\eta n}{3\sqrt{0.02412\eta}} \leq 8.586n\sqrt{\eta}
$$

as desired. $\qquad\square$

### A.3. Proof of Theorem 2.3

Equipped with Propositions A.4 and A.11, we are ready to formally prove Theorem 2.3.

*Proof of Theorem 2.3.* We will start by assuming that the underlying distribution has mean 0 and variance 1 (by the shift-and-scale equivariance of Estimator 1), and furthermore, that the initial estimate $\kappa$ computed in Step 1 is exactly the true mean of 0. At the end of this proof, we will show that the $\kappa = 0$ assumption introduces only a negligible amount of mean estimation error.

Fixing the initial estimate $\kappa = 0$, then, Estimator 1 on input the clean samples $X$ and confidence parameter $\delta'$ computes the influence parameter $\alpha := \alpha_{\delta',0}$, and returns an estimate $\hat{\mu}_{clean,\alpha} := \frac{1}{n}\sum_i x_i(1 - \min(\alpha x_i^2, 1))$. As a part of their analysis, Lee & Valiant (2022) showed that

$$|\hat{\mu}_{clean,\alpha}| \leq (\sqrt{2} + o(1))\sqrt{\frac{\log\frac{1}{\delta'}}{n}}$$

with probability at least $1 - \delta'$. We note that the above guarantee can in fact be slightly strengthened, so that the failure probability becomes $\delta'/16$, at the expense of a slightly increased $o(1)$ by no more than a constant multiplicative factor.

Now we consider changing the "$\alpha$" value from $\alpha$ to $\tilde{\alpha}$ computed from the corrupted samples, but applying these $\alpha$ influence parameters on the same clean sample set, and bound the difference in the resulting mean estimates. Specifically, consider $\tilde{\alpha} := \tilde{\alpha}_{\delta',3\eta}$, which is the influence parameter computed with robustness level $\frac{1}{3}\log\frac{1}{\delta'} + 3\eta n$ from the *corrupted* samples. Let the mean estimate of using $\tilde{\alpha}$ on the **clean** samples be $\hat{\mu}_{clean,\tilde{\alpha}} := \frac{1}{n}\sum_i x_i(1 - \min(\tilde{\alpha}x_i^2, 1))$. Then, by Proposition A.4, with probability at least $1 - \frac{6}{8}\delta'$ we have

$$|\hat{\mu}_{clean,\tilde{\alpha}} - \hat{\mu}_{clean,\alpha}| = \frac{1}{n}\left|\sum_i x_i\min(\tilde{\alpha}x_i^2, 1) - \sum_i x_i\min(\alpha x_i^2, 1)\right| \leq 125.5\sqrt{\eta}$$

Next we consider the effect of replacing the effect of corruption, but after fixing the influence parameter to be $\tilde{\alpha}$. Specifically, define $\hat{\mu}_{corrupt,\tilde{\alpha}}$ to be the mean estimate of the corrupted samples using $\tilde{\alpha}$, namely $\frac{1}{n}\sum_i \tilde{x}_i(1 - \min(\tilde{\alpha}\tilde{x}_i^2, 1))$. Then, by Proposition A.11, with probability at least $1 - \frac{1}{8}\delta'$, we have

$$|\hat{\mu}_{corrupt,\tilde{\alpha}} - \hat{\mu}_{clean,\tilde{\alpha}}| = \frac{1}{n}\left|\sum_i \tilde{x}_i(1 - \min(\tilde{\alpha}\tilde{x}_i^2, 1)) - \sum_i x_i(1 - \min(\tilde{\alpha}x_i^2, 1))\right| \leq 8.586\sqrt{\eta}$$

By union bound and triangle inequality, summing over all three error terms, we have that with probability at least $1 - \frac{15}{16}\delta'$, $\hat{\mu}_{corrupt,\tilde{\alpha}}$ satisfies

$$|\hat{\mu}_{corrupt,\tilde{\alpha}}| \leq (\sqrt{2} + o(1))\sqrt{\frac{\log\frac{1}{\delta'}}{n}} + 134.086\sqrt{\eta} \leq (\sqrt{2} + o(1))\sqrt{\frac{\log\frac{1}{\delta'}}{n}} + 135\sqrt{\eta}$$

Finally, observe that on a mean-0-variance-1 distribution, and with $\eta$-corruption, the only difference between $\hat{\mu}_{corrupt,\tilde{\alpha}}$ and the output $\hat{\mu}$ of Estimator 1 (on input as stated in the theorem statement) lies in $\hat{\mu}_{corrupt,\tilde{\alpha}}$ assuming that the initial mean estimate $\kappa$ was the true mean of 0. On the other hand, $\hat{\mu}$ (Estimator 1) uses an estimator (for example, median-of-means) to compute $\kappa$.

We use the following fact shown in Lee & Valiant (2022) about the structural properties of Estimator 1, to analyze the effect of the $\kappa$ assumption:

**Fact A.17** (Lee & Valiant 2022). *Let $X$ be a fixed set of samples of size $n$, and let $\delta > 0$ be a confidence parameter. Let $\hat{\mu} = \hat{\mu}(X, \delta)$ denote the output of Estimator 1 of LV22. Then Estimator 1 is affine invariant, i.e.:*

$$\hat{\mu}(aX + b, \delta) = a\hat{\mu}(X, \delta) + b$$

*Additionally, let $\hat{\mu}_\kappa(X, \delta)$ denote the output of Estimator 1 but where Step 1 is omitted, and the initial estimate $\kappa$ is instead considered as an input. Then:*

$$\left|\frac{\partial\hat{\mu}_\kappa(X, \delta)}{\partial\kappa}\right| = O\left(\sqrt{\frac{\log\frac{1}{\delta}}{n}}\right)$$

We also use the following fact/assumption on the robustness of the initial estimate $\kappa$ against adversarial corruption: it is a folklore result that median-of-means satisfies this. A proof of Fact A.18 is given in Appendix F for completeness, since we are unaware of literature explicitly writing down this proof. As discussed in Estimator 1, we are free to use other estimators to compute $\kappa$ as long as Fact A.18 holds analogously.

**Fact A.18** (Folklore). *For any distribution $D$ with mean $\mu$ and standard deviation $\sigma$, let $\tilde{X}$ be a set of $n$ $\eta$-corrupted samples from $D$. The median-of-means estimate $\kappa$ from grouping samples into $O(\log \frac{1}{\delta'} + \eta n)$ buckets, on input $\tilde{X}$, satisfies*

$$\mathbb{P}\left(|\kappa - \mu| \geq O\left(\sigma\sqrt{\frac{\log \frac{1}{\delta'}}{n}} + \eta\right)\right) \leq \frac{1}{16}\delta'$$

By Fact A.18, with probability at least $1 - \frac{1}{16}\delta'$, $|\kappa| \leq O\left(\sqrt{\frac{\log \frac{1}{\delta'}}{n}} + \eta\right)$, and by the Lipschitz bound of Fact A.17, we can bound the (absolute) difference between $\hat{\mu}$ and $\hat{\mu}_{corrupt,\tilde{\alpha}}$ by

$$|\hat{\mu} - \hat{\mu}_{corrupt,\tilde{\alpha}}| \leq O\left(|\kappa|\sqrt{\frac{\log \frac{1}{\delta'}}{n}}\right) \leq O\left(\sqrt{\left(\frac{\log \frac{1}{\delta'}}{n}\right)^2 + \eta\frac{\log \frac{1}{\delta'}}{n}}\right) \leq o\left(\sqrt{\frac{\log \frac{1}{\delta'}}{n}}\right) + o(\sqrt{\eta})$$

Thus, on a mean-0-variance 1 distribution, with probability at least $1 - \delta'$ over the clean samples (and with an arbitrary $\eta$-corruption), Estimator 1 outputs

$$|\hat{\mu}| \leq (\sqrt{2} + o(1))\sqrt{\frac{\log \frac{1}{\delta'}}{n}} + (135 + o(1))\sqrt{\eta}$$

Combined with the affine invariance of Estimator 1 as stated in Fact A.17, if the underlying distribution instead had mean $\mu$ and variance $\sigma^2$, we have the mean estimation guarantee

$$|\hat{\mu} - \mu| \leq \sigma \cdot \left((\sqrt{2} + o(1))\sqrt{\frac{\log \frac{1}{\delta'}}{n}} + (135 + o(1))\sqrt{\eta}\right)$$

as desired. $\qquad\square$

We note that the $o(1)\sqrt{\eta}$ term in the estimation error is solely incurred by the estimation error from the initial estimate $\kappa$.

## A.4. Proof of Theorem 2.2

In this section, we present the proof of Theorem 2.2. While the high level idea of our proof is similar to that of Theorem 2.3, there are some important distinctions, most importantly in what we require the failure probability of our component lemmas to be.

Recall that in Theorem 2.3, while we gave the parameter $\delta$ to Estimator 1, we relaxed the failure probability of the estimator to $\delta' \geq \delta$. On the other hand, Theorem 2.2 actually analyzes Estimator 1 at failure probability $\delta$, which in turn requires the prerequisite lemmas to have failure probability $\leq \delta$.

Towards that end, notice that given the desired failure probability $\delta$, if we let $\delta'$ be such that $\frac{1}{3}\log \frac{1}{\delta} = \frac{1}{3}\log \frac{1}{\delta'} + 8\eta n$, then $\delta = \delta'e^{-24\eta n}$. Thus, the goal of this section is to devise analogs of Proposition A.4 and A.11 with $1 - \delta = 1 - \delta'e^{-24\eta n}$ as failure probability instead. We stress that this reparameterization is purely for analytical purposes; Neither the estimator nor the user know anything about the corruption parameter $\eta$ or the analytical assumption we use to define $\delta'$. Our choice of the constant 8 in front of the $\eta n$ term is different from the previous section, and stems from a somewhat arbitrary numerical choice to prevent constants in the theorem from blowing up, while also implicitly posing constraints on $\eta$.

We present the counterpart of Proposition A.4 first, bounding the error coming from the change in $\alpha$ value on uncorrupted samples, with a smaller failure probability.

**Proposition A.19.** *Suppose both $\frac{\log \frac{1}{\delta}}{n}$ and $\delta$ are bounded by some small universal constant. Let $\alpha$ be the influence parameter computed from the **clean** samples with robustness level $\frac{1}{3}\log \frac{1}{\delta'} + 8\eta n$, namely, $\alpha := \alpha_{\delta',8\eta}$. Let $\tilde{\alpha}$ be the influence parameter computed from the **corrupted** samples with robustness level $\frac{1}{3}\log \frac{1}{\delta'} + 8\eta n$, namely, $\tilde{\alpha} := \tilde{\alpha}_{\delta',8\eta}$. Then with probability at least $1 - \frac{10}{11}\delta' e^{-24\eta n}$, the mean estimate using $\alpha$ on the clean samples differs from the mean estimate using $\tilde{\alpha}$ on the **clean** samples by at most $195.065\sqrt{\eta}$, i.e.,*

$$\left| \sum_i x_i \min(\tilde{\alpha}x_i^2, 1) - \sum_i x_i \min(\alpha x_i^2, 1) \right| \leq 195.065n\sqrt{\eta}$$

We begin proving Proposition A.19 by stating an alternative version of Lemma A.10:

**Lemma A.20.** *Suppose both $\frac{\log \frac{1}{\delta}}{n}$ and $\delta$ are bounded by some small universal constant. Then $\alpha := \alpha_{\delta',8\eta} \geq 0.000214\frac{1}{n}(\log \frac{1}{\delta'} + 24\eta n)$ with probability at least $1 - \frac{4}{11}\delta' \cdot e^{-24\eta n}$.*

The proof is identical to that of Lemma A.10, with the appropriate $\log \frac{1}{\delta}$ and $\delta$ terms replaced with $\log \frac{1}{\delta'} + 24\eta n$ and $\delta' \cdot e^{-24\eta n}$ instead.

Now that we have lower bounded $\alpha$ (with high probability), analogously to the proof of Proposition A.4, we now wish to also lower bound $\tilde{\alpha}$. The proof of this lower bound will deviate from that in Proposition A.4—in Proposition A.4 we were analyzing and comparing $\tilde{\alpha} := \tilde{\alpha}_{\delta',3\eta}$ and $\alpha := \alpha_{\delta',0}$; here in Proposition A.19, we are comparing $\tilde{\alpha} := \tilde{\alpha}_{\delta',8\eta}$ against $\alpha := \alpha_{\delta',8\eta}$.

For Proposition A.4, the "$\eta$-subscripts" between the two "$\alpha$" values differ by $3\eta$, so we could apply Corollary A.3 and the monotonicity of $\alpha_{\delta',\eta}$ in the "$\eta$-subscript" to yield $\tilde{\alpha} := \tilde{\alpha}_{\delta',3\eta} \geq \alpha_{\delta',\eta} \geq \alpha := \alpha_{\delta',0}$. Here, we need a new argument to lower bound $\tilde{\alpha} := \tilde{\alpha}_{\delta',8\eta}$, shown in the following lemma.

**Lemma A.21.** *Suppose both $\frac{\log \frac{1}{\delta}}{n}$ and $\delta$ are bounded by some small universal constant. Then $\tilde{\alpha} := \tilde{\alpha}_{\delta',8\eta} \geq 0.0000354\frac{1}{n}(\log \frac{1}{\delta'} + 24\eta n)$ with probability at least $1 - \frac{4}{11}\delta' \cdot e^{-24\eta n}$.*

*Proof.* Notice that $\tilde{\alpha} := \tilde{\alpha}_{\delta',8\eta} \geq \alpha_{\delta',6\eta}$ by Corollary A.3. Letting $\kappa := c_1 \frac{1}{n}(\log \frac{1}{\delta'} + 24\eta n)$ for some constant $c_1 < \frac{1}{4}$, we have

$$\begin{aligned}
\frac{1}{4}(24\eta n + \log \frac{1}{\delta'}) &= 6\eta n + \frac{1}{4}\log \frac{1}{\delta'} \\
&\leq 6\eta n + \frac{1}{3}\log \frac{1}{\delta'} \\
&= \sum_i \min(\alpha_{\delta',6\eta}x_i^2, 1) \\
&\leq \sum_i \min(\tilde{\alpha}x_i^2, 1) \qquad \text{by Corollary A.3 as above} \\
&\leq \sum_{i:x_i^2 \text{ not in the top } \kappa \text{ quantile}} \min(\tilde{\alpha}x_i^2, 1) + \sum_{i:x_i^2 \text{ in the top } \kappa \text{ quantile}} \min(\tilde{\alpha}x_i^2, 1) \\
&\leq \sum_{i:x_i^2 \text{ not in the top } \kappa \text{ quantile}} \tilde{\alpha}x_i^2 + \sum_{i:x_i^2 \text{ in the top } \kappa \text{ quantile}} 1 \\
&= \sum_{i:x_i^2 \text{ not in the top } \kappa \text{ quantile}} \tilde{\alpha}x_i^2 + \kappa n \\
&= \sum_{i:x_i^2 \text{ not in the top } \kappa \text{ quantile}} \tilde{\alpha}x_i^2 + c_1(24\eta n + \log \frac{1}{\delta'})
\end{aligned}$$

This implies that

$$\sum_{i:x_i^2 \text{ not in the top } \kappa \text{ quantile}} \tilde{\alpha}x_i^2 \geq (\frac{1}{4} - c_1)(\log \frac{1}{\delta'} + 24\eta n)$$

With a slightly modified version of Corollary A.16, bounding $\sum_{i:x_i^2 \text{ not in the top } \kappa \text{ quantile}} x_i^2$ by $\frac{(c_3+1)c_2^2}{(c_2-2)^2}n$ with probability at least $1 - \frac{4}{11}\delta' \cdot e^{-24\eta n}$, we can choose $c_1 = 0.202, c_2 = 398.432, c_3 = 1328.46$, and obtain $\tilde{\alpha} \geq 0.0000354(\log \frac{1}{\delta'} + 24\eta n)$ as desired. $\qquad\square$

*Proof of Proposition A.19.* First, notice that for all $i$ such that $|x_i| \geq \max(\sqrt{\frac{1}{\alpha}}, \sqrt{\frac{1}{\tilde{\alpha}}}) = \sqrt{\frac{1}{\min(\alpha,\tilde{\alpha})}}$, the corresponding term in the left hand side becomes 0. Thus, using the notation $\sum_{\leq}$ to denote summing over elements $|x_i| \leq \sqrt{\frac{1}{\min(\alpha,\tilde{\alpha})}}$, the left hand side in the guarantee of Proposition A.19 is equal to

$$\left| \sum_{\leq} x_i \min(\tilde{\alpha}x_i^2, 1) - \sum_{\leq} x_i \min(\alpha x_i^2, 1) \right|$$

Then, rearranging the sums, we have

$$\left| \sum_{\leq} x_i \min(\tilde{\alpha}x_i^2, 1) - \sum_{\leq} x_i \min(\alpha x_i^2, 1) \right|$$
$$= \sum_{\leq} \left| x_i \left( \min(\tilde{\alpha}x_i^2, 1) - \min(\alpha x_i^2, 1) \right) \right|$$
$$\leq \sqrt{\left( \sum_{\leq} x_i^2 \right) \left( \sum_{\leq} (\min(\tilde{\alpha}x_i^2, 1) - \min(\alpha x_i^2, 1))^2 \right)} \qquad \text{by Cauchy-Schwarz}$$

for which we can bound the two terms separately.

To bound the first term, since we sum over only those terms where $|x_i| \leq \sqrt{\frac{1}{\min(\alpha_{\delta',8\eta},\tilde{\alpha})}}$ for all $i$, and by Lemma A.20 and A.21, $\min(\alpha_{\delta',8\eta}, \tilde{\alpha}) \geq \frac{0.0000354}{n}(\log \frac{1}{\delta'} + 24\eta n)$ with probability $1 - \frac{8}{11}\delta' \cdot e^{-24\eta n}$, we have that $x_i^2 \leq \frac{28249n}{\log \frac{1}{\delta'}+24\eta n}$ for all $i$. Since $X$ has mean 0 and variance 1, we know that $\mathbb{E}[x_i^2] \leq \mathbb{E}[X^2] = 1$, and $\mathrm{Var}[x_i^2] \leq \mathbb{E}[x_i^4] \leq \frac{28249n}{\log \frac{1}{\delta'}+24\eta n}\mathbb{E}[x_i^2] \leq \frac{28249n}{\log \frac{1}{\delta'}+24\eta n}$. Thus, by Bernstein's inequality,

$$\mathbb{P}\left[ \sum_{\leq} x_i^2 \geq 19025n \right] \leq \mathbb{P}\left[ \frac{1}{n}\sum_{\leq} x_i^2 - 1 \geq 19024 \right]$$
$$\leq \mathbb{P}\left[ \frac{1}{n}\sum_{\leq} x_i^2 - \mathbb{E}[x_i^2] \geq 19024 \right]$$
$$\leq 2\exp\left( -\frac{19024^2 n}{56498\frac{n}{\log \frac{1}{\delta'}+24\eta n} + 56498 \cdot 19024\frac{n}{\log \frac{1}{\delta'}+24\eta n}/3} \right)$$
$$\leq 2\exp\left( -1.1(\log \frac{1}{\delta'} + 24\eta n) \right)$$
$$\leq 2\exp\left( -(\log \frac{11}{\delta'} + 24\eta n) \right) \quad \text{for suff. small } \delta$$

$$= \frac{2}{11}\delta' \cdot e^{-24\eta n}$$

In other words, conditioning on Lemma A.20 and A.21 both holding, with probability at least $1 - \frac{2}{11}\delta' \cdot e^{-24\eta n}$, $\sum_{\leq} x_i^2 \leq 19025n$.

To bound the second term, note that depending on the order between $\alpha := \alpha_{\delta',8\eta}$ and $\tilde{\alpha} := \tilde{\alpha}_{\delta',8\eta}$, either $\min(\alpha x_i^2, 1) - \min(\tilde{\alpha} x_i^2, 1) \geq 0$ for all $i$, or $\min(\tilde{\alpha} x_i^2, 1) - \min(\alpha x_i^2, 1) \geq 0$ for all $i$ holds. Without loss of generality, we assume that $\tilde{\alpha} \geq \alpha$, and that $\min(\tilde{\alpha} x_i^2, 1) - \min(\alpha x_i^2, 1) \geq 0$ for all $i$.

Then, we have

$$\sum_{\leq} \left(\min(\tilde{\alpha} x_i^2, 1) - \min(\alpha x_i^2, 1)\right)^2 \leq \sum_i \left(\min(\tilde{\alpha} x_i^2, 1) - \min(\alpha x_i^2, 1)\right)^2 \leq \sum_i \left(\min(\tilde{\alpha} x_i^2, 1) - \min(\alpha x_i^2, 1)\right)$$

since $\tilde{\alpha} \geq \alpha$, and that $\min(\tilde{\alpha} x_i^2, 1) - \min(\alpha x_i^2, 1) \geq 0$ for all $i$.

To further upper bound the last quantity, we have

$$\sum_i \left(\min(\tilde{\alpha} x_i^2, 1) - \min(\alpha x_i^2, 1)\right)$$

$$= \sum_i \min(\tilde{\alpha} x_i^2, 1) - \sum_i \min(\alpha x_i^2, 1)$$

$$\leq \sum_i \min(\alpha_{\delta',10\eta} x_i^2, 1) - \sum_i \min(\alpha_{\delta',8\eta} x_i^2, 1) \quad \text{by Corollary A.3 and by the definition of } \alpha$$

$$= \frac{1}{3}\log\frac{1}{\delta'} + 10\eta n - \frac{1}{3}\log\frac{1}{\delta'} - 8\eta n$$

$$= 2\eta n$$

Finally, summarizing, we have that with high probability at least $1 - \frac{10}{11}\delta' \cdot e^{-24\eta n}$:

$$\sqrt{\left(\sum_i x_i^2\right)\left(\sum_i \left(\min(\tilde{\alpha} x_i^2, 1) - \min(\alpha x_i^2, 1)\right)^2\right)}$$

$$\leq \sqrt{19025n \cdot 2\eta n}$$

$$= n\sqrt{38050\eta} \leq 195.065n\sqrt{\eta}$$

as desired. $\qquad\square$

We now present the counterpart of Proposition A.11, bounding the error due to the adversarial corruption with tighter failure probability.

**Proposition A.22.** *Suppose both $\frac{\log\frac{1}{\delta}}{n}$ and $\delta$ are bounded by some small universal constant. Let $\tilde{\alpha}$ be the influence parameter computed from the **corrupted** samples with robustness level $\frac{1}{3}\log\frac{1}{\delta'} + 8\eta n$, namely, $\tilde{\alpha} := \tilde{\alpha}_{\delta',8\eta}$. Then with probability at least $1 - \frac{4}{11}\delta'e^{-24\eta n}$, the mean estimate using $\tilde{\alpha}$ on the **clean** samples differ from the mean estimate using $\tilde{\alpha}$ on the **corrupted** samples by at most $26.411\sqrt{\eta}$, i.e.,*

$$\left|\sum_i \tilde{x}_i(1 - \min(\tilde{\alpha}\tilde{x}_i^2, 1)) - \sum_i x_i(1 - \min(\tilde{\alpha} x_i^2, 1))\right| \leq 26.411n\sqrt{\eta}$$

*Proof.* With arguments identical to that in the proof of Proposition A.11, the maximum error the adversary can incur by corrupting $\eta n$ terms is $\frac{4\eta n}{3\sqrt{3\tilde{\alpha}}}$. To arrive at a similar lower bound of $\tilde{\alpha}$ in terms of $\eta$, notice that by Lemma A.21, we have that with probability at least $1 - \frac{4}{11}\delta' \cdot e^{-24\eta n}$,

$$\tilde{\alpha} \geq 0.0000354\frac{1}{n}\left(\log\frac{1}{\delta'} + 24\eta n\right) \geq 0.0000354\frac{1}{n}(24\eta n) \geq 0.0008496\eta$$

Thus, we have that

$$\left|\sum_i \tilde{x}_i(1 - \min(\tilde{\alpha}\tilde{x}_i^2, 1)) - \sum_i x_i(1 - \min(\tilde{\alpha} x_i^2, 1))\right| \leq \frac{4\eta n}{3\sqrt{3\tilde{\alpha}}} \leq \frac{4\eta n}{3\sqrt{3 \cdot 0.0008496\eta}} \leq 26.411n\sqrt{\eta}$$

$\qquad\square$

Equipped with Propositions A.19 and A.22, we are ready to formally prove Theorem 2.2.

*Proof of Theorem 2.2.* We will start by assuming that the underlying distribution has mean 0 and variance 1 (by the shift-and-scale equivariance of Estimator 1), and furthermore, that the initial estimate $\kappa$ computed in Step 1 is exactly the true mean of 0. At the end of this proof, we will show that the $\kappa = 0$ assumption introduces only a negligible amount of mean estimation error.

Fixing the initial estimate $\kappa = 0$, then, Estimator 1 on input the clean samples $X$ and confidence parameter $\delta$ computes the influence parameter $\alpha := \alpha_{\delta,0}$, and returns an estimate $\hat{\mu}_{clean,\alpha} := \frac{1}{n} \sum_i x_i(1 - \min(\alpha x_i^2, 1))$. Note that $\alpha_{\delta,0} = \alpha_{\delta',8\eta}$, since $\frac{1}{3}\log\frac{1}{\delta} = \frac{1}{3}\log\frac{1}{\delta'} + 8\eta n$ by definition of $\delta'$.

As a part of their analysis, Lee & Valiant (2022) showed that

$$|\hat{\mu}_{clean,\alpha}| \le (\sqrt{2} + o(1))\sqrt{\frac{\log\frac{1}{\delta}}{n}}$$

with probability at least $1 - \delta$. We note that the above guarantee can in fact be slightly strengthened, so that the failure probability becomes $\delta/22$, at the expense of a slightly increased $o(1)$ by no more than a constant multiplicative factor.

Now we consider changing the "$\alpha$" value from $\alpha$ to $\tilde{\alpha}$ computed from the corrupted samples, but applying these $\alpha$ influence parameters on the same clean sample set, and bound the difference in the resulting mean estimates. Specifically, consider $\tilde{\alpha} := \tilde{\alpha}_{\delta',8\eta}$, which is the influence parameter computed with robustness level $\frac{1}{3}\log\frac{1}{\delta'} + 8\eta n$ from the **corrupted** samples. Let the mean estimate of using $\tilde{\alpha}$ on the **clean** samples be $\hat{\mu}_{clean,\tilde{\alpha}} := \frac{1}{n} \sum_i x_i(1 - \min(\tilde{\alpha} x_i^2, 1))$. Then, by Proposition A.19, with probability at least $1 - \frac{10}{11}\delta' \cdot e^{-24\eta n} = 1 - \frac{10}{11}\delta$ we have

$$|\hat{\mu}_{clean,\tilde{\alpha}} - \hat{\mu}_{clean,\alpha}| = \frac{1}{n}\left|\sum_i x_i\min(\tilde{\alpha} x_i^2, 1) - \sum_i x_i\min(\alpha x_i^2, 1)\right| \le 195.065\sqrt{\eta}$$

Next we consider the effect of replacing the effect of corruption, but after fixing the influence parameter to be $\tilde{\alpha}$. Specifically, define $\hat{\mu}_{corrupt,\tilde{\alpha}}$ to be the mean estimate of the corrupted samples using $\tilde{\alpha}$, namely $\frac{1}{n} \sum_i \tilde{x}_i(1 - \min(\tilde{\alpha}\tilde{x}_i^2, 1))$. Then, by Proposition A.22, with probability 1 *conditioned on* Proposition A.19 and thus Lemma A.21 holding, we have

$$|\hat{\mu}_{corrupt,\tilde{\alpha}} - \hat{\mu}_{clean,\tilde{\alpha}}| = \frac{1}{n}\left|\sum_i \tilde{x}_i(1 - \min(\tilde{\alpha}\tilde{x}_i^2, 1)) - \sum_i x_i(1 - \min(\tilde{\alpha} x_i^2, 1))\right| \le 26.411\sqrt{\eta}$$

By union bound and triangle inequality, summing over all three error terms, we have that with probability at least $1 - \frac{21}{22}\delta$, $\hat{\mu}_{corrupt,\tilde{\alpha}}$ satisfies

$$|\hat{\mu}_{corrupt,\tilde{\alpha}}| \le (\sqrt{2} + o(1))\sqrt{\frac{\log\frac{1}{\delta}}{n}} + 221.476\sqrt{\eta} \le (\sqrt{2} + o(1))\sqrt{\frac{\log\frac{1}{\delta}}{n}} + 222\sqrt{\eta}$$

Finally, observe that on a mean-0-variance-1 distribution, and with $\eta$-corruption, the only difference between $\hat{\mu}_{corrupt,\tilde{\alpha}}$ and the output $\hat{\mu}$ of Estimator 1 (on input as stated in the theorem statement) lies in $\hat{\mu}_{corrupt,\tilde{\alpha}}$ assuming that the initial mean estimate $\kappa$ was the true mean of 0. On the other hand, $\hat{\mu}$ (Estimator 1) computes $\kappa$ from data.

We use Fact A.17 as well as the following fact/assumption about the robustness of the initial estimate $\kappa$ against adversarial corruption to analyze the effect of the $\kappa$ assumption. Fact A.23 is known to hold true for median-of-means as a folklore result, which we show in Appendix F for completeness. As mentioned in Estimator 1, we can choose to use other estimators to compute the initial estimate $\kappa$ as long as the estimator satisfies Fact A.23.

**Fact A.23** (Folklore). *For any distribution $D$ with mean $\mu$ and standard deviation $\sigma$, let $\tilde{X}$ be a set of $n$ $\eta$-corrupted samples from $D$. Assuming that $\eta \le \frac{1}{24n}\log\frac{1}{\delta}$, the median-of-means estimate $\kappa$ from grouping samples into $O(\log\frac{1}{\delta})$ buckets, on input $\tilde{X}$, satisfies*

$$\mathbb{P}\left(|\kappa - \mu| \ge O\left(\sigma\sqrt{\frac{\log\frac{1}{\delta}}{n}}\right)\right) \le \frac{1}{22}\delta$$

By Fact A.23, with probability at least $1 - \frac{1}{22}\delta$, $|\kappa| \leq O\left(\sqrt{\frac{\log \frac{1}{\delta}}{n}}\right)$, and by the Lipschitz bound of Fact A.17, we can bound the (absolute) difference between $\hat{\mu}$ and $\hat{\mu}_{corrupt,\tilde{\alpha}}$ by

$$|\hat{\mu} - \hat{\mu}_{corrupt,\tilde{\alpha}}| \leq O\left(|\kappa|\sqrt{\frac{\log \frac{1}{\delta}}{n}}\right) \leq O\left(\sqrt{\left(\frac{\log \frac{1}{\delta}}{n}\right)^2}\right) \leq o\left(\sqrt{\frac{\log \frac{1}{\delta}}{n}}\right)$$

Thus, on a mean-0-variance 1 distribution, with probability at least $1 - \delta$ over the clean samples (and with an arbitrary $\eta$-corruption), Estimator 1 outputs

$$|\hat{\mu}| \leq (\sqrt{2} + o(1))\sqrt{\frac{\log \frac{1}{\delta}}{n}} + 222\sqrt{\eta}$$

Combined with the affine invariance of Estimator 1 as stated in Fact A.17, if the underlying distribution instead had mean $\mu$ and variance $\sigma^2$, we have the mean estimation guarantee

$$|\hat{\mu} - \mu| \leq \sigma \cdot \left((\sqrt{2} + o(1))\sqrt{\frac{\log \frac{1}{\delta}}{n}} + 222\sqrt{\eta}\right)$$

as desired. $\qquad\square$

## B. Robustness of Estimator 1 in Weaker Models

As corollaries to our main results, stating that Estimator 1 is robust against adversarially corrupted data, we also show that Estimator 1 is robust against two (slightly weaker) contamination models, namely Huber contamination and TV contamination. For simplicity, we present and prove direct corollaries of Theorem 2.2 only. Corollaries of Theorem 2.3 follow similarly.

**Definition B.1** (Huber contamination (Huber, 1992)). Given a corruption parameter $\eta$ and a distribution $D$ on the uncorrupted data, we say that a set of $n$ samples is an $\eta$-Huber-contaminated sample from $D$ if it is drawn i.i.d. from some distribution $(1 - \eta)D + \eta E$ for an arbitrary distribution $E$.

The Huber contamination model (Huber, 1992) can be regarded as being weaker than the strong contamination model, because the corruption is always drawn randomly and obliviously from a fixed distribution $E$ chosen by the adversary; on the other hand, the adversary in the strong contamination model gets to choose the corruptions *adaptively* after inspecting all the samples. We point out however that, due to the random nature of the number of corrupted samples in the Huber model (and TV contamination model later in Definition B.5), these models are not strictly weaker than strong contamination, despite being "weaker in expectation". Nonetheless, by Chernoff bounds, the number of corrupted samples will concentrate to $O(\eta n)$ except with $\exp(-\Omega(\eta n))$ probability.

Later on, in Appendix C, we aim to show the neighborhood optimality of Estimator 1 as a corollary of its robustness against Huber contamination. In those results, we aim for a failure probability at most $\delta$ when the algorithm is given $\delta$ as input, as opposed to a failure probability that is (slightly) larger than $\delta$. Consequently, we write our theorem below (Theorem B.4) for Huber-contamination robustness with a failure probability $\delta/2 + \exp(-\Omega(\eta n))$, so that it is upper bounded by $\delta$ when $\eta$ is sufficiently large.[2]

We will show Theorem B.4 as a corollary of Corollary B.2, which is a variant of the strong-contamination robustness result in Theorem 2.2, with failure probability $\delta/2$ instead of $\delta$. Then, recalling that the probability of having too many corruptions is at most $\exp(-\Omega(\eta n))$, a union bound gives the target failure probability in Theorem B.4.

**Corollary B.2.** *Given any distribution $D$ with mean $\mu$ and variance $\sigma^2$, parameters $n, \delta, \eta > 0$, let $\tilde{X}$ be a set of $n$ $\eta$-corrupted samples from $D$.*

---

[2]We point out that, somewhat counter-intuitively, the larger $\eta$ is, and the stronger the corruption is, the less likely that Huber contamination produces many more corrupted samples than adversarial corruption.

*Suppose both $\frac{\log \frac{1}{\delta}}{n}$ and $\delta$ are bounded by some small universal constant, and suppose $\eta \leq \frac{1}{24n}\log\frac{1}{\delta}$. Then, with probability at least $1 - \frac{1}{2}\delta$ over the sampling process, Estimator 1 on input $\delta$ and $\tilde{X}$ will output an estimate $\hat{\mu}$ with error at most*

$$|\hat{\mu} - \mu| \leq \sigma \cdot \left( (1 + o(1))\sqrt{\frac{2\log\frac{1}{\delta}}{n}} + 222\sqrt{\eta} \right)$$

The proof of Corollary B.2 is almost identical to that of Theorem 2.2. The main difference is, instead of using Fact 1.1 to guarantee the performance of Estimator 1 on i.i.d. uncorrupted samples, we use a stronger variant which is Fact B.3 below. Specifically, Fact B.3 inputs the parameter $\delta$ to Estimator 1 but asks for a failure probability of $\delta/2$ instead of $\delta$, at the expense of a slightly larger "$o(1)$" term. Given the above target parameters, Fact B.3 is *not* a black-box corollary of Fact 1.1. Nonetheless, it follows directly from the analysis in Lee and Valiant's original paper (2022), so we state it without proof below.

**Fact B.3** (Lee & Valiant 2022). *Given any distribution $D$ with mean $\mu$ and variance $\sigma^2$, parameters $n, \delta > 0$, let $X$ be a set of $n$ independent samples from $D$. Then, with probability at least $1 - \delta/2$ over the sampling process, Estimator 1 on input $\delta$ and $X$ will output an estimate $\hat{\mu}$ with error at most*

$$|\hat{\mu} - \mu| \leq \sigma \cdot \left( (1 + o(1))\sqrt{\frac{2\log\frac{1}{\delta}}{n}} \right)$$

*Here, the $o(1)$ term tends to 0 as $\left( \frac{\log\frac{1}{\delta}}{n}, \delta \right) \to (0, 0)$ and, crucially, is independent of $D$.*

We can now state and prove the robustness of Estimator 1 against Huber contamination.

**Theorem B.4.** *Suppose both $\frac{\log\frac{1}{\delta}}{n}$ and $\delta$ are bounded by some small universal constant. Given any distribution $D$ with mean $\mu$ and variance $\sigma^2$, parameters $n, \delta, \eta > 0$, and a set $X$ of $n$ $\eta$-Huber-contaminated samples from $D$, for some $\eta \leq \frac{1}{24en}\log\frac{1}{\delta}$. Estimator 1 of LV22, when given access to $n, \delta$, and $X$ only, will, with probability at least $1 - (\frac{1}{2}\delta + \exp(-80\eta n))$ over the sampling process, yield an estimate $\hat{\mu}$ with error at most*

$$|\hat{\mu} - \mu| \leq \sigma \cdot \left( (1 + o(1))\sqrt{\frac{2\log\frac{1}{\delta}}{n}} + 222\sqrt{33\eta} \right)$$

*Proof.* Note that sampling from distribution $(1 - \eta)D + \eta E$ is exactly identical to sampling from $D$ with probability $1 - \eta$ and from $E$ with $\eta$. We bound the probability that such a sampling process samples more than $33\eta n$ samples from $E$.

Let $z_i$ denote the indicator variable for the event that sample $x_i$ is sampled from $E$. By multiplicative Chernoff:

$$\mathbb{P}[\sum_{i=1}^{n} z_i \geq 33\eta n] \leq \left( \frac{e^{33-1}}{33^{33}} \right)^{\eta n}$$
$$\leq \exp(-80\eta n)$$

Thus with probability at least $1 - \exp(-80\eta n)$, the Huber-contaminated sample set has at most $33\eta n$ samples corrupted.

Conditioned on this happening, Corollary B.2 applies to the set of contaminated samples $X$ with an adversary capable of arbitrarily corrupting $33\eta n$ samples, with probability at least $1 - \frac{1}{2}\delta$. Our theorem thus follows from a union bound. □

Generalizing from Huber contamination slightly is the TV contamination model, which draws samples from a distribution $D'$ that is within $\eta$ TV distance from the genuine underlying distribution $D$.

**Definition B.5** (TV contamination (Diakonikolas & Kane, 2023)). *Given a corruption parameter $\eta$ and a distribution $D$ on the uncorrupted data, we say that a set of $n$ samples is an $\eta$-TV-contaminated sample from $D$ if it is drawn i.i.d. from some distribution $D'$ such that $D_{\text{TV}}(D, D') \leq \eta$.*

And correspondingly, we have the following theorem showing that Estimator 1 is robust against TV contamination.

**Theorem B.6.** *Suppose both $\frac{\log \frac{1}{\delta}}{n}$ and $\delta$ are bounded by some small universal constant. Given any distribution $D$ with mean $\mu$ and variance $\sigma^2$, parameters $n, \delta, \eta > 0$, and a set $X$ of $n$ $\eta$-TV-contaminated samples from $D$, for some $\eta \leq \frac{1}{24en}\log\frac{1}{\delta}$. Estimator 1 of LV22, when given access to $n, \delta$, and $X$ only, will, with probability at least $1 - (\frac{1}{2}\delta + \exp(-80\eta n))$ over the sampling process, yield an estimate $\hat{\mu}$ with error at most*

$$|\hat{\mu} - \mu| \leq \sigma \cdot \left( (1 + o(1))\sqrt{\frac{2\log\frac{1}{\delta}}{n}} + 222\sqrt{33\eta} \right)$$

*Proof.* Let $D'$ be any distribution such that $D_{TV}(D, D') \leq \eta$. There exists a coupling between $D$ and $D'$ such that for any sample index $i$, the probability that the sample $x_i$ from $D$ and the sample $x_i'$ from $D'$ differs is at most $\eta$.

Let $z_i$ denote the indicator variable for the event that $x_i$ differs from $x_i'$. By multiplicative chernoff:

$$\mathbb{P}[\sum_{i=1}^{n} z_i \geq 33\eta n] \leq \left( \frac{e^{33-1}}{33^{33}} \right)^{\eta n}$$
$$\leq \exp(-80\eta n)$$

Thus with probability at least $1 - \exp(-80\eta n)$, the TV-contaminated sample set has at most $33\eta n$ samples corrupted.

Conditioned on this happening, Corollary B.2 applies to the set of contaminated samples $X$ with an adversary capable of arbitrarily corrupting $33\eta n$ samples, with probability at least $1 - \frac{1}{2}\delta$. Our theorem thus follows from a union bound. $\square$

## C. Neighborhood Optimality of Estimator 1

Neighborhood optimality is a notion of instance-by-instance optimality beyond the worst-case defined in Dang et al. (2023). In this section, we show that Estimator 1 is also asymptotically neighborhood optimal as a corollary to its robustness against TV contamination. We point out that our result matches that of Dang et al. (2023), which states that the median-of-means estimator is asymptotically neighborhood optimal, with the same choice of neighborhood structure and asymptotes.

For mean estimation in $\mathbb{R}$, Estimator 1 achieves the optimal sub-Gaussian rate even up to the constants. Their optimality is only in the worst-case regime, i.e., there *exists* an instance of mean estimation on which no estimator can perform better. A natural question to this observation is: Can we beat the sub-Gaussian rate for some distributions?

Dang et al. (2023) presented the following new universal estimation lower bound function:

**Definition 2.5** (Dang et al. 2023). Given a (continuous) distribution $D$ with mean $\mu$ and a real number $t \in [0, 1]$, define the *t-trimming* operation on $D$ as follows: select a radius $r$ such that the probability mass in $[\mu - r, \mu + r]$ equals $1 - t$; then, return the distribution $D$ **conditioned** on lying in $[\mu - r, \mu + r]$.

Given $n$ and $\delta$, define the trimmed distribution $D_{n,\delta}^*$ to be the $\frac{0.45}{n}\log\frac{1}{\delta}$-trimmed version of $D$. When $\delta$ is implicit, we may denote this as $D_n^*$. Now define the error function $\epsilon_{n,\delta}(D) = |\mu - \mu_n^*| + \sigma_n^*\sqrt{\frac{\log\frac{1}{\delta}}{n}}$, where $\mu_n^*$ and $\sigma_n^*$ are the mean and standard deviation of $D_n^*$ respectively.

The ground truth error function $\epsilon_{n,\delta}$ in Definition 2.5 applies simultaneously for all distributions with a finite mean, not just for the worst-case distribution, in the sense that for any distribution $p$ we can construct a neighboring distribution $q$ that is indistinguishable with probability at least $1 - \delta$ with $n$ samples, but the mean of $p$ and $q$ are well-separated by $O(\epsilon_{n,\delta}(p))$ in distance. Thus, no estimator can beat the error function on $p$ and $q$ simultaneously, since any estimator that can will be a distinguisher between $p$ and $q$.

To formalize this notion of lower bound and optimality, Dang et al. (2023) presented the following pair of definitions:

**Definition 5.1** (Neighborhood Pareto Bounds (Dang et al., 2023)). Let $n$ be the number of samples and $\delta$ be the failure probability. Given a neighborhood function $N_{n,\delta} : \mathcal{P}_1 \to 2^{\mathcal{P}_1}$, we say that the error function $\epsilon_{n,\delta}(D) : \mathcal{P}_1 \to \mathbb{R}_0^+$ is a neighborhood Pareto bound for $\mathcal{P}_1$ with respect to $N_{n,\delta}$ if for all distributions $D \in \mathcal{P}_1$, *no* estimator $\hat{\mu}$ taking $n$ i.i.d. samples can simultaneously achieve the following two conditions:

- For all $D' \in N_{n,\delta}(D)$, with probability $1 - \delta$ over the $n$ i.i.d. samples from $D'$, we have $|\hat{\mu} - \mu_{D'}| \leq \epsilon_{n,\delta}(D')$.

- With probability $1 - \delta$ over the $n$ i.i.d. samples from $D$, $|\hat{\mu} - \mu_D| < \epsilon_{n,\delta}(D)$.

**Definition 5.2** (($\kappa, \tau$)-Neighborhood Optimal Estimators (Dang et al., 2023)). Let $\kappa > 1$ be a multiplicative loss factor in estimation error, and $\tau > 1$ be a multiplicative loss factor in sample complexity.
Given the parameters $\kappa, \tau > 1$, sample complexity $n$, failure probability $\delta$ and neighborhood function $N_{n,\delta}$, a mean estimator $\hat{\mu}$ is ($\kappa, \tau$)-neighborhood optimal with respect to $N_{n,\delta}$ if there exists an error function $\epsilon_{n,\delta}(D)$ such that $\min(\epsilon_{n/\tau,\delta}(D), \epsilon_{n,\delta}(D))$ is a neighborhood Pareto bound[3], and $\hat{\mu}$ gives estimation error at most $\kappa \cdot \epsilon_{n,\delta}(D)$ with probability at least $1 - \delta$ when taking $n$ i.i.d. samples from any distribution $D \in \mathcal{P}_1$.

Definition 5.1 enforces that it is impossible to "beat" the neighborhood Pareto bound locally, performing as good over the local neighborhood, while strictly better on the center $p$. It essentially enforces admissibility over every such local neighborhood, forming a smooth interpolation between instance optimality and admissibility, which are both classical definitions of optimality beyond the worst-case that fails in context of mean estimation in $\mathbb{R}$.

Dang et al. (2023) showed that there exists a neighborhood function $N_{n,\delta}$ for which $\epsilon_{n,\delta}$ as defined in Definition 2.5 is a neighborhood Pareto bound, and for which median-of-means is ($\kappa, 3$)-neighborhood optimal for some sufficiently large constant $\kappa$. We capture the necessary components of their analysis and state them without proof in the following fact:

**Fact C.1.** *There exists a neighborhood function $N_{n,\delta} : \mathcal{P}_1 \to 2^{\mathcal{P}_1}$ for which $\epsilon_{n,\delta}$ as defined in Definition 2.5 is a neighborhood Pareto bound. Any estimator that obtains estimation error $O(\epsilon_{n,\delta}(p))$ for all distributions $p \in \mathcal{P}_1$ is ($\kappa, 3$)-neighborhood optimal with respect to $N_{n,\delta}$ for some sufficiently large constant $\kappa$.*

Thus, to show that Estimator 1 is neighborhood optimal, it suffices to show that it asymptotically matches the performance of the ground truth error bound $\epsilon_{n,\delta}$. We present a reanalysis of Estimator 1 using its robustness against TV contamination that proves its neighborhood optimality in such a way.

**Theorem C.2.** *Suppose both $\frac{\log \frac{1}{\delta}}{n}$ and $\delta$ are bounded by some small universal constant. Denote by $N_{n,\delta}$ the neighborhood function whose existence is implied by Fact C.1. Estimator 1 is ($\kappa, 3$)-neighborhood optimal with respect to $N_{n,\delta}$ for some sufficiently large constant $\kappa$.*

*Proof.* Let $\epsilon_{n,\delta}$ be the ground truth error function as defined in Definition 2.5. By Fact C.1, it suffices to show that Estimator 1 achieves an error rate of $O(\epsilon_{n,\delta}(p))$ for all distributions $p$.

Let $p$ be any distribution with mean $\mu_p$. Let $c$ be the constant such that $\frac{c}{n} \log \frac{2}{\delta} = \frac{1}{80n} \log \frac{1}{\delta} \leq \frac{1}{24en} \log \frac{1}{\delta}$. Let $p_n^*$ be the $\frac{c}{n} \log \frac{2}{\delta}$-trimmed version of $p$ as defined in Definition 2.5, and let $\mu_{p_n^*}$ and $\sigma_{p_n^*}$ be the mean and standard deviation of $p_n^*$ respectively. Let $\epsilon_{n,\delta}$ be the error function as defined in Definition 2.5.

Observe that $p_n^*$ satisfies that $D_{TV}(p_n^*, p) = \frac{c}{n} \log \frac{2}{\delta}$. Thus, by Theorem B.6 with $D = p_n^*$ and $\eta = \frac{c}{n} \log \frac{2}{\delta}$, we have that with probability at least $1 - (\frac{1}{2}\delta + \exp(-80\eta n)) = 1 - \delta$,

$$|\hat{\mu} - \mu_{p_n^*}| \leq \sigma_{p_n^*} \cdot \left( (1 + o(1))\sqrt{\frac{2\log \frac{1}{\delta}}{n}} + 222\sqrt{\frac{33c \log \frac{2}{\delta}}{n}} \right)$$

$$= O\left( \sigma_{p_n^*} \sqrt{\frac{\log \frac{1}{\delta}}{n}} \right)$$

Thus by the triangle inequality, we have that with probability at least $1 - \delta$,

$$|\hat{\mu} - \mu_p| \leq |\mu_p - \mu_{p_n^*}| + |\hat{\mu} - \mu_{p_n^*}| \leq |\mu_p - \mu_{p_n^*}| + O\left( \sigma_{p_n^*} \sqrt{\frac{\log \frac{1}{\delta}}{n}} \right) = O(\epsilon_{n,\delta})$$

Combined with Fact C.1, this implies that Estimator 1 is indeed ($\kappa, 3$)-neighborhood optimal with respect to $N_{n,\delta}$ for some sufficiently large constant $\kappa$, as desired. $\square$

---

[3]While it is intuitive to expect that an error function decreases in $n$, it might not be true in general. Indeed, the function used by Dang et al. (2023) (Definition 2.5) is not necessarily monotonic. This is the reason for the "$\min$" in the neighborhood Pareto bound requirement.

We point out that the proof of Theorem B.6 and Theorem C.2 does not depend on the specific characteristics of Estimator 1, and instead only relies on Theorem 2.2 holding. Thus, we can obtain a similar neighborhood optimality result for any mean estimator that is sub-Gaussian and robust against adversarial corruption, and enjoys asymptotically matching bounds as in Theorem 2.2:

**Corollary C.3.** *Any estimator that, when given $\delta > 0$ and a set of $n$ $\eta$-corrupted sample from distribution $D$ with mean $\mu$ and variance $\sigma^2$, yields a mean estimate $\hat{\mu}$ satisfying*

$$|\hat{\mu} - \mu| \leq O\left(\sigma\left(\sqrt{\frac{\log \frac{1}{\delta}}{n}} + \sqrt{\eta}\right)\right)$$

*is $(\kappa, 3)$-neighborhood optimal with respect to $N_{n,\delta}$ for some sufficiently large constant $\kappa$.*

## D. Remaining Proofs of Section 6

In Section 6, we outlined the proof strategy of our main theorem for low moment performances of Estimator 1, which follows that of Lee & Valiant (2022), and provided the statement of the reformulation of Estimator 1 as a 2-parameter $\psi$-estimator. In this section, we present the full proof of Theorem 2.4, following the structure and organization of Lee & Valiant (2022). We present each component lemma along with the intuition behind it, and refer the reader to Lee & Valiant (2022) for more detailed motivations and discussions. For completeness, we restate our main theorem:

**Theorem 2.4.** *Given any distribution $D$ with mean $\mu$ and $z^{th}$ moment $M_z$ for some $z \in (1, 2)$, let $X$ be a set of $n$ i.i.d. samples from $D$. Then, with probability at least $1 - \delta$ over the randomness of $X$, Estimator 1 on input $\delta$ and $X$ will output an estimate $\hat{\mu}$ with error at most*

$$|\hat{\mu} - \mu| \leq (M_z)^{\frac{1}{z}} \cdot (1 + o(1)) \left(c_z \frac{\log \frac{1}{\delta}}{n}\right)^{1 - \frac{1}{z}}$$

*where $c_z = 2(5.6)^{\frac{1}{z-1} - 1}$. Here, the $o(1)$ term tends to $0$ as $\left(\frac{\log \frac{1}{\delta}}{n}, \delta\right) \to (0, 0)$, in a manner independent of $D$ and independent of $z$.*

From the structural properties of Estimator 1 stated explicitly in Fact A.17, we make the simplifying assumption that the underlying distribution $D$ has mean 0 and $z^{th}$ moment $M_z = 1$, and that the initial estimate $\kappa$ in Step 1 of Estimator 1 is replaced by 0. For the rest of the appendix, we will refer to the error bound $(c_z \log \frac{1}{\delta}/n)^{1-1/z}$ as $\epsilon$ (omitting the $1 + o(1)$ factor).

Definition 6.1 reduced proving Chernoff bounds for Estimator 1, which is a sum of *dependent* terms, to proving Chernoff bounds for the sums of independent terms of the 2-parameter $\psi$-estimator. Thus, it suffices to show that with high probability, the $\psi$-estimator in Definition 6.1 returns an estimate $\hat{\mu}$ that is close to 0—or equivalently, *every* pair $(\hat{\mu}', \hat{\alpha}')$ with $\hat{\mu}'$ far away from 0 must satisfy $\psi(X, \hat{\mu}', \hat{\alpha}') \neq 0$. We turn to analyze the following proposition, capturing this reduction:

**Proposition D.1.** *There exists a universal constant $c > 0$ such that, for all $1 < z \leq 2$, fixing $\epsilon' = \left(1 + \frac{c \log \log \frac{1}{\delta}}{\log \frac{1}{\delta}}\right)\epsilon$ where $\epsilon = \left(c_z \frac{\log \frac{1}{\delta}}{n}\right)^{1 - \frac{1}{z}}$ and $c_z = 2(5.6)^{\frac{1}{z-1} - 1}$, we have that for all distributions $D$ with mean 0 and $z^{th}$ moment 1, with probability at least $1 - \frac{\delta}{2}$ over the set of samples $X$, for all $\hat{\mu}, \hat{\alpha}$ where $|\hat{\mu}| > \epsilon'$ and $\hat{\alpha} > 0$, the vector $\psi(X, \hat{\mu}, \hat{\alpha}) \neq 0$.*

We stress that the universal constant $c$ in fact *does not* depend on $z$.

Towards proving Proposition D.1, we extend Lee & Valiant (2022)'s strategy, analyzing the function $\psi(X, \hat{\mu}, \hat{\alpha})$ on a finite bounded mesh of values of $\hat{\mu}, \hat{\alpha}$ covering the most delicate range for our analysis, and show via standard $\epsilon$-net arguments that the proposition holds for *any* choice of $|\hat{\mu}| > \epsilon'$, $\hat{\alpha}$. Specifically, we show that $\psi(X, \hat{\mu}, \hat{\alpha})$, on the finite mesh, is far from the origin in some direction, and is Lipschitz within the mesh. We then show how to reduce the analysis outside of the mesh to that inside the mesh.

To find a mesh to cover the relevant range of $\hat{\alpha}$ in the right "scale-invariant" way, we reparameterize $\hat{\alpha}$ by $\hat{w} := \frac{\log^2(1/\delta)}{3\hat{\alpha}n^2\epsilon^2}$, where $\epsilon$ in the denominator (as defined in Proposition D.1) encodes the desired dependence on $z$. We choose an evenly-spaced mesh over $\hat{w} \in [0.05, 555]$. This is analogous to the mesh on "$\hat{v}$" in Lee & Valiant (2022)'s analysis.

The main technical component of our (and Lee & Valiant (2022)'s) proof strategy is the proof that for each $\hat{\mu}, \hat{w} = \frac{\log^2(1/\delta)}{3\hat{\alpha}n^2\epsilon^2}$ from the finite mesh we analyze, $\psi(X, \hat{\mu}, \hat{\alpha})$ is far away from $\mathbf{0}$ *in some direction*. We linearize this claim and prove a stronger result, that there exists a specific direction $\mathbf{d}(\hat{w})$ such that with high probability, $\psi(X, \hat{\mu}, \hat{\alpha})$ is more than $\frac{1}{\log(1/\delta)}$ distance away from the origin in direction $\mathbf{d}$. More formally, we prove the following lemma:

**Lemma D.2.** *Let $D$ be an arbitrary distribution with mean $0$ and $z^{th}$ moment $1$ for some $1 < z \le 2$. There exists a universal constant $c$ independent of $z$ where the following is true. Fixing $\hat{\mu} = \left(1 + \frac{c \log \log \frac{1}{\delta}}{\log \frac{1}{\delta}}\right) \epsilon$ where $\epsilon = \left(c_z \frac{\log \frac{1}{\delta}}{n}\right)^{1 - \frac{1}{z}}$ and $c_z = 2(5.6)^{\frac{1}{z-1} - 1}$, then for all $\delta$ smaller than some universal constant, and for all $\hat{w} \in [0.05, 555]$, there exists a vector $\mathbf{d}(\hat{w}) := (d_\mu, d_\alpha)$ where $d_\mu \ge 0$, and both $\frac{n\epsilon}{\log \frac{1}{\delta}}|d_\mu|$ and $|d_\alpha|$ are bounded by a universal constant, such that*

$$\mathbb{P}_{X \leftarrow D^n}\left(\mathbf{d}(\hat{w}) \cdot \psi\left(X, \hat{\mu} = \epsilon', \hat{\alpha} = \frac{\log^2(1/\delta)}{3\hat{w}n^2\epsilon^2}\right) > \frac{1}{\log \frac{1}{\delta}}\right) \ge 1 - \frac{\delta}{\log^4 \frac{1}{\delta}}$$

*Furthermore, for $\hat{w} = 0.05$ we have $d_\mu = \sqrt{3.75}\frac{\log \frac{1}{\delta}}{n\epsilon}$, $d_\alpha = \sqrt{3}$; and for $\hat{w} = 555$ we have $d_\mu = 0$, $d_\alpha < 0$.*

From here, Proposition D.1 follows from a union bound over pairs of $(\hat{\mu}, \hat{\alpha})$ on the finite mesh, the monotonicity of $\psi(X, \hat{\mu}, \hat{\alpha})$ beyond the mesh granted by the boundary conditions specified in Lemma D.2, and the Lipschitzness of $\psi(X, \hat{\mu}, \hat{\alpha})$ within the region covered by the mesh, formalized in the following lemma:

**Lemma D.3.** *Consider an arbitrary set of $n$ samples $X$. Consider the expressions $\psi_\mu(X, \hat{\mu}, \hat{\alpha})$, $\psi_\alpha(X, \hat{\alpha})$, reparameterized in terms of $\hat{w} = \frac{\log^2 \frac{1}{\delta}}{3\hat{\alpha}n^2\epsilon^2}$ in place of $\hat{\alpha}$. Suppose the equation $\psi_\alpha(X, \hat{\alpha}) = 0$ has a solution in the range $\hat{w} \in [0.05, 555]$. Then the functions $\frac{\log \frac{1}{\delta}}{n\epsilon}\psi_\mu(X, \hat{\mu}, \hat{\alpha})$ and $\psi_\alpha(X, \hat{\alpha})$ are Lipschitz with respect to $\hat{w}$ on the entire interval $\hat{w} \in [0.05, 555]$, with Lipschitz constant $c\log \frac{1}{\delta}$ for some universal constant $c$.*

With these components, we now prove Proposition D.1, and subsequently formally prove our main theorem, Theorem 2.4, before returning to prove Lemma D.2 and D.3 in Appendix D.1.

*Proof of Proposition D.1.* By symmetry, instead of considering positive and negative $\hat{\mu}$, it suffices to consider the case $\hat{\mu} > \epsilon'$ and show that this case succeeds with probability at least $1 - \frac{\delta}{4}$.

To prove the claim, we first prove a stronger statement on a restricted domain, that with probability at least $1 - \frac{\delta}{4}$ over the randomness of the sample set $X$, for each $\hat{w} \in [0.05, 555]$ there exists a vector $\mathbf{d} = (d_\mu, d_\alpha)$ such that $\mathbf{d} \cdot \psi(X, \epsilon', \hat{\alpha}) > 0$, with $d_\mu > 0$ throughout, and , for $\hat{w} = 0.05$ we have $d_\mu = \sqrt{3.75}\frac{\log \frac{1}{\delta}}{n\epsilon}$, $d_\alpha = \sqrt{3}$; and for $\hat{w} = 555$ we have $d_\mu = 0$, $d_\alpha < 0$.

We will first apply Lemma D.2 to each $\hat{w}$ in a discrete mesh: let $M$ consist of evenly spaced points between 0.05 and 555 with spacing $1/\log^3 \frac{1}{\delta}$ (thus with $\Theta(\log^3 \frac{1}{\delta})$ many points).

By Lemma D.2 and a union bound over these $\Theta(\log^3 \frac{1}{\delta})$ points, we have that with probability at least $1 - \frac{\delta}{\Theta(\log \frac{1}{\delta})}$ (which is at least $1 - \frac{\delta}{4}$ for $\delta$ smaller than some universal constant) over the set of $n$ samples $X$, for all $\hat{w} \in M$, there exists a vector $\mathbf{d}(\hat{w})$ such that $\mathbf{d}(\hat{w}) \cdot \psi(X, \hat{\mu} = \epsilon', \hat{\alpha}) > 1/\log \frac{1}{\delta}$, where $\mathbf{d}$ further satisfies the desired positivity and boundary conditions, and where both $\frac{n\epsilon}{\log \frac{1}{\delta}}|d_\mu|$ and $|d_\alpha|$ are bounded by a universal constant. For the rest of the proof, we will only consider sets of samples $X$ satisfying the above condition.

Now consider an arbitrary $\hat{w}' \in [0.05, 555] \setminus M$ and consider the vector $\psi$ evaluated at $\hat{\alpha}' = \frac{\log^2 \frac{1}{\delta}}{3n^2\epsilon^2\hat{w}'}$. We wish to extend the dot product inequality to hold also for $\hat{w}'$. If $\psi_\alpha \ne 0$ then there is nothing to prove: set $\mathrm{d}_\mu = 0$ and $d_\alpha = \text{sign}(\psi_\alpha)$; otherwise, $\psi_\alpha = 0$ means we may apply Lemma D.3 to conclude that both $\frac{\log \frac{1}{\delta}}{n\epsilon}\psi_\mu(X, \hat{\mu}, \hat{\alpha}')$ and $\psi_\alpha(X, \hat{\mu}, \hat{\alpha}')$ are Lipschitz with respect to $\hat{w}'$ on the interval $\hat{w}' \in [0.05, 555]$ with Lipschitz constant $\tilde{c}\log \frac{1}{\delta}$ for some universal constant $c$.

Consider the closest $\hat{w} \in M$ to $\hat{w}'$, which by definition of $M$ is at most $1/\log^3 \frac{1}{\delta}$ away. By assumption on $X$, there exists a vector $\mathbf{d}$ such that $\mathbf{d} \cdot \psi(X, \hat{\mu} = \epsilon', \hat{\alpha}) > 1/\log \frac{1}{\delta}$, with $d_\mu > 0$ and both $\frac{n\epsilon}{\log \frac{1}{\delta}}|d_\mu|$ and $|d_\alpha|$ are bounded by a universal constant. Because of the Lipschitz bounds on $\psi$, combined with the bounds on the size of $d_\mu$ and $d_\alpha$, we conclude that the Lipschitz constant of the dot product (treating the vector $\mathbf{d}$ as fixed) is $O(\log \frac{1}{\delta})$. Thus, the large positive dot product at $\hat{w}$ implies at least a positive dot product nearby at $\hat{w}'$: $\mathbf{d} \cdot \psi(X, \hat{\mu} = \epsilon', \hat{w}') > \frac{1}{\log \frac{1}{\delta}} - O(\log \frac{1}{\delta})\frac{1}{\log^3 \frac{1}{\delta}} > 0$, for sufficiently small $\delta$ as given in the proposition statement.

Having shown the stronger version of the claim for the restriction $\hat{\mu} = \epsilon'$ and $\hat{w} \in [0.05, 555]$ we now extend to the entire domain via three monotonicity arguments. Explicitly, assume the set of samples $X$ satisfies the dot product inequality above with the vector function $\mathbf{d}(\hat{w})$, where $\mathbf{d}(\hat{w})$ satisfies the boundary conditions at $\hat{w} = 0.05$ and $555$ specified in Lemma D.2. From this assumption, we will show that $\psi \neq 0$ for *any* positive $\hat{w} = \frac{\log^2 \frac{1}{\delta}}{3n^2 \epsilon^2 \hat{\alpha}}$, and for *any* $\hat{\mu} \geq \epsilon'$.

First consider $\hat{w} > 555$, still fixing $\hat{\mu} = \epsilon'$. The function $\psi_\alpha = \sum_{i=1}^n (\min(\hat{\alpha} x_i^2, 1) - \frac{1}{3n} \log \frac{1}{\delta})$ is an increasing function of $\hat{\alpha}$, and thus a decreasing function of $\hat{w} = \frac{\log^2 \frac{1}{\delta}}{3n^2 \epsilon^2 \hat{\alpha}}$. Since for $\hat{w} = 555$, the dot product $\mathbf{d} \cdot \psi > 0$ with $d_\mu = 0, d_\alpha < 0$, the dot product will thus remain positive for this same choice of $\mathbf{d}$ as we increase $\hat{w}$ from $555$.

Next, for $\hat{w} < 0.05$, again fixing $\hat{\mu} = \epsilon'$, we analogously show that the dot product of $\psi(X, \epsilon', \hat{\alpha})$ with the fixed vector $\mathbf{d}(0.05)$ will increase as we decrease $\hat{w}$. The $i$-th term in the sums defining $\psi_\mu$ or $\psi_\alpha$ depends on $\hat{\alpha}$ (and thus $\hat{w}$) only in the factor $\min(\hat{\alpha} x_i, 1)$. Further, there is no dependence unless the first term attains the min, namely $|x_i| \leq \sqrt{1/\hat{\alpha}}$, which in turn is upper bounded by $\sqrt{0.15} \frac{n\epsilon}{\log \frac{1}{\delta}}$ because of our assumption that $\hat{w} < 0.05$. Thus, the only $i$-th terms in the dot product which have $\hat{\alpha}$ dependency are simply equal to $d_\mu \hat{\alpha} x_i^3 + d_\alpha \hat{\alpha} x_i^2 = \hat{\alpha} x_i^2 (d_\alpha + x_i d_\mu)$. By our choice of $d_\mu(0.05) = \sqrt{3.75} \frac{\log \frac{1}{\delta}}{n\epsilon}$ and $d_\alpha = \sqrt{3}$ from Lemma D.2, the expression $(d_\alpha + x_i d_\mu) \geq \sqrt{3} - \sqrt{0.15}\sqrt{3.75}$ is thus always non-negative, and thus the overall dot product cannot decrease as we send $\hat{w}$ to $0$ as desired.

We have thus shown that, for all non-negative $\hat{\alpha} = \frac{\log^2 \frac{1}{\delta}}{3n^2 \epsilon^2 \hat{w}}$, there is a vector $\mathbf{d}$ with $d_\mu \geq 0$ whose dot product with $\psi(X, \epsilon', \hat{\alpha})$ is greater than $0$. We complete the proof by noting that the only dependence on $\hat{\mu}$ in $\psi$ is that $\psi_\mu$ is (trivially) increasing in $\hat{\mu}$. Since $d_\mu \geq 0$, increasing $\hat{\mu}$ from $\epsilon'$ will only increase the dot product, and thus the dot product remains strictly greater than $0$, implying that $\psi(X, \hat{\mu}, \hat{\alpha}) \neq 0$, as desired. $\qquad\square$

To close out this section, we formally show that Proposition D.1 implies our desired main theorem, Theorem 2.4 for completeness, using the following fact about the performance of the median-of-means estimator for the initial estimate $\kappa$:

**Fact D.4** (Bubeck et al. 2012). *For any distribution $D$ with mean $\mu$ and $z^{th}$ moment $\sigma_z$, the median-of-means estimate $\kappa$ from grouping samples into $O(\log \frac{1}{\delta})$ buckets, on input $n$ samples, satisfies*

$$\mathbb{P}\left(|\kappa - \mu| > O\left((\sigma_z)^{\frac{1}{z}} \left(\frac{\log \frac{1}{\delta}}{n}\right)^{1 - \frac{1}{z}}\right)\right) \leq \frac{1}{2}\delta$$

*Here, the big-O notation hides a universal constant that is crucially also independent of $z$.*

Again, we are free to choose to use other estimators for $\kappa$ in Step 1 of Estimator 1 as long as the above fact holds true for the estimator being used.

*Proof of Theorem 2.4.* We start by making the simplifying assumption that $\mu = 0$ and $M_z = 1$, and reformulate Estimator 1 with Step 1 replaced with $\kappa = 0$ as a 2-parameter $\psi$-estimator, that takes in $n$ independent samples $X = x_1, \cdots, x_n$ from $D$, and solves for the (unique) pair $\hat{\mu}, \hat{\alpha}$ satisfying $\psi_\mu = 0$ and $\psi_\alpha = 0$, where the functions are defined as follows:

$$\psi_\mu(X, \hat{\mu}, \hat{\alpha}) = \sum_{i=1}^n (\hat{\mu} - x_i(1 - \min(\hat{\alpha} x_i^2, 1)))$$

$$\psi_\alpha(X, \hat{\mu}, \hat{\alpha}) = \sum_{i=1}^n \left(\min(\hat{\alpha} x_i^2, 1) - \frac{1}{3n} \log \frac{1}{\delta}\right)$$

and outputs $\hat{\mu}$ in the solution. We denote by $\psi$ the 2-element vector $(\psi_\mu, \psi_\alpha)$.

By Proposition D.1, with probability at least $1 - \frac{\delta}{2}$, any $\hat{\mu}', \hat{\alpha}'$ with $|\hat{\mu}'| > (1 + o(1)) \left(c_z \frac{\log \frac{1}{\delta}}{n}\right)^{1 - \frac{1}{z}}$ and $\hat{\alpha}' > 0$ satisfies that $\psi(X, \hat{\mu}', \hat{\alpha}') \neq \mathbf{0}$, and thus the solution $\hat{\mu}$ found by Estimator 1 by solving $\psi = \mathbf{0}$ satisfies $|\hat{\mu}| \leq (1 + o(1)) \left(c_z \frac{\log \frac{1}{\delta}}{n}\right)^{1 - \frac{1}{z}}$.

We now remove the simplifying assumptions using the structural properties of Estimator 1. Let $\kappa$ be the initial estimate in Step 1 of Estimator 1, say, computed by median-of-means. By Fact D.4, with probability at least $1 - \frac{1}{2}\delta$,

$$|\kappa| \leq O\left(\left(\frac{\log \frac{1}{\delta}}{n}\right)^{1 - \frac{1}{z}}\right)$$

and by the Lipschitz bound of Fact A.17 with Lipschitz constant $O\left(\sqrt{\frac{\log \frac{1}{\delta}}{n}}\right)$, this incurs an error on Estimator 1 of at most

$$O\left(\left(\frac{\log \frac{1}{\delta}}{n}\right)^{1 - \frac{1}{z}} \cdot \sqrt{\frac{\log \frac{1}{\delta}}{n}}\right) \leq o\left(\left(\frac{\log \frac{1}{\delta}}{n}\right)^{1 - \frac{1}{z}}\right)$$

Since $c_z^{1 - \frac{1}{z}}$ is lower bounded by a universal constant, the above expression is also $o\left(\left(c_z \frac{\log \frac{1}{\delta}}{n}\right)^{1 - \frac{1}{z}}\right)$.

Now consider any distribution with mean $\mu$ and $z^{\text{th}}$ moment $M_z$. By the affine invariance of Estimator 1 of Fact A.17, Estimator 1 suffers a multiplicative factor of $(M_z)^{\frac{1}{z}}$ in the estimation error.

Thus with probability at least $1 - \delta$, we have

$$|\hat{\mu}(X) - \mu| \leq (M_z)^{\frac{1}{z}} \cdot (1 + o(1)) \left(c_z \frac{\log \frac{1}{\delta}}{n}\right)^{1 - \frac{1}{z}}$$

as desired. $\qquad\square$

## D.1. Proof of Lemma D.2 and D.3

In this section, we present the proof of Lemma D.2, motivated by the discussion in Section 6.2 and consequently Lee & Valiant (2022), modeling the worst-case Chernoff bound as a max-min linear programming game. We later present the short proof of Lemma D.3.

**Lemma D.2.** *Let $D$ be an arbitrary distribution with mean $0$ and $z^{\text{th}}$ moment $1$ for some $1 < z \leq 2$. There exists a universal constant $c$ independent of $z$ where the following is true. Fixing $\hat{\mu} = \left(1 + \frac{c \log \log \frac{1}{\delta}}{\log \frac{1}{\delta}}\right)\epsilon$ where $\epsilon = \left(c_z \frac{\log \frac{1}{\delta}}{n}\right)^{1 - \frac{1}{z}}$ and $c_z = 2(5.6)^{\frac{1}{z-1} - 1}$, then for all $\delta$ smaller than some universal constant, and for all $\hat{w} \in [0.05, 555]$, there exists a vector $\mathbf{d}(\hat{w}) := (d_\mu, d_\alpha)$ where $d_\mu \geq 0$, and both $\frac{n\epsilon}{\log \frac{1}{\delta}}|d_\mu|$ and $|d_\alpha|$ are bounded by a universal constant, such that*

$$\mathbb{P}_{X \leftarrow D^n}\left(\mathbf{d}(\hat{w}) \cdot \psi\left(X, \hat{\mu} = \epsilon', \hat{\alpha} = \frac{\log^2(1/\delta)}{3\hat{w}n^2\epsilon^2}\right) > \frac{1}{\log \frac{1}{\delta}}\right) \geq 1 - \frac{\delta}{\log^4 \frac{1}{\delta}}$$

*Furthermore, for $\hat{w} = 0.05$ we have $d_\mu = \sqrt{3.75}\frac{\log \frac{1}{\delta}}{n\epsilon}$, $d_\alpha = \sqrt{3}$; and for $\hat{w} = 555$ we have $d_\mu = 0$, $d_\alpha < 0$.*

*Proof.* We instead bound the contrapositive statement, namely,

$$\mathbb{P}_{X \leftarrow D^n}\left(\mathbf{d}(\hat{w}) \cdot \psi\left(X, \hat{\mu} = \epsilon', \hat{\alpha} = \frac{\log^2(1/\delta)}{3\hat{w}n^2\epsilon^2}\right) \leq \frac{1}{\log \frac{1}{\delta}}\right) \leq \frac{\delta}{\log^4 \frac{1}{\delta}}$$

We start by applying a standard Chernoff bound analysis

$$\mathbb{P}_{X \leftarrow D^n}\left(\mathbf{d}(\hat{w}) \cdot \psi\left(X, \hat{\mu}, \hat{\alpha}\right) \leq \frac{1}{\log \frac{1}{\delta}}\right)$$

$$= \mathbb{P}_{X \leftarrow D^n}\left(\exp\left(-\mathbf{d}(\hat{w}) \cdot \psi\left(X, \hat{\mu}, \hat{\alpha}\right)\right) \geq \exp\left(-\frac{1}{\log \frac{1}{\delta}}\right)\right)$$

$$\leq 2 \mathop{\mathbb{E}}_{X \leftarrow D^n} \left( \exp\left( -\mathbf{d}(\hat{w}) \cdot \psi\left( X, \hat{\mu}, \hat{\alpha} \right) \right) \right) \qquad \text{by Markov's and } \exp\left( -\frac{1}{\log \frac{1}{\delta}} \right) \leq 2 \text{ for suff. small } \delta$$

$$= 2 \mathop{\mathbb{E}}_{x \leftarrow D} \left( \exp\left( -\mathbf{d}(\hat{w}) \cdot \psi\left( x, \hat{\mu}, \hat{\alpha} \right) \right) \right)^n \qquad \text{by independence}$$

$$= 2 \left( \exp\left( -d_\mu \hat{\mu} + d_\alpha \frac{1}{3n} \log \frac{1}{\delta} \right) \mathop{\mathbb{E}}_{x \leftarrow D} \left( \exp\left( d_\mu x (1 - \min(\hat{\alpha} x^2, 1)) - d_\alpha \min(\hat{\alpha} x^2, 1) \right) \right) \right)^n$$

Motivated by our discussion in Section 6.2, we state the following technical claim, Lemma D.5, reminiscent of the constraint in (2). We formally prove Lemma D.5 in Appendix D.2.

**Lemma D.5.** *For all $\hat{w} \in [0.05, 555]$, there exists $a > 0, b$ such that for all $1 < z \leq 2$ and $y \in \mathbb{R}$, letting $c_z = 2(5.6)^{\frac{1}{z-1}-1}$, the following holds true:*

$$ay(1 - \min(y^2, 1)) - b \min(y^2, 1) \leq \log\left( 1 + ay + |y|^z (3\hat{w})^{\frac{z}{2}} c_z^{z-1} \left( -1 + \frac{a}{\sqrt{3\hat{w}}} - \frac{b}{3} \right) \right) \tag{4}$$

*where $a$ and $b$ are bounded by constants. Further, for $\hat{w} = 0.05$, the pair $a = 0.75, b = \sqrt{3}$ works.*

For $\hat{w} \in [0.05, 555)$, we use Lemma D.5, substituting $y = \sqrt{\hat{\alpha}} x$, to choose $d_\mu = \sqrt{\hat{\alpha}} x = \frac{\log \frac{1}{\delta}}{n \epsilon \sqrt{3\hat{w}}} a$, $d_\alpha = b$. In particular, for $\hat{w} = 0.05$, we have $d_\mu = \sqrt{3.75} \frac{\log \frac{1}{\delta}}{n\epsilon}$ and $d_\alpha = \sqrt{3}$. Then, the failure probability is bounded by

$$2 \left( \exp\left( -d_\mu \hat{\mu} + d_\alpha \frac{1}{3n} \log \frac{1}{\delta} \right) \mathop{\mathbb{E}}_{\substack{x \leftarrow D \\ y = x\sqrt{\hat{\alpha}}}} \left( 1 + ay + |y|^z (3\hat{w})^{\frac{z}{2}} c_z^{z-1} \left( -1 + \frac{a}{\sqrt{3\hat{w}}} - \frac{b}{3} \right) \right) \right)^n$$

$$= 2 \left( \exp\left( -d_\mu \hat{\mu} + d_\alpha \frac{1}{3n} \log \frac{1}{\delta} \right) \left( 1 + \left( \frac{\log \frac{1}{\delta}}{n\epsilon\sqrt{3\hat{w}}} \right)^z (3\hat{w})^{\frac{z}{2}} c_z^{z-1} \left( -1 + \frac{d_\mu \epsilon n}{\log \frac{1}{\delta}} - \frac{d_\alpha}{3} \right) \right) \right)^n$$

$$\text{since } D \text{ has mean } 0, z^{\text{th}} \text{ moment } 1$$

$$= 2 \left( \exp\left( -d_\mu \hat{\mu} + d_\alpha \frac{1}{3n} \log \frac{1}{\delta} \right) \left( 1 + \frac{\log \frac{1}{\delta}}{n} \left( -1 + \frac{d_\mu \epsilon n}{\log \frac{1}{\delta}} - \frac{d_v}{3} \right) \right) \right)^n$$

$$\leq 2 \left( \exp\left( -d_\mu \hat{\mu} + d_\alpha \frac{1}{3n} \log \frac{1}{\delta} + \frac{\log \frac{1}{\delta}}{n} \left( -1 + \frac{d_\mu \epsilon n}{\log \frac{1}{\delta}} - \frac{d_\alpha}{3} \right) \right) \right)^n$$

$$\text{since } 1 + x \leq e^x \text{ for any } x$$

$$= 2 \left( \exp\left( -d_\mu \hat{\mu} + d_\alpha \frac{1}{3n} \log \frac{1}{\delta} - \frac{1}{n} \log \frac{1}{\delta} + d_\mu \epsilon - d_\alpha \frac{1}{3n} \log \frac{1}{\delta} \right) \right)^n$$

$$= 2 \exp\left( -d_\mu n(\hat{\mu} - \epsilon) - \log \frac{1}{\delta} \right)$$

$$= 2 \exp\left( -\frac{a}{\sqrt{3\hat{w}}} c \log \log \frac{1}{\delta} - \log \frac{1}{\delta} \right) \qquad \text{by choice of } d_\mu \text{ and } \hat{\mu}$$

$$\leq \frac{\delta}{\log^4 \frac{1}{\delta}}$$

as desired for large enough $c$, since $\frac{a}{\sqrt{3\hat{w}}}$ is greater than some positive constant.

For $\hat{w} = 555$, we use the following technical claim:

**Claim D.6.** *For all $1 < z \leq 2$, for all values of $y$, the following holds true:*

$$4 \min(y^2, 1) \leq \log(1 + 54|y|^z)$$

which is easily verifiable by two subclaims in the form of $4y^2 \leq \log(1 + 54y^2)$ for $-1 \leq y \leq 1$ and $4 \leq \log(1 + 54|y|)$ for $|y| > 1$.

We choose $d_\mu = 0$ and $d_\alpha = -4$. Substituting yields

$$2 \left( \exp\left( -\frac{4}{3n} \log\frac{1}{\delta} \right) \underset{x \leftarrow D}{\mathbb{E}} \left( \exp\left( 4\min(\hat{\alpha}x^2, 1) \right) \right) \right)^n$$

$$\leq 2\delta^{4/3} \underset{x \leftarrow D}{\mathbb{E}} (1 + 54\left(\sqrt{\hat{\alpha}}\right)^z |x|^z)^n \qquad \text{by Claim D.6}$$

$$= 2\delta^{4/3}(1 + 54\left( \frac{\log\frac{1}{\delta}}{n\epsilon\sqrt{3\hat{w}}} \right)^z)^n \qquad \text{since } D \text{ has } z^{\text{th}} \text{ moment } 1$$

$$\leq 2\delta^{4/3} \exp\left( n \cdot 54 \left( \frac{\log\frac{1}{\delta}}{n\epsilon\sqrt{3 \cdot 555}} \right)^z \right) \qquad \text{since } 1 + x \leq e^x$$

$$= 2\delta^{4/3} \exp\left( n \cdot 54 \left( \frac{1}{2^{1-\frac{1}{z}}5.6^{\frac{2}{z}-1}\sqrt{1665}} \left( \frac{\log\frac{1}{\delta}}{n} \right)^{\frac{1}{z}} \right)^z \right) \qquad \text{by definition of } \epsilon$$

$$= 2\delta^{4/3} \exp\left( \frac{54}{2^{z-1}5.6^{2-z}\sqrt{1665^z}} \log\frac{1}{\delta} \right)$$

$$\leq 2\delta^{4/3}\delta^{-0.237} \qquad \text{for all values of } 1 < z \leq 2$$

$$\leq 2\delta^{1.096} \leq \frac{\delta}{\log^4\frac{1}{\delta}} \qquad \text{for suff. small } \delta$$

as desired. $\qquad\qquad\qquad\square$

We have thus proven one of two components necessary to prove Proposition D.1. We restate and prove the remaining component, which is a Lipschitz bound over the region covered by our finite mesh over $\hat{w} \in [0.05, 555]$:

**Lemma D.3.** *Consider an arbitrary set of $n$ samples $X$. Consider the expressions $\psi_\mu(X, \hat{\mu}, \hat{\alpha})$, $\psi_\alpha(X, \hat{\alpha})$, reparameterized in terms of $\hat{w} = \frac{\log^2\frac{1}{\delta}}{3\hat{\alpha}n^2\epsilon^2}$ in place of $\hat{\alpha}$. Suppose the equation $\psi_\alpha(X, \hat{\alpha}) = 0$ has a solution in the range $\hat{w} \in [0.05, 555]$. Then the functions $\frac{\log\frac{1}{\delta}}{n\epsilon}\psi_\mu(X, \hat{\mu}, \hat{\alpha})$ and $\psi_\alpha(X, \hat{\alpha})$ are Lipschitz with respect to $\hat{w}$ on the entire interval $\hat{w} \in [0.05, 555]$, with Lipschitz constant $c\log\frac{1}{\delta}$ for some universal constant $c$.*

*Proof.* Consider the derivative with respect to $\hat{w}$ of $\psi_\alpha(X, \hat{\mu}, \hat{\alpha}) = \sum_{i=1}^n \left( \min\left( \frac{\log^2\frac{1}{\delta}}{3n^2\epsilon^2\hat{w}}x_i^2, 1 \right) - \frac{1}{3n}\log\frac{1}{\delta} \right)$. The $\hat{w}$ derivative of $\min\left( \frac{\log^2\frac{1}{\delta}}{3n^2\epsilon^2\hat{w}}x_i^2, 1 \right)$ is either $-\frac{\log^2\frac{1}{\delta}}{3n^2\epsilon^2\hat{w}^2}x_i^2 = -\frac{1}{\hat{w}}\hat{\alpha}x_i^2$ or $0$, depending on which term in the min is the smallest, and in either case has magnitude at most $\frac{1}{\hat{w}}\min(\hat{\alpha}x_i^2, 1)$. Thus, the overall $\hat{w}$ derivative of $\psi_\alpha(X, \hat{\mu}, \hat{\alpha})$ has magnitude at most $\frac{1}{\hat{w}}\sum_i \min(\hat{\alpha}x_i^2, 1)$. Since by assumption $\sum_i \min(\hat{\alpha}x_i^2, 1) = \frac{1}{3}\log\frac{1}{\delta}$ for some $\hat{w} \in [0.05, 555]$, the derivative with respect to $\hat{w}$ must be within a constant factor of $\log\frac{1}{\delta}$ across the entire range, as desired.

Similarly, consider the derivative with respect to $\hat{w}$ of $\psi_\mu(X, \hat{\mu}, \hat{\alpha}) = \sum_{i=1}^n (\hat{\mu} - x_i(1 - \min(\hat{\alpha}x_i^2, 1)))$. The $\hat{w}$ derivative of $(\hat{\mu} - x_i(1 - \min(\hat{\alpha}x_i^2, 1)))$ is either $-\frac{1}{\hat{w}}\hat{\alpha}x_i^3$ or $0$, depending on whether $x_i \leq \sqrt{1/\hat{\alpha}}$, and thus the magnitude of the entire derivative is bounded by $\frac{1}{\hat{w}\sqrt{\hat{\alpha}}}\sum_i \min(\hat{\alpha}x_i^2, 1)$. Since $\sum_i \min(\hat{\alpha}x_i^2, 1)$ is bounded by a constant times $\log\frac{1}{\delta}$, and $\frac{1}{\hat{w}\sqrt{\hat{\alpha}}}$ is bounded by a constant times $\frac{1}{\sqrt{\hat{w}\hat{\alpha}}} = \frac{\sqrt{3}n\epsilon}{\log\frac{1}{\delta}}$ since $\hat{w} \in [0.05, 555]$, the magnitude of the derivative of $\frac{\log\frac{1}{\delta}}{n\epsilon}\psi_\mu(X, \hat{\mu}, \hat{\alpha})$ is bounded by a constant times $\log\frac{1}{\delta}$, as desired. $\qquad\square$

### D.2. Proof of Lemma D.5

In this section, we prove the technical Lemma D.5.

**Lemma D.5.** *For all $\hat{w} \in [0.05, 555]$, there exists $a > 0, b$ such that for all $1 < z \leq 2$ and $y \in \mathbb{R}$, letting $c_z = 2(5.6)^{\frac{1}{z-1}-1}$, the following holds true:*

$$ay(1 - \min(y^2, 1)) - b\min(y^2, 1) \leq \log\left( 1 + ay + |y|^z(3\hat{w})^{\frac{z}{2}}c_z^{z-1}\left( -1 + \frac{a}{\sqrt{3\hat{w}}} - \frac{b}{3} \right) \right) \qquad (4)$$

*where $a$ and $b$ are bounded by constants. Further, for $\hat{w} = 0.05$, the pair $a = 0.75, b = \sqrt{3}$ works.*

*Proof.* We remark that Lemma D.5 is more nuanced than its counterpart in Lee & Valiant (2022), in part due to the introduction of a new variable $z$ in the exponent. However, notice that taking the derivative with respect to $z$ of the right hand side of (4), omitting the outer logarithm—which is a monotone function—and omitting terms independent of $z$, yields

$$\frac{\partial}{\partial z} |y|^z (3\hat{w})^{\frac{z}{2}} c_z^{z-1} = (|y|\sqrt{3\hat{w}})^z \cdot 2^{z-1} \cdot 5.6^{2-z} \cdot \log\left(\frac{2|y|\sqrt{3\hat{w}}}{5.6}\right)$$

whose sign is determined solely by the sign of $\log\left(\frac{2|y|\sqrt{3\hat{w}}}{5.6}\right)$, and thus is independent of $z$. This implies that the right hand side of (4) is monotone in $z$, and thus it suffices to show that (4) holds for the boundary cases $z \to 1$ and $z = 2$. Since

$$\lim_{z \to 1} c_z^{z-1} = \lim_{z \to 1} (2(5.6)^{\frac{1}{z-1}-1})^{z-1} = 5.6,$$

we evaluate the limit $z \to 1$ of (4) by simply substituting $z = 1$ into the equation and replacing the term $c_z^{z-1}$ with 5.6.

We now choose the values of the parameters $a, b$. We follow the choices of Lee & Valiant (2022) for $w < 5.5$: for $w = 0.05$ we set $a = 0.75, b = \sqrt{3}$ as promised in the lemma statement; and for $w \in (0.05, 5.5]$ we choose $a = \frac{-\sqrt{6}+\sqrt{6+96\hat{w}}}{2\sqrt{2\hat{w}}}$, i.e., the positive root of the equation $\sqrt{2\hat{w}}(a^2 - 12) + \sqrt{6}a = 0$, and $b = 3 - \frac{a^2}{2}$. Note that these choices set $a > 0$, as promised in the lemma. For $w \geq 5.5$ we differ from Lee & Valiant (2022) and instead choose the constants $a = 4.2$ and $b = -3$ (independent of $\hat{w}$).

**Case $z = 2$ and $w \in [0.05, 5.5)$:**   In this case our lemma is identical to its counterpart in Lee & Valiant (2022) and thus needs no proof.

**Case $z = 2$ and $w \geq 5.5$:**   In this case we choose $a = 4.2$ and $b = -3$, and thus Equation 4 simplifies to

$$4.2y(1 - \min(y^2, 1)) + 3\min(y^2, 1) \leq \log\left(1 + 4.2y + y^2\sqrt{3\hat{w}} \cdot 2 \cdot 4.2\right)$$

For $y$ outside the range $[-1, 1]$, the first two terms become the constant 3; and since the polynomial inside the logarithm expression on the right hand side is monotonically increasing in $y$ outside of $y \in [-1, 1]$, when $\hat{w} \geq 5.5$, it is sufficient to prove the inequality for $y \in [-1, 1]$.

Further, the logarithm term is the only term that depends on $\hat{w}$, and the logarithm monotonically increases with $\hat{w}$, so thus it is sufficient to prove the inequality for the smallest value of $\hat{w}$, namely $\hat{w} = 5.5$. We thus show:

$$4.2y(1 - y^2) + 3y^2 \leq \log\left(1 + 4.2y + y^2\sqrt{3 \cdot 5.5} \cdot 2 \cdot 4.2\right)$$

To prove this, we note that both sides are smooth for $y \in [-1, 1]$; thus we take derivatives of both sides and set them equal to each other. The derivative of the left hand side is a quadratic polynomial; and the derivative of the right hand side is the multiplicative inverse of a quadratic polynomial. So thus the solutions to this equation are the solutions to a quartic. And it is straightforward to find these four solutions for $y$ with a computer algebra package, and confirm that, at all of these four possible extrema and the points $y = -1, y = 1$, the desired inequality is true.

**Case $z = 1$ and $w \geq 5.5$**   Next, we turn to the case of $z = 1$ and $w \geq 5.5$, where we set $a = 4.2$ and $b = -3$. Equation 4 now simplifies to a condition independent of $\hat{w}$:

$$4.2y(1 - \min(y^2, 1)) + 3\min(y^2, 1) \leq \log\left(1 + 4.2y + |y| \cdot 5.6 \cdot 4.2\right)$$

As above, the expression inside the logarithm on the right hand side is increasing outside the range $y \in [-1, 1]$ and the left hand side is constant outside this range, so it is sufficient to show this inequality for $y \in [-1, 1]$ in which case it simplifies to

$$4.2y(1 - y^2) + 3y^2 \leq \log\left(1 + 4.2y + |y| \cdot 5.6 \cdot 4.2\right)$$

For negative $y$, the left hand side is convex and the right hand side is concave, so the left hand side minus the right hand side is convex, and its maximum must be attained at one of its two endpoints, $y = -1$ or $y = 0$. Numerically checking both cases confirms the inequality for $y \leq 0$. For positive $y$, as in the previous case, we take derivatives of the left and right hand side and set them equal to each other, leaving us with a cubic equation, which thus has closed form solutions. We thus check that the desired inequality is true at all the positive roots of the cubic, along with the extreme points $y = 0, y = 1$.

**Case** $z = 1$ **and** $w \in (0.05, 5.5)$**:** In this case we have chosen $a = \frac{-\sqrt{6} + \sqrt{6 + 96\hat{w}}}{2\sqrt{2\hat{w}}}$—the positive root of the equation $\sqrt{2\hat{w}}(a^2 - 12) + \sqrt{6}a = 0$—and $b = 3 - \frac{a^2}{2}$. Since $a$ is an increasing function of $\hat{w}$ it is easy to check that, for $\hat{w}$ in our range $(0.05, 5.5)$ we have $a \in (0, 3.12]$. Because of these relations between $a, b, \hat{w}$, we can simplify part of the expression in (4) inside the logarithm as

$$\sqrt{3\hat{w}}\left(-1 + \frac{a}{\sqrt{3\hat{w}}} - \frac{b}{3}\right) = \frac{1}{2}a$$

Using this relation, and substituting $b = 3 - \frac{a^2}{2}$ into the left hand side of (4) simplifies (4) to:

$$ay(1 - \min(y^2, 1)) - (3 - \frac{a^2}{2})\min(y^2, 1) \leq \log\left(1 + ay + 2.8a|y|\right)$$

As above, we point out that the left hand side is constant for $y$ outside the range $[-1, 1]$ and the right hand side is monotonic away from this interval, so it suffices to prove the inequality for $y \in [-1, 1]$ in which case it simplifies to

$$ay(1 - y^2) - (3 - \frac{a^2}{2})y^2 \leq \log\left(1 + ay + 2.8a|y|\right) \tag{5}$$

We first show the $y \geq 0$ case. Since $a > 0$ we have $ay \geq 0$. Substituting $ay \to x$ above yields the following inequality, which we show for $x \geq 0$ and $a \in (0, 3.12]$:

$$x + \frac{x^2}{2} - \frac{1}{a^2}(x^3 + 3x^2) \leq \log\left(1 + 3.8x\right)$$

The term $x^3 + 3x^2$ is nonnegative for $x \geq 0$, and thus the left hand side is increasing in $a$. Thus it suffices to prove the inequality for the maximum value of $a = 3.12$, and all $x \geq 0$.

As above, we prove this by pointing out that both sides are smooth functions for $x \geq 0$, so we take their derivatives and set them equal to each other, which yields a cubic equation in $x$. Our inequality takes its extreme values at either a positive root of the cubic, or at the boundary value $x = 0$; we thus confirm the inequality in these cases to prove it in general.

We now show the $y \leq 0$ case.

We point out that the left hand side of Equation 5, $ay(1 - y^2) - (3 - \frac{a^2}{2})y^2$, is increasing in $y$ at $y = 0$, convex for sufficiently negative $y$, and can only transition from convex to concave once. Meanwhile, the right hand side, $\log\left(1 - 1.8ay\right)$, is decreasing everywhere for $y \leq 0$, and concave everywhere.

Let $c$ be the location where the cubic function $ay(1 - y^2) - (3 - \frac{a^2}{2})y^2$ transitions from convex to concave. For those $y$ in the (possibly empty) interval $[c, 0]$, the cubic is concave—since it is also increasing at $y = 0$, it must be increasing on this entire interval. Recalling that the right hand side of Equation 5 is decreasing for $y \leq 0$, the difference between the left and right hand sides attains its maximum in the interval $y \in [c, 0]$ at $y = 0$, which is just 0, satisfying the inequality.

And for those $y < c$, where the left hand side is convex, then the difference between the left and right hand sides is convex, and thus its maximum must occur either at the left extreme, $y = -1$, or the right extreme, $y = c$; however the difference at $y = c$ we already showed was at most the difference at $y = 0$, so overall, the maximum difference between the left and right hand sides must occur at either $y = -1$ or $y = 0$. As above, the $y = 0$ case is trivial, and it remains to prove the $y = -1$ case. For the $y = -1$ case Equation 5 becomes:

$$\frac{a^2}{2} - 3 \leq \log\left(1 + 1.8a\right)$$

The left hand side is convex and the right hand side is concave; so we prove the inequality by numerically checking both endpoints: $a = 0$ and $a = 3.12$.

**Case $z = 1$ and $\hat{w} = 0.05$:**   In this case we choose $a = 0.75$ and $b = \sqrt{3}$ and Equation 4 becomes

$$0.75y(1 - \min(y^2, 1)) - \sqrt{3}\min(y^2, 1) \le \log\left(1 + 0.75y + 2.086|y|\right)$$

where 2.086 is a lower bound on $15\sqrt{3\hat{w}}\left(-1 + \frac{a}{\sqrt{3\hat{w}}} - \frac{b}{3}\right)$

As usual, for $y$ outside $[-1, 1]$ we point out that the left hand side is constant while the right hand side monotonically increases, so it suffices to show the inequality for $y \in [-1, 1]$.

For $y \in [-1, 1]$, we trivially lower bound the right hand side by $\log(1 + 0.75y)$ (dropping the $2.086|y|$ term), and lower bound this logarithm expression with the quadratic $0.75y - 0.75y^2$. The difference between the left hand side and this polynomial lower bound on the right hand side is thus $-.75y^3 + (.75 - \sqrt{3})y^2$, which is negative for $y > 1 - \frac{4}{\sqrt{3}}$, which is below $-1$, and hence our inequality holds on the entire interval $y \in [-1, 1]$.

Thus we have shown all cases of the desired inequality. $\qquad\square$

## E. Proof of Theorem 2.8

In this section, we present the proof of Theorem 2.8. We restate the theorem for completeness:

**Theorem 2.8.** *Let $D$ be a distribution with mean $\mu$ and variance $\sigma^2$.*

*Let $\hat{\mu}$ denote Estimator 1 on input parameter $\delta$ and $n$ i.i.d. samples from $D$. Also let $\bar{X}_n$ denote the sample mean. Then, fixing $\delta$ and $D$ and taking $n \to \infty$, we have*

$$\sqrt{n}\hat{\mu} \xrightarrow{p} \sqrt{n}\bar{X}_n$$

*that is, $|\sqrt{n}\hat{\mu} - \sqrt{n}\bar{X}_n| \xrightarrow{p} 0$, that $\sqrt{n}\hat{\mu}$ converges to $\sqrt{n}\bar{X}_n$ in probability.*

*As a corollary, by the Central Limit Theorem, we have*

$$\sqrt{n}(\hat{\mu} - \mu) \xrightarrow{d} \mathcal{N}\left(0, \sigma^2\right)$$

*That is, $\hat{\mu}$ is asymptotically normal and efficient.*

*Proof.* Without loss of generality, by the shift-and-scale equivariance of both Lee and Valiant's estimator and the sample mean, we assume $\mu = 0$ for notational simplicity, and that the variance of $D$ is 1.

The second part of the theorem follows from the straightforward reasoning that, since Estimator 1 converges to the sample mean (as claimed in the first part of the theorem statement), and since the sample mean converges to a Gaussian (from the Central Limit Theorem), then Estimator 1 also converges to the same Gaussian. Formally, this uses Slutsky's theorem and the fact that convergence in probability implies convergence in distribution. Hence it only remains to show the first part of the theorem, that Lee and Valiant's estimator converges to the sample mean in probability, for fixed $\delta$ and as $n \to \infty$.

The claim that $\sqrt{n}\hat{\mu} \xrightarrow{p} \sqrt{n}\bar{X}_n$ is equivalent by definition to the statement that, for any fixed $\epsilon > 0$,

$$\lim_{n \to \infty} \mathbb{P}(\sqrt{n}|\hat{\mu} - \bar{X}_n| > \epsilon) = 0$$

First, recall that $\hat{\mu}$ differs from the sample mean by removing a total of $\frac{1}{3}\log\frac{1}{\delta}$ weighted samples before taking the average. Thus, $\sqrt{n}|\hat{\mu} - \bar{X}_n|$ is upper bounded by $\Theta\left(\log\frac{1}{\delta}\right)|x_{\max}|/\sqrt{n}$ where $x_{\max}$ is the largest sample in magnitude (recalling that we assumed $\mu = 0$):

$$\sqrt{n}\left|\kappa + \frac{1}{n}\sum_i(x_i - \kappa)(1 - \min(\alpha(x_i - \kappa)^2, 1)) - \frac{1}{n}\sum_i x_i\right|$$

$$= \frac{1}{\sqrt{n}}\left|\sum_i(x_i - \kappa)\min(\alpha(x_i - \kappa)^2, 1)\right|$$

$$\le \frac{2|x_{\max}|}{\sqrt{n}}\sum_i \min(\alpha(x_i - \kappa)^2, 1)$$

$$= \frac{2}{3} \log \frac{1}{\delta} \frac{|x_{\max}|}{\sqrt{n}}$$

Thus, the event $\sqrt{n}|\hat{\mu} - \bar{X}_n| > \epsilon$ implies $|x_{\max}| > \epsilon\sqrt{n}/\Theta(\log\frac{1}{\delta})$, which we now show is an event that occurs with probability $\to 0$ as $n \to \infty$.

We show the following claim, that the probability a *single* sample from $D$ is larger than $\epsilon\sqrt{n}/\Theta(\log\frac{1}{\delta})$ is at most $o(1/n)$.

**Claim E.1.** *Fix any $\epsilon > 0$ and $\delta \in (0,1)$. Suppose we draw a single sample $X$ from a distribution $D$ with mean 0 and variance 1. Then,*

$$\mathbb{P}\left(|X| > \epsilon\sqrt{n}\Big/\Theta\left(\log\frac{1}{\delta}\right)\right) = o\left(\frac{1}{n}\right)$$

*Here, the $o(\cdot)$ is in the limit where $\epsilon$, $\delta$ and $D$ are fixed, and $n \to \infty$.*

We prove Claim E.1 at the end. Claim E.1 implies that the expected number of samples (among the $n$ drawn) with magnitude exceeding $\epsilon\sqrt{n}/\Theta(\log\frac{1}{\delta})$ is $o(1)$. Thus, by Markov's inequality, the probability that at least one sample exceeds that threshold is also $o(1)$, showing Theorem 2.8.

We finish with the short proof of Claim E.1. Here, for a generic non-negative random variable $Y$, we use a refined version of Markov's inequality that

$$\mathbb{P}(Y > a) \leq \frac{\mathbb{E}[Y\mathbb{1}[Y \geq a]]}{a}$$

Applying this to $X^2$ in Claim E.1, we have

$$\mathbb{P}\left(X^2 > \epsilon^2 n \Big/\Theta\left(\log^2\frac{1}{\delta}\right)\right) \leq \mathbb{E}\left[X^2\mathbb{1}\left[X^2 \geq \epsilon^2 n\Big/\Theta\left(\log^2\frac{1}{\delta}\right)\right]\right] \cdot \frac{\Theta\left(\log^2\frac{1}{\delta}\right)}{\epsilon^2 n}$$

Since $\mathbb{E}[X^2] = 1$ and $\epsilon$, $\delta$ and $D$ are fixed, we have $\mathbb{E}\left[X^2\mathbb{1}\left[X^2 \geq \epsilon^2 n/\Theta\left(\log^2\frac{1}{\delta}\right)\right]\right] = o(1)$ as $n \to \infty$. Thus, the right hand side in the above inequality is $o(1/n)$, showing Claim E.1.

This completes the proof that $\hat{\mu}$ converges to $\bar{X}_n$ in probability, showing the theorem. $\square$

## F. Proofs of the Folklore Robustness of Median-of-Means

For completeness, we provide formal proofs of Facts A.18 and A.23, two folklore facts about the robustness of the median-of-means estimator against adversarial corruption, which we use in our proofs of Theorems 2.2 and 2.3.

**Fact A.18** (Folklore). *For any distribution $D$ with mean $\mu$ and standard deviation $\sigma$, let $\tilde{X}$ be a set of $n$ $\eta$-corrupted samples from $D$. The median-of-means estimate $\kappa$ from grouping samples into $O(\log\frac{1}{\delta'} + \eta n)$ buckets, on input $\tilde{X}$, satisfies*

$$\mathbb{P}\left(|\kappa - \mu| \geq O\left(\sigma\sqrt{\frac{\log\frac{1}{\delta'}}{n}} + \eta\right)\right) \leq \frac{1}{16}\delta'$$

*Proof.* Let $k$ denote the chosen amount of buckets. Choose $k = 16(\log\frac{1}{\delta'} + \eta n)$. Let $\mu_i$ denote the mean of the $i$-th bucket *before any adversarial corruption*. Then each $\mu_i$ is independent with variance $\sigma\sqrt{\frac{16(\log\frac{1}{\delta'}+\eta n)}{n}}$. By Chebyshev's inequality, $\mathbb{P}[|\mu_i - \mu| \geq 4\sigma\sqrt{\frac{16(\log\frac{1}{\delta'}+\eta n)}{n}}] \leq \frac{1}{16}$. Let $d_i$ denote the indicator variable for the event $|\mu_i - \mu| \leq 4\sigma\sqrt{\frac{16(\log\frac{1}{\delta'}+\eta n)}{n}}$, then $\mathbb{E}(d_i) \geq \frac{15}{16}$. By Hoeffding's inequality, $\mathbb{P}[\sum_i d_i \leq 9(\log\frac{1}{\delta'} + \eta n)] \leq e^{-4.5(\log\frac{1}{\delta'}+\eta n)} \leq e^{-(\log\frac{1}{\delta'}+\eta n)}$, which is at most $e^{-\log\frac{16}{\delta'}} = \frac{1}{16}\delta'$ for large enough $n$. Note that for $|\kappa - \mu| \geq 4\sigma\sqrt{\frac{16(\log\frac{1}{\delta'}+\eta n)}{n}}$, at most $k/2 = 8(\log\frac{1}{\delta'} + \eta n)$ buckets can have $|\mu_i - \mu| \leq 4\sigma\sqrt{\frac{16(\log\frac{1}{\delta'}+\eta n)}{n}}$. Accounting for the adversarial corruption, which can affect at most $\eta n$ buckets, at most $9(\log\frac{1}{\delta'} + \eta n)$ buckets can have $|\mu_i - \mu| \leq 4\sigma\sqrt{\frac{16(\log\frac{1}{\delta'}+\eta n)}{n}}$ *before adversarial corruption*, which happens with probability at most $\frac{1}{16}\delta'$ as desired. $\square$

**Fact A.23** (Folklore). *For any distribution $D$ with mean $\mu$ and standard deviation $\sigma$, let $\tilde{X}$ be a set of $n$ $\eta$-corrupted samples from $D$. Assuming that $\eta \leq \frac{1}{24n}\log\frac{1}{\delta}$, the median-of-means estimate $\kappa$ from grouping samples into $O(\log\frac{1}{\delta})$ buckets, on input $\tilde{X}$, satisfies*

$$\mathbb{P}\left(|\kappa - \mu| \geq O\left(\sigma\sqrt{\frac{\log\frac{1}{\delta}}{n}}\right)\right) \leq \frac{1}{22}\delta$$

*Proof.* Let $k$ denote the chosen amount of buckets. Choose $k = 16\log\frac{1}{\delta}$. Let $\mu_i$ denote the mean of the $i$-th bucket *before any adversarial corruption*. Then each $\mu_i$ is independent with variance $\sigma\sqrt{\frac{16\log\frac{1}{\delta}}{n}}$. By Chebyshev's inequality, $\mathbb{P}[|\mu_i - \mu| \geq 4\sigma\sqrt{\frac{16\log\frac{1}{\delta}}{n}}] \leq \frac{1}{16}$. Let $d_i$ denote the indicator variable for the event $|\mu_i - \mu| \leq 4\sigma\sqrt{\frac{16\log\frac{1}{\delta}}{n}}$, then $\mathbb{E}(d_i) \geq \frac{15}{16}$. By Hoeffding's inequality, $\mathbb{P}[\sum_i d_i \leq 9\log\frac{1}{\delta}] \leq e^{-4.5\log\frac{1}{\delta}}$, which is at most $e^{-\log\frac{22}{\delta}} = \frac{1}{22}\delta$ for sufficiently small $\delta$. Note that for $|\kappa - \mu| \geq 4\sigma\sqrt{\frac{16\log\frac{1}{\delta}}{n}}$, at most $k/2 = 8\log\frac{1}{\delta}$ buckets can have $|\mu_i - \mu| \leq 4\sigma\sqrt{\frac{16(\log\frac{1}{\delta'} + \eta n)}{n}}$. Accounting for the adversarial corruption, which can affect at most $\eta n \leq \log\frac{1}{\delta}$ buckets, at most $9\log\frac{1}{\delta}$ buckets can have $|\mu_i - \mu| \leq 4\sigma\sqrt{\frac{16\log\frac{1}{\delta}}{n}}$ *before adversarial corruption*, which happens with probability at most $\frac{1}{22}\delta$ as desired. $\square$

