# OpenReview forum: "All-Purpose Mean Estimation over R: Optimal Sub-Gaussianity with Outlier Robustness and Low Moments Performance"
_ICML.cc/2025/Conference — ICML 2025 oral_

### Official Review · Reviewer_vs1w · 2025-03-06

**Overall Recommendation:** 4

**Summary:**

Mean estimation is the following simple task: given samples from a probability distribution D on R, estimate the mean mu of D. Although mean estimation has been studied since the dawn of time, various elementary and fundamental questions about mean estimation are still being tackled.

A natural goal is to obtain optimal *nonasymptotic* errors. That is, for a given number of samples n and error rate delta, design an estimator hat(mu) which satisfies an inequality of the form Pr( |hat(mu) - mu| < error ) \geq 1-delta, for as small a function error(n,delta) as possible. If D is Gaussian, the empirical mean gives the best possible estimator, but since the real world isn't really Gaussian, we would like to obtain estimators with such guarantees, ideally with the function error(n,delta) matching the one we would get for Gaussian D, under much weaker assumptions on D.

Lee and Valiant in 2022 constructed a mean estimator which obtains the optimal error(n,delta) under only the assumption that D has bounded variance. Their estimator even adapts to the variance of D.

While the Lee-Valiant estimator is highly robust in the sense of tolerating a broad class of underlying distributions D, the question this paper addresses is whether it is robust in other senses, in particular:

-- adversarial contamination/data poisoning -- what if a small fraction of the sample are chosen adversarially?
-- nonexistent variance -- what if even the variance of D does not exist?

The authors prove that the Lee-Valiant estimator satisfies strong robustness guarantees in both these senses. First, they show that it is resilient to an \( \eta \)-fraction of adversarially corrupted samples in the strong contamination model, achieving the optimal error bound \( O(\sigma \sqrt{\eta}) \). Second, they establish that when \( D \) has only a finite \( z \)-th moment for \( z \in (1,2) \), the estimator attains the minimax-optimal error rate, matching known lower bounds. Third, they show that it satisfies *neighborhood optimality*, meaning that it adapts to the structure of the underlying distribution and achieves the best possible error beyond worst-case guarantees. Finally, they prove that the estimator is asymptotically normal, converging efficiently to the true mean as the sample size grows.

These results show that the Lee-Valiant estimator is not just optimal in the standard setting but also robust across a range of challenging conditions, making it a strong candidate for practical use over alternatives like median-of-means or trimmed means.

**Claims And Evidence:**

Yes

**Essential References Not Discussed:**

n/a

**Experimental Designs Or Analyses:**

n/a

**Methods And Evaluation Criteria:**

Yes

**Other Comments Or Suggestions:**

non

**Other Strengths And Weaknesses:**

The paper is well written and elegantly argues for the usefulness of the Lee-Valiant estimator.

**Questions For Authors:**

non

**Relation To Broader Scientific Literature:**

Robustness is statistics is a long running theme with thousands of publications. Prior works have developed estimators which achieve all of the guarantees of the lee-valiant estimator separately, but as far as I am aware no single estimator is so far known (until this work) which has the sub-Gaussian error guarantee (including the right constant), robustness to adversarial contamination, and minimax optimal rates under weaker moment assumptions.

**Theoretical Claims:**

no

---

> ### Author Rebuttal · Authors · 2025-04-01
>
> Thank you for your appreciation of our work!

---

### Official Review · Reviewer_UmEB · 2025-03-12

**Overall Recommendation:** 4

**Summary:**

This paper considers the problem of designing "all-purpose"mean estimation algorithms that can be applied to a variety of scenarios. To be specific, the authors consider the estimator by Lee & Valiant (2022). This algorithm, is previously shown to be optimal in the standard finite variance and i.i.d. setting. This paper, further give the result that Lee & Valiant (2022)'s algorithm is optimal in other settings. To be specific, they proved that the algorithm is robust to data corruption, and also optimal for distributions that have infinite variance.

**Claims And Evidence:**

The proof of robustness to corruption is given in Thm 2.2 and 2.3. Optimality under infinite variance is proven in Theorem 2.4.

**Essential References Not Discussed:**

No missing essential references.

**Experimental Designs Or Analyses:**

No experiment in this work.

**Methods And Evaluation Criteria:**

No evaluations is involved by this work. It is a learning theory work.

**Other Comments Or Suggestions:**

I would suggest that the authors give more explanation of why the robustness under the mentioned scenario can be viewed as "all-purpose". It might be good to emphasize why those scenarios are important.

**Other Strengths And Weaknesses:**

Strengths: Proof the optimality of the previously proposed algorithm under some scenarios. The optimality results is important to the community.

Weakness: The term “all-purpose” may be an overstatement. I am not sure why the robustness under certain scenario can be regarded as "all=purpose".

**Questions For Authors:**

The studied estimator is 1-d. Is there multi-dimensional estimator proven, or shown in experiment, to be efficient?

Is there any other setups that worth considering? Other than contamination and infinite variance?

**Relation To Broader Scientific Literature:**

Not sure how this work is related to broader scientific literature.

**Theoretical Claims:**

I didn't check the proof correctness in detail.

---

> ### Author Rebuttal · Authors · 2025-04-01
>
> Thank you for your positive review of our paper. We answer your questions below.
>
> **Beyond 1-d**: Currently, even the construction of a 2-d mean estimator achieving the analogous guarantees of Fact 1.1 (i.e. with sharp constants) is an *open problem* in the field, so no such estimator is known at this moment. On the other hand, Lee and Valiant have a different paper named "Optimal Sub-Gaussian Mean Estimation in Very High Dimensions", which studies the regime where the "effective dimension" of a distribution is much much larger $\log^2 1/\delta$, and they yield $1+o(1)$-tight estimation error. The robustness of that estimator is basically trivial to analyze, although the extra error term is different from the $\sqrt{\eta}$ term we get in 1-d: that other estimator does not handle low-dimensional distributions well, and the $\sqrt{\eta}$ term from data corruption is a "low-dimensional phenomenon".
>
> **Other setups, the phrase "all-purpose"**: We chose to study adversarial corruption, distributions with infinity variance, and asymptotic normality, because these are compelling and fundamental notions of performance that one could hope for in a mean estimator. These properties are the focus of widely known or folklore results on mean estimation. Thus we are motivated to ask whether all these properties can still hold in addition to the sharp performance guarantee achieved by the recent Lee-Valiant estimator. There are (of course) many other settings that are worth considering, for example what if the samples are not i.i.d., but many desiderata might be incompatible with the extremely strong 1+o(1) performance guarantee of the Lee-Valiant estimator. As for the name "all-purpose", we also weren't completely happy with the naming -- we wanted to convey the meaning of "swiss-army knife" but without the clunkiness of the phrase. We welcome any suggestions!

---

### Official Review · Reviewer_eHht · 2025-03-13

**Overall Recommendation:** 4

**Summary:**

This paper concerns the problem of estimating the mean of i.i.d. real-valued samples.
The authors studies an estimator due to Lee & Valiant 2022 and show that this estimator enjoys several properties not known before.
These include
- optimal robustness against adversarial outliers
- optimal accuracy for heavy-tailed distributions
- optimal adaptation to unknown distributions

**Claims And Evidence:**

All claims come with rigorous proofs.

**Essential References Not Discussed:**

NA

**Experimental Designs Or Analyses:**

NA

**Methods And Evaluation Criteria:**

NA

**Other Comments Or Suggestions:**

NA

**Other Strengths And Weaknesses:**

1. I'm not sure I understand the difference between Theorems 2.2 and 2.3.
Besides the technical nuances in $\delta'$ vs. $\delta$, what's the fundamental reason to prove one in addition to the other?
They seem to have essentially the same quality given that neither $222$ in Theorem 2.2 nor $135$ in Theorem 2.3 is expected to be sharp.

1. Line 146-149 "Comparing Theorem 2.2 with Theorem 2.3, the former asks for a failure probability $\delta' > \delta$, but correspondingly has a smaller sub-Gaussian error term (since $\log\frac{1}{\delta'} < \log\frac{1}{\delta}$)".
Do you mean "latter" instead of "former" since there is no $\delta'$ in Theorem 2.2.

1. After reading the first 8 pages, I'm still confused about what neighborhood optimality means.
It was not really defined in Definition 2.5.
I suggest the authors formally define this somewhere in the main text.

1. The convergence equation $\sqrt{n} \hat{\mu} \to \sqrt{n} \bar{X}$ in Theorem 2.8 and elsewhere is inaccurate since the RHS is $n$-dependent.
Do you mean something like $|\sqrt{n} \hat{\mu} - \sqrt{n} \bar{X}|\to 0$?

**Questions For Authors:**

1. Unless I missed it, this paper doesn't seem to discuss at all what happens for mean estimation beyond *one dimension*, which is a natural question to ask and potentially more relevant to modern statistics.
Robustness guarantees for high-dimensional mean estimation can be quite hard to derive.
Even defining the notion of median is nontrivial, as briefly alluded to in the paper.
I wonder if anything in the spirit of Lee & Valiant 2022 can be (or has been?) done.

**Relation To Broader Scientific Literature:**

NA

**Theoretical Claims:**

I didn't check the proofs in the appendix.

---

> ### Author Rebuttal · Authors · 2025-04-01
>
> Thank you for your positive review and questions. Here, we answer the questions raised.
>
> **Thm 2.2 vs 2.3**: The difference is indeed subtle, but neither theorem implies the other, and since different readers might consider one or the other "more natural", we included both, to avoid leaving readers with lingering questions. (We consider Theorem 2.2 as being more natural; but 2.3 is more similar to what has appeared in prior literature.)
>
> **Line 146-149**: Thanks for catching the typo
>
> **Neighborhood Optimality**: Yes, we unfortunately had to make the conscious choice to omit the full definition, given the submission page limit. The definition is rather subtle, and is an adaptation of "local minimax" from the statistics literature. Essentially, the notion means "local admissibility" or "local Pareto efficiency" --- the performance of an estimator is neighborhood optimal if, even after we restrict attention to the "local neighborhood" of a distribution $D$, no other estimator can outperform our estimator simultaneously on all distributions in that neighborhood. We will include the definition (including the choices of local neighborhoods) to the main paper if accepted, using the extra page space.
>
> **Convergence equation**: we use the notation $X \to Y$ to mean $|X - Y| \to 0$ as you pointed out. We originally thought it is a reasonably common notation, but we are happy to further clarify in the paper to remove ambiguity.
>
> **Beyond 1-d**: Currently, even the construction of a 2-d mean estimator achieving the analogous guarantees of Fact 1.1 (i.e. with sharp constants) is an *open problem* in the field, so no such estimator is known at this moment. On the other hand, Lee and Valiant have a different paper named "Optimal Sub-Gaussian Mean Estimation in Very High Dimensions", which studies the regime where the "effective dimension" of a distribution is much much larger $\log^2 1/\delta$, and they yield $1+o(1)$-tight estimation error. The robustness of that estimator is basically trivial to analyze, although the extra error term is different from the $\sqrt{\eta}$ term we get in 1-d: that other estimator does not handle low-dimensional distributions well, and the $\sqrt{\eta}$ term from data corruption is a "low-dimensional phenomenon".

---

### Official Review · Reviewer_aQTr · 2025-03-19

**Overall Recommendation:** 4

**Summary:**

This paper can be seen as a follow up for the seminal work of Lee & Valiant (2022), which proposed an optimal mean estimator for distributions over the set of real values, i.e. $\mathbb{R}$. Based on that, this paper further shows that the mean estimator of Lee & Valiant (2022) is ``all-purpose'', that is, it is robust to an $\eta$-fraction of corruptions under the strong contamination model; meanwhile, for heavy-tailed distribution with bounded $z$-th moment ($z\in(1,2)$), it has optimal estimation error comparing to a lower bound in Devroye et al. (2016). Futhermore, it also shows that all estimators with similar error guarantees imply the neighborhood optimal, hence solving an open question raised in Dang et al. (2023). Finally, the paper also proves that the estimator is asymptotically normal and efficient, further supporting its performance in many learning scenarios.

**Claims And Evidence:**

All claims are very clear with convincing evidence.

**Essential References Not Discussed:**

Sufficiently discussed.

**Experimental Designs Or Analyses:**

N/A

**Methods And Evaluation Criteria:**

The method, or the algorithm, follows from prior work. The main contribution is the theoretical analysis for robust settings.

**Other Comments Or Suggestions:**

N/A

**Other Strengths And Weaknesses:**

The paper brings very strong theoretical guarantees for robust mean estimation in one-dimensional space. It provides insights for the optimality of the estimator proposed by Lee & Valiant (2022) and median-of-means. The theoretical analysis and discussion look sound to me and the paper is well written.

**Questions For Authors:**

Can you comment on the broad impact of the analysis for the estimation of heavy-tailed distributions? In addition, if these techniques are applicable to higher-dimensional estimation?

**Relation To Broader Scientific Literature:**

This work solves an important problem by extending the performance guarantees of the mean estimator Th proposed by Lee & Valiant (2022) to robust mean estimation literature. More importantly, it shows that the nice feature of the optimality in leading constant terms of the error rate still carries over to the robust setting.

**Theoretical Claims:**

The theoretical theorems and lemmas, proof ideas, look sound to me. The algorithm remains unchanged. The paper aims to show that even when the algorithm learns from a corrupted set of data samples $\tilde{X}$, an important ``influence parameter'' $\tilde{\alpha}$ calculated based on $\tilde{X}$ will be lower bounded, hence upper bounding the influence from the corrupted samples.

---

> ### Author Rebuttal · Authors · 2025-04-01
>
> Thank you for your appreciation of our work. To answer your question regarding high(er)-dimensional mean estimation: currently, even the construction of a 2-d mean estimator achieving the analogous guarantees of Fact 1.1 (i.e. with sharp constants) is an *open problem* in the field, so no such estimator is known at this moment. Furthermore, the neighborhood optimality notion is also only understood in 1-d. Hence, it's hard to say if our techniques will directly extend to high(er) dimensions, since our analyses do depend on the precise estimator being studied.
>
> That said, asymptotic normality should be easiest to prove for any eventual 2-d mean estimator with sharp constant guarantees, since it's a much easier guarantee than Fact 1.1-style results. Also, we speculate that our low-moment analysis of the 1-d Lee-Valiant estimator can plausibly extend to 2-d (or constant-d), if the eventual 2-d estimator subsumes the 1-d Lee-Valiant estimator and is also naturally analyzed via the maximin linear programming games capturing Chernoff Bounds (cf Section 6.2).
>
> About the broader impact of our analysis: while we hope other authors will be intrigued by our results and try to prove analogous results in other settings, the techniques in our paper are very tailored to the specifics of the estimator---for the sake of a tight analysis. Let us know if there are more specific aspects of this that you might like to see discussed in the final paper.

---

### Decision · Program_Chairs · 2025-05-01

**Decision:**

Accept (oral)

**Comment:**

**Summary.**

The topic of this paper is 1D robust mean estimation. Lee and Valiant (2022) proposed an estimator for the mean which is optimal and tight down to the leading constant for the class of distributions with finite variance (algorithm reported in Estimator 1 and guarantee reported in Fact 1.1.).

In this paper, the authors demonstrate through careful analysis that the estimator (without any modification to it):

* Achieves optimal guarantees in the adversarial contamination model.

* Achieves optimal guarantees for the class of distributions with finite $z$th moment for $z \in (1,2)$ (specifically, it achieves the lower bound in Devroye et al., 2016 up to constants).

* Achieves so-called “neighborhood optimality” (Dang et al., 2023).

* Is asymptotic normal and efficient (in the finite variance, uncontaminated setting).

Following the above, the authors deduce that the Lee and Valiant’s estimator is “all purpose”, that is, it should be the one used in practice, in lieu of more classical algorithms such as median-of-means or trimmed mean.

**Strengths.**

* The problem is of strong interest to the ICML community (and the more general ML community).

* No soundness concerns were raised, and the writing is clear.

* Novelty: this is the first time an estimator was found to satisfy optimality to the leading constant in the finite variance setting, and is at the same time robust to adversarial contamination and optimal for distributions with infinite variance.

* The main claim that the estimator is “all purpose” is convincing; this is particularly important for practioners, as it removes the need to make additional assumptions on their data.

**Weaknesses.**

The presented analysis may not extend to higher dimensions (aQTr , eHht, UmEB); however, devising optimal estimator in dimensions higher than one is known to be a hard problem in the field.

**Discussion and reviewer consensus.**

There is strong consensus among the reviewer that the paper should be accepted.

**Additional remarks.**

Some minor points were raised by eHht, I recommend the authors to include more details on neighborhood optimality in their final version if the paper gets accepted.

**Overall evaluation.**

This is a serious contribution to the ICML program; I recommend acceptance.